

SciPost Phys. Lect. Notes 10 (2019)

# Lectures on faster-than-light travel and time travel

**Barak Shoshany**

Perimeter Institute for Theoretical Physics,
31 Caroline St. N., Waterloo, ON, Canada, N2L 2Y5

⋆ bshoshany@perimeterinstitute.ca

## Abstract

These lecture notes were prepared for a 25-hour course for advanced undergraduate students participating in Perimeter Institute's Undergraduate Summer Program. The lectures cover some of what is currently known about the possibility of superluminal travel and time travel within the context of established science, that is, general relativity and quantum field theory. Previous knowledge of general relativity at the level of a standard undergraduate-level introductory course is recommended, but all the relevant material is included for completion and reference. No previous knowledge of quantum field theory, or anything else beyond the standard undergraduate curriculum, is required. Advanced topics in relativity, such as causal structures, the Raychaudhuri equation, and the energy conditions are presented in detail. Once the required background is covered, concepts related to faster-than-light travel and time travel are discussed. After introducing tachyons in special relativity as a warm-up, exotic spacetime geometries in general relativity such as warp drives and wormholes are discussed and analyzed, including their limitations. Time travel paradoxes are also discussed in detail, including some of their proposed resolutions.

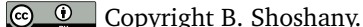

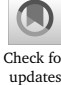
# 1   Introduction

> "Space is big. Really big. You just won't believe how vastly, hugely, mind-bogglingly big it is. I mean, you may think it's a long way down the road to the chemist, but that's just peanuts to space."
>
> – Douglas Adams, *The Hitchhiker's Guide to the Galaxy*

In science fiction, whenever the plot encompasses more than one solar system, faster-than-light travel is an almost unavoidable necessity. In reality, however, it seems that space travel is limited by the speed of light. In fact, even just accelerating to a significant fraction of the speed of light is already a hard problem by itself, e.g. due to the huge energy requirements and the danger of high-speed collisions with interstellar dust. However, this is a problem of

---

[1] The author would like to thank Daniel Gottesman and Aephraim Steinberg for discussions which proved helpful in writing this section.

engineering, not physics – and we may assume that, given sufficiently advanced technology, it could eventually be solved.

Unfortunately, even once we are able to build and power a spaceship that can travel at close to the speed of light, the problem still remains that interstellar distances are measured in light years – and therefore will take years to complete, no matter how close we can get to the speed of light. The closest star system, Alpha Centauri, is roughly 4.4 light years away, and thus the trip would take at least 4.4 years to complete. Other star systems with potentially habitable exoplanets are located tens, hundreds, or even thousands of light years away; the diameter of the Milky Way galaxy is estimated at $175 \pm 25$ thousand light years[2].

For a one-way trip, the long time it takes to reach the destination may not be an insurmountable obstacle. First of all, humanity is already planning to send people to Mars, a journey which is estimated to take around 9 months. Thus it is not inconceivable to send people on a journey which will take several years, especially if technological advances make the trip more tolerable.

Second, relativistic time dilation means that, while for an observer on Earth it would seem that the spaceship takes at least 4.4 years to complete the trip to Alpha Centauri, an observer on the spaceship will measure an arbitrarily short proper time on their own clock, which will get shorter the closer they get to the speed of light[3].

Furthermore, both scientists and science fiction authors have even imagined journeys lasting hundreds or even thousands of years, while the passengers are in suspended animation in order for them to survive the long journey. Others have envisioned generation ships, where only the distant descendants of the original passengers actually make it to the destination[4].

However, while a trip lasting many years may be possible – and, indeed, might well be the **only** way humankind could ever realistically reach distant star systems, no matter how advanced our technology becomes – this kind of trip is only feasible for initial colonization of distant planets. It is hard to imagine going on vacation to an exotic resort on the planet Proxima Centauri b in the Alpha Centauri system, when the journey takes 4.4 years or more. Moreover, due to the relativistic time dilation mentioned above, when the tourists finally arrive back on Earth they will discover that all of their friends and relatives have long ago died of old age – not a fun vacation at all!

So far, the scenarios mentioned are safely within the realm of established science. However, science fiction writers often find them to be too restrictive. In a typical science-fiction scenario, the captain of a spaceship near Earth might **instantaneously** receives news of an alien attack on a human colony on Proxima Centauri b and, after a quick 4.4-light-year journey using the ship's warp drive, will arrive at the exoplanet just in time to stop the aliens[5]. Such scenarios require faster-than-light communication and travel, both of which are considered by most to be disallowed by the known laws of physics.

Another prominent staple of science fiction is the concept of time travel[6]. However, any use of a time machine seems to bluntly violate the principle of causality, and inherently bring upon irreconcilable paradoxes such as the so-called grandfather paradox. Unfortunately, works

---

[2]And, for intergalactic travel, the Andromeda galaxy, for example, is $2.54 \pm 0.11$ **millions** of light years away. But let's solve one problem at a time...

[3]Recall that the time dilation factor is given (in units where $c \equiv 1$) by $\gamma = 1/\sqrt{1 - v^2}$, which is equal to 1 for $v = 0$ but approaches infinity in the limit $v \to 1$, that is, as the velocity approaches the speed of light.

[4]Another possibility is that, by the time humanity develops interstellar travel, humans will also have much longer lifespans and/or will be able to inhabit artificial bodies, which would make a journey lasting decades not seem so long any more.

[5]And, speaking of aliens, most scenarios where aliens from a distant planet visit Earth assume the aliens are capable of superluminal travel.

[6]In these notes, by "time travel" we will always mean travel to the **past**. Travel to the future is trivial – you can just sit and wait to time to pass, or use special or general relativistic time dilation to make it pass faster – and does not violate causality or create any paradoxes.

of science fiction which treat time travel paradoxes in a logical and consistent manner are extremely rare, and this is no surprise given that even us physicists don't really understand how to make it consistent. As we will see below, time travel paradoxes may, in fact, be resolved, but a concrete mathematical model of paradox-free time travel has not yet been constructed.

In writing these notes, I relied heavily on the three excellent books on the subject by Visser [1], Lobo [2] and Krasnikov [3], as well as the popular general relativity textbooks by Carroll [4], Wald [5], Hawking & Ellis [6], and Poisson [7]. Many of the definitions, proofs and discussions are based on the material in these books. The reader is also encouraged to read Baez & Muniain [8] for a great introduction to relevant concepts in differential geometry.

Importantly, throughout these notes we will be using *Planck units*[7] , where $c = G = \hbar = 1$, and our metric signature of choice will be $(-, +, +, +)$.

## 2 An Outline of General Relativity

### 2.1 Basic Concepts

#### 2.1.1 Manifolds and Metrics

Let spacetime be represented by a 4-dimensional *pseudo-Riemannian manifold M* equipped with a *metric* **g** of signature[8] $(-, +, +, +)$. The metric is a symmetric tensor with components $g_{\mu\nu}$ in some coordinate system[9] $\{x^{\mu}\}$, where $\mu, \nu \in \{0, 1, 2, 3\}$. It has the *line element*[10]

$$\mathrm{d}s^2 = g_{\mu\nu} \, \mathrm{d}x^{\mu} \otimes \mathrm{d}x^{\nu}, \tag{1}$$

and determinant $g \equiv \det \mathbf{g}$.

The expression $\mathrm{d}s^2$ may be understood as the infinitesimal and curved version of the expression $\Delta s^2 = -\Delta t^2 + \Delta x^2 + \Delta y^2 + \Delta z^2$ for the *spacetime interval* between two points in special relativity, where $\Delta$ represents the coordinate difference between the points. In a curved spacetime, the difference between points loses its meaning since one cannot compare vectors at two separate tangent spaces without performing a parallel transport (see Sec. 2.3.1).

The simplest metric is the *Minkowski metric*, which is diagonal:

$$\eta_{\mu\nu} \equiv \mathrm{diag}(-1, 1, 1, 1) = \begin{pmatrix} -1 & 0 & 0 & 0 \\ 0 & 1 & 0 & 0 \\ 0 & 0 & 1 & 0 \\ 0 & 0 & 0 & 1 \end{pmatrix}. \tag{2}$$

It describes a flat spacetime, and its line element, in the Cartesian coordinates $\{t, x, y, z\}$, is simply

$$\mathrm{d}s^2 = -\mathrm{d}t^2 + \mathrm{d}x^2 + \mathrm{d}y^2 + \mathrm{d}z^2. \tag{3}$$

---

[7]Another popular convention in the literature is the *reduced Planck units* where instead of $G = 1$ one takes $8\pi G = 1$. This simplifies some equations (e.g. the Einstein equation becomes $G_{\mu\nu} = T_{\mu\nu}$ instead of $G_{\mu\nu} = 8\pi T_{\mu\nu}$) but complicated others (e.g. the coefficient of $-\mathrm{d}t^2$ in the Schwarzschild metric becomes $1 - M/4\pi r$ instead of $1 - 2M/r$). Here we will take $G = 1$ which seems like the more natural choice, since it continues the trend of setting fundamental dimensionful constant to 1.

[8]This means that **g** has one negative and three positive eigenvalues.

[9]We will use a popular abuse of notation where $g_{\mu\nu}$ will be called "the metric", even though the metric is actually a tensor **g** which happens to have these components in some coordinate system. Similarly, $x^{\mu}$ will be called a "vector" even though the vector is actually **x**, and so on.

[10]Here we are, of course, using the *Einstein summation convention*, where we automatically sum over any index which is repeated twice: once as an **upper** index and once as a **lower** index. So for example, $x_{\mu}y^{\mu} \equiv \sum_{\mu=0}^{3} x_{\mu}y^{\mu}$ and $g_{\mu\nu}x^{\nu} \equiv \sum_{\nu=0}^{3} g_{\mu\nu}x^{\nu}$. The index which appears twice is said to be *contracted*.

### 2.1.2 Tangent Spaces and Vectors

At every point $p \in M$ there is a *tangent space* $T_p M$, consisting of *tangent vectors* to the manifold at that particular point. For example, if $M$ is a sphere, the tangent space is the plane which intersects that sphere at exactly one point $p$, and the tangent vectors are vectors in that plane.

Given two tangent vectors $u^\mu$ and $v^\mu$, the metric imposes the *inner product* and *norm squared*

$$\langle \mathbf{u}, \mathbf{v} \rangle \equiv g_{\mu\nu} u^\mu v^\nu, \qquad |\mathbf{v}|^2 \equiv \langle \mathbf{v}, \mathbf{v} \rangle = g_{\mu\nu} v^\mu v^\nu. \tag{4}$$

Since the manifold is not Riemannian, the norm squared is not positive-definite; it can be negative, positive, or even zero for a non-zero vector. Thus it doesn't really make sense to talk about "the norm" $|\mathbf{v}|$, since it can be imaginary, and when we say "the norm" we will always mean the norm squared. Given a tangent vector $v^\mu$, if $|\mathbf{v}|^2 < 0$ it is called *timelike*, if $|\mathbf{v}|^2 > 0$ it is called *spacelike*, and if $|\mathbf{v}|^2 < 0$ it is called *lightlike* or *null*[11].

Note that we have been using upper indices for vectors. We now define *covectors* (or *1-forms*), which have components with lower indices, and act on vectors to produce a scalar. The metric relates a vector to a covector by *lowering an index*, for example $u_\nu = g_{\mu\nu} u^\mu$. We then find that the inner product (4) may be written as the product[12] of a vector with a covector: $\langle \mathbf{u}, \mathbf{v} \rangle \equiv g_{\mu\nu} u^\mu v^\nu = u_\nu v^\nu$. Similarly, for the metric, we have used two lower indices, and wrote it as a matrix. The corresponding inverse matrix gives us the components of the *inverse metric* $g^{\mu\nu}$, which satisfies $g^{\mu\lambda} g_{\lambda\nu} = \delta^\mu_\nu$.

Finally, if we assign a tangent vector to every point in the manifold, then we have a *vector field* $v^\mu(\mathbf{x})$. Similarly, we may talk about a *scalar field* $\phi(\mathbf{x})$, which assigns a number to every point, a covector field $v_\mu(\mathbf{x})$, a tensor field $g_{\mu\nu}(\mathbf{x})$, and so on.

### 2.1.3 Curves and Proper Time

Let $x^\mu(\lambda)$ be a *curve* (or *path*, or *worldline*) parametrized by some real parameter $\lambda$. For each value of $\lambda$, $x^\mu(\lambda)$ is a point on the manifold. This point has a tangent space, and the tangent vector to the curve, defined as

$$\dot{x}^\mu \equiv \frac{\mathrm{d}x^\mu}{\mathrm{d}\lambda}, \tag{5}$$

is a vector in this tangent space.

If $x^\mu(\lambda)$ describes the worldline of a massive particle, then its tangent vector is timelike everywhere; $|\dot{\mathbf{x}}(\lambda)|^2 < 0$ for all $\lambda$. In this case, we may calculate the *proper time* along the path, which is the *coordinate-independent* time as measured by a clock moving with the particle, and thus is an observable; in contrast, the *coordinate time* $x^0 \equiv t$ is not an observable since it, of course, depends on the arbitrary choice of coordinates.

The differential of proper time is defined as minus the line element, that is, $\mathrm{d}\tau^2 \equiv -\mathrm{d}s^2$ or[13]

$$\mathrm{d}\tau^2 = -g_{\mu\nu} \mathrm{d}x^\mu \otimes \mathrm{d}x^\nu. \tag{6}$$

---

[11]Some people use the convention where the metric signature has the opposite sign, $(+,-,-,-)$. In this case, the definitions for timelike and spacelike have the opposite signs as well. Basically, "timelike" means "has a norm squared with the same sign as the time dimension in the metric signature", and similarly for "spacelike".

[12]This explains why, in Footnote 10, we said that the summation convention only works if the same index appears once as an upper index and once as a lower index. The summation simply gives us the inner product between the contracted vector and covector, or more generally, between the corresponding component of two tensors. One cannot contract two upper indices, since the inner product in that case would require first adding the metric to the expression.

[13]The minus sign comes from the fact that the time dimension has a minus sign in the metric signature. Thus, if we want the difference in proper time between a past event and a future event to be positive, we must cancel that minus sign.

Employing a slight abuse of notation, we may "divide" this expression by $d\lambda^2$ where $\lambda$ is the (arbitrary) curve parameter:

$$\frac{d\tau^2}{d\lambda^2} = \left(\frac{d\tau}{d\lambda}\right)^2 = -g_{\mu\nu}\frac{dx^\mu}{d\lambda}\frac{dx^\nu}{d\lambda} = -g_{\mu\nu}\dot{x}^\mu\dot{x}^\nu = -|\dot{\boldsymbol{x}}|^2. \tag{7}$$

Thus, we learn that

$$\frac{d\tau}{d\lambda} = \sqrt{-|\dot{\boldsymbol{x}}|^2} \implies d\tau = \sqrt{-|\dot{\boldsymbol{x}}|^2}\,d\lambda, \tag{8}$$

where the expression inside the square root is positive since $|\dot{\boldsymbol{x}}|^2 < 0$ for a timelike path. Now we can find the total proper time $\tau$ along a path (that is, from one value of $\lambda$ to a subsequent value of $\lambda$) by integrating this differential:

$$\tau \equiv \int d\tau = \int \sqrt{-|\dot{\boldsymbol{x}}|^2}\,d\lambda = \int \sqrt{-g_{\mu\nu}\dot{x}^\mu\dot{x}^\nu}\,d\lambda. \tag{9}$$

This allows us, in principle, to calculate $\tau$ as a function of $\lambda$, and then use the proper time in place of $\lambda$ as the parameter for our curve. When using the proper time as a parameter – and **only** then! – the tangent vector $\dot{x}^\mu \equiv dx^\mu/d\tau$ is called the *4-velocity*, and it is automatically normalized[14] to $|\dot{\boldsymbol{x}}|^2 = -1$. From now on, unless stated otherwise, we will **always** parametrize timelike paths with proper time, since the normalization $|\dot{\boldsymbol{x}}|^2 = -1$ simplifies many calculations, and allows us to talk about 4-velocity in a well-defined way.

### 2.1.4 Massless Particles

So far we have discussed massive particles, whose worldlines are timelike. The worldline of a massless particle, on the other hand, is a null path. For a null path we **cannot** define a proper time, since by definition its tangent is a null vector, so $|\dot{\boldsymbol{x}}(\lambda)|^2 = 0$ for any choice of parameter $\lambda$ (although, of course, the tangent vector itself is not the zero vector). Therefore, from (9) we have that $\tau = 0$ between any two points on the path. This is why we sometimes say that massless particles, such as photons, "do not experience the passage of time"; the proper time along their worldlines vanishes.

This means that for null paths, there is no preferred parameter; however, as we will see below, null geodesics possess a family of preferred parameters called *affine parameters*. Furthermore, note that massless particles do **not** have a well-defined 4-velocity, since the definition of 4-velocity makes use of the proper time.

## 2.2 Covariant Derivatives and Connections

### 2.2.1 Defining Tensors

A function **T** defined on a manifold is called a *tensor* of *rank* $(p,q)$ if its components have $p$ upper indices and $q$ lower indices: $T^{\mu_1\cdots\mu_p}_{\nu_1\cdots\nu_q}$, and if, under a transformation from a coordinate system $\{x^\mu\}$ to another one $\{x^{\mu'}\}$, each upper index $\mu_i$ receives a factor of $\partial x^{\mu'_i}/\partial x^{\mu_i}$ and each lower index $\nu_i$ receives a factor of $\partial x^{\nu_i}/\partial x^{\nu'_i}$ (note that the original and primed coordinates switch places). Vectors and covectors are specific cases of tensors, with rank $(1,0)$ and $(0,1)$ respectively.

---

[14]Indeed, using the chain rule and (7), we have

$$|\dot{\boldsymbol{x}}|^2 = g_{\mu\nu}\frac{dx^\mu}{d\tau}\frac{dx^\nu}{d\tau} = \frac{g_{\mu\nu}\frac{dx^\mu}{d\lambda}\frac{dx^\nu}{d\lambda}}{\left(\frac{d\tau}{d\lambda}\right)^2} = \frac{g_{\mu\nu}\frac{dx^\mu}{d\lambda}\frac{dx^\nu}{d\lambda}}{-g_{\mu\nu}\frac{dx^\mu}{d\lambda}\frac{dx^\nu}{d\lambda}} = -1. \tag{10}$$

For example, the components of a vector $v^\mu$, a covector $u_\mu$ and a rank $(0, 2)$ tensor $g_{\mu\nu}$ transform as follows[15]:

$$v^{\mu'} = \frac{\partial x^{\mu'}}{\partial x^\mu} v^\mu, \qquad u_{\mu'} = \frac{\partial x^\mu}{\partial x^{\mu'}} u_\mu, \qquad g_{\mu'\nu'} = \frac{\partial x^\mu}{\partial x^{\mu'}} \frac{\partial x^\nu}{\partial x^{\nu'}} g_{\mu\nu}. \tag{11}$$

If the tensors transforms this way, we say that they are *covariant* or *transform covariantly*. This is very important, since the **abstract** quantities **v**, **u** and **g** are guaranteed to be *invariant* under the transformation if and only if their **components** transform exactly in this way.

### 2.2.2 Covariant Derivatives

The standard way to define differentiation on a manifold is by using the *covariant derivative* $\nabla_\mu$, which generalizes the usual partial derivative $\partial_\mu \equiv \partial / \partial x^\mu$ and is defined[16] as follows:

- On a scalar field $\phi$: $\nabla_\mu \phi \equiv \partial_\mu \phi$.

- On a vector field $v^\nu$: $\nabla_\mu v^\nu \equiv \partial_\mu v^\nu + \Gamma^\nu_{\mu\lambda} v^\lambda$.

- On a covector $u_\nu$: $\nabla_\mu u_\nu \equiv \partial_\mu u_\nu - \Gamma^\lambda_{\mu\nu} u_\lambda$.

- On a rank $(p, q)$ tensor **T** with components $T^{\nu_1 \cdots \nu_p}_{\sigma_1 \cdots \sigma_q}$: The first term of $\nabla_\mu T^{\nu_1 \cdots \nu_p}_{\sigma_1 \cdots \sigma_q}$ will be $\partial_\mu T^{\nu_1 \cdots \nu_p}_{\sigma_1 \cdots \sigma_q}$, and to that we **add** one term $\Gamma^{\nu_i}_{\mu\lambda} T^{\nu_1 \cdots \lambda \cdots \nu_p}_{\sigma_1 \cdots \sigma_q}$ for each upper index $\nu_i$, and **subtract** one term $\Gamma^\lambda_{\mu\sigma_i} T^{\nu_1 \cdots \nu_p}_{\sigma_1 \cdots \lambda \cdots \sigma_q}$ for each lower index $\sigma_i$, generalizing the expressions above.

$\Gamma^\lambda_{\mu\nu}$ are called the *connection coefficients*. Importantly, $\Gamma^\lambda_{\mu\nu}$ are **not** the components of a tensor, since they do **not** transform covariantly under a change of coordinates. However, the partial derivative itself does not transform as a tensor either, and it just so happens that the unwanted terms from each of the transformations exactly cancel each others, so while $\partial_\mu T^{\nu_1 \cdots \nu_p}_{\sigma_1 \cdots \sigma_q}$ is **not** a tensor, the covariant derivative $\nabla_\mu T^{\nu_1 \cdots \nu_p}_{\sigma_1 \cdots \sigma_q}$ of any tensor **is** itself a tensor, of rank $(p, q + 1)$. This is exactly why covariant derivatives are so important: partial derivatives do not transform covariantly in a general curved spacetime, and thus do not generate tensors, but covariant derivatives do.

### 2.2.3 The Levi-Civita Connection

Just like the partial derivative, the covariant derivative satisfies linearity and the Leibniz rule. It also commutes with contractions, meaning that if we contract two indices of a tensor, e.g. $T^{\mu\lambda}{}_{\lambda\nu} \equiv g_{\lambda\rho} T^{\mu\lambda\rho}{}_\nu$, and then apply the covariant derivative, then in the new tensor (of one higher rank) that we have formed, the same two indices will still be contracted: $\nabla_\sigma T^{\mu\lambda}{}_{\lambda\nu} = g_{\lambda\rho} \nabla_\sigma T^{\mu\lambda\rho}{}_\nu$. However, the covariant derivative is not unique since it depends on the choice of $\Gamma^\lambda_{\mu\nu}$. A **unique** choice of connection coefficients may be obtained by requiring that the connection is also:

- *Torsion-free*, meaning that the *torsion tensor* $T^\lambda{}_{\mu\nu} \equiv \Gamma^\lambda_{\mu\nu} - \Gamma^\lambda_{\nu\mu} \equiv 2\Gamma^\lambda_{[\mu\nu]}$ vanishes, or equivalently, the connection coefficients are symmetric in their lower indices: $\Gamma^\lambda_{\mu\nu} = \Gamma^\lambda_{\nu\mu}$.

---

[15]You don't need to remember where the primes are placed, since they are located in the only place that makes sense for contracting with the components of the tensor, taking into account that an **upper** index in the denominator, e.g. the $\mu$ on $\partial / \partial x^\mu$, counts as a **lower** index – since $\partial_\mu \equiv \partial / \partial x^\mu$ – and thus should be contracted with an **upper** index.

[16]Some sources use a notation where a comma indicates partial derivatives: $\partial_\mu u_\nu \equiv u_{\nu,\mu}$, and a semicolon indicates covariant derivatives: $\nabla_\mu u_\nu \equiv u_{\nu;\mu}$. We will not use this notation here.

- Metric-compatible, that is, the covariant derivative of the metric vanishes: $\nabla_\lambda g_{\mu\nu} = 0$. Note that this is a stronger condition than commuting with contractions, since we can also write $T^{\mu\lambda}{}_{\lambda\nu} \equiv \delta^\rho_\lambda T^{\mu\lambda}{}_{\rho\nu}$, so commuting with contractions merely implies that $\nabla_\mu \delta^\rho_\lambda = 0$, which one would expect given that the Kronecker delta $\delta^\rho_\lambda$ is just the identity matrix.

With these additional constraints, there is one unique connection, called the *Levi-Civita connection*, whose coefficients are sometimes called *Christoffel symbols*, and are given as a function of the metric by:

$$\Gamma^\lambda_{\mu\nu} = \frac{1}{2} g^{\lambda\sigma} \left( \partial_\mu g_{\nu\sigma} + \partial_\nu g_{\mu\sigma} - \partial_\sigma g_{\mu\nu} \right). \tag{12}$$

The reader is encouraged to prove this and all other unproven claims in this section.

## 2.3 Parallel Transport and Geodesics

### 2.3.1 Parallel Transport

Since each point on the manifold has its own tangent space, it is unclear how to relate vectors (or more generally, tensors) at different points, since they belong to different vector spaces. Again, imagine the sphere, with the tangent spaces being planes which touch it only at one point. A tangent vector to one plane, if moved to another point, will not be tangent to the plane at that point.

A unique way of taking a tensor from one point to another point on the manifold is given by *parallel transport*. Let $x^\mu(\lambda)$ be a curve, and let $v^\mu$ be a vector whose value is known on a particular point on the curve (e.g. at $\lambda = 0$). To parallel transport $v^\mu$ to another point along the curve (e.g. $\lambda = 1$), we solve the equation

$$\dot{x}^\mu \nabla_\mu v^\nu = 0, \tag{13}$$

where $\dot{x}^\mu \equiv dx^\mu/d\lambda$. Using the definition of the covariant derivative, this may be written explicitly as

$$\dot{x}^\mu \partial_\mu v^\sigma + \Gamma^\sigma_{\mu\nu} \dot{x}^\mu v^\nu = 0. \tag{14}$$

Furthermore, using the chain rule $\frac{dx^\mu}{d\lambda} \partial_\mu = \frac{d}{d\lambda}$, we may write this as

$$\frac{d}{d\lambda} v^\sigma + \Gamma^\sigma_{\mu\nu} \dot{x}^\mu v^\nu = 0. \tag{15}$$

Similar parallel transport equations are easily obtained for tensors of any rank.

### 2.3.2 Geodesics and Affine Parameters

Let us apply parallel transport to the tangent vector to the same curve we are parallel-transporting along, that is, take $v^\nu \equiv \dot{x}^\nu$. Then we get:

$$\dot{x}^\mu \nabla_\mu \dot{x}^\sigma = 0 \quad \implies \quad \ddot{x}^\sigma + \Gamma^\sigma_{\mu\nu} \dot{x}^\mu \dot{x}^\nu = 0. \tag{16}$$

This equation is called the *geodesic equation*, and it generalizes the notion of a "straight line" to a curved space. Indeed, for the flat Minkowski space, we have $\Gamma^\sigma_{\mu\nu} = 0$ and thus the geodesics are given by curves satisfying $\ddot{x} = 0$, which describe straight lines.

Equation (16) demands that the vector tangent to the curve is parallel transported in the direction of the curve. This means that the resulting vector must have the same direction and magnitude. We can, in fact, weaken this condition and allow the resulting vector to have a different magnitude, while still demanding that its direction remains unchanged. This still

captures the idea behind geodesics, namely that they are a generalization of straight lines in flat space. The resulting equation takes the form:

$$\dot{x}^{\mu}\nabla_{\mu}\dot{x}^{\sigma} = \alpha\dot{x}^{\sigma}, \tag{17}$$

where $\alpha$ is some function on the curve. However, any curve which satisfies this equation may be reparametrized so that it satisfies (16). Indeed, (17) may be written explicitly as follows:

$$\frac{d^2 x^{\sigma}}{d\lambda^2} + \Gamma^{\sigma}_{\mu\nu}\frac{dx^{\mu}}{d\lambda}\frac{dx^{\nu}}{d\lambda} = \alpha\frac{dx^{\sigma}}{d\lambda}. \tag{18}$$

Now, consider a curve which solves (17) with some parameter $\lambda$ and introduce a new parameter $\mu(\lambda)$. Then we have

$$\frac{d}{d\lambda} = \frac{d\mu}{d\lambda}\frac{d}{d\mu} \quad \Longrightarrow \quad \frac{d^2}{d\lambda^2} = \frac{d^2\mu}{d\lambda^2}\frac{d}{d\mu} + \left(\frac{d\mu}{d\lambda}\right)^2\frac{d^2}{d\mu^2}, \tag{19}$$

and we may rewrite the equation as follows:

$$\frac{d^2\mu}{d\lambda^2}\frac{dx^{\sigma}}{d\mu} + \left(\frac{d\mu}{d\lambda}\right)^2\frac{d^2 x^{\sigma}}{d\mu^2} + \Gamma^{\sigma}_{\mu\nu}\left(\frac{d\mu}{d\lambda}\right)^2\frac{dx^{\mu}}{d\mu}\frac{dx^{\nu}}{d\mu} = \alpha\frac{d\mu}{d\lambda}\frac{dx^{\sigma}}{d\mu}. \tag{20}$$

Rearranging, we get

$$\frac{d^2 x^{\sigma}}{d\mu^2} + \Gamma^{\sigma}_{\mu\nu}\frac{dx^{\mu}}{d\mu}\frac{dx^{\nu}}{d\mu} = \left(\frac{d\lambda}{d\mu}\right)^2\left(\alpha\frac{d\mu}{d\lambda} - \frac{d^2\mu}{d\lambda^2}\right)\frac{dx^{\sigma}}{d\mu}. \tag{21}$$

Therefore the right-hand side will vanish if $\mu(\lambda)$ is a solution to the differential equation

$$\frac{d^2\mu}{d\lambda^2} = \alpha(\lambda)\frac{d\mu}{d\lambda}. \tag{22}$$

Such a solution always exists, and thus we have obtained the desired parameterization. The parameter $\mu$ for which the geodesic equation reduces to the form (16) is called an *affine parameter*. Note that this is in fact a whole family of parameters, since any other parameter given by $\nu \equiv A\mu + B$ with $A, B$ real constants is also an affine parameter, as can be easily seen from (22).

The geodesic equation is one of the two most important equations in general relativity; the other is Einstein's equation, which we will discuss in Sec. 2.6.

### 2.3.3 Massive Particles and Geodesics

A *test particle* is a particle which has a negligible effect on the curvature of spacetime. Such a particle's path will always be a *timelike geodesic* if it's a massive particle (such as an electron), or a *null geodesic* if it's a massless particle (such as a photon), as long as it is in *free fall* – meaning that no forces act on it other than gravity.

For a massive particle with *(rest) mass*[17] $m$ and 4-velocity $\dot{x}^{\mu} = dx^{\mu}/d\tau$, the *4-momentum* is given by $p^{\mu} \equiv m\dot{x}^{\mu}$. Recall again that the 4-velocity is only defined if the curve is parametrized by the proper time $\tau$. Massless particles have neither mass, nor 4-velocity, since proper time is undefined for a null geodesic. Therefore the definition $p^{\mu} \equiv m\dot{x}^{\mu}$ would not make sense, and

---

[17]Some textbooks also define a "relativistic mass" which depends on the velocity or frame of reference. However, this concept is not useful in general relativity (and even in special relativity it mostly just causes confusion). In these notes, as in most of the theoretical physics literature, the rest mass $m$ is assumed to be a constant which is assigned to the particle once and for all. For example, the electron has roughly $m = 511\,\text{keV}$, or the pure number $m = 4.2 \times 10^{-23}$ in Planck units, independently of its velocity or reference frame.

we simply define the 4-momentum as the tangent vector with respect to some affine parameter: $p^\mu \equiv dx^\mu/d\lambda$.

We will sometimes write the geodesic equation in terms of the particle's 4-momentum as follows:

$$p^\nu \nabla_\nu p^\mu = 0. \tag{23}$$

In other words, an unaccelerated (free-falling) particle keeps moving in the direction of its momentum. An observer moving with 4-velocity $u^\mu$ then measures the energy of the particle to be

$$E \equiv -p_\mu u^\mu. \tag{24}$$

As a simple example, consider a flat spacetime with the Minkowski metric $\eta_{\mu\nu} \equiv \text{diag}(-1,1,1,1)$. A massive particle with 4-momentum $p^\mu \equiv m\dot{x}^\mu$ is measured by an observer at rest, that is, with 4-velocity $u^\mu = (1,0,0,0)$. The energy measured will then be

$$E = -\eta_{\mu\nu}p^\mu u^\nu = -\eta_{00}p^0 u^0 = p^0. \tag{25}$$

In other words, in this case the energy is simply the time component of the 4-momentum. Motivated by this result, we take the 4-momentum to be of the form $p^\mu \equiv (E, \vec{p})$ where $\vec{p} \equiv (p^1, p^2, p^3)$ is the (spatial[18]) 3-momentum. Since $|\dot{x}|^2 = -1$, the norm of the 4-momentum is given by

$$|\mathbf{p}|^2 = m^2 |\dot{x}|^2 = -m^2. \tag{26}$$

On the other hand, by direct calculation we find

$$|\mathbf{p}|^2 = -E^2 + |\vec{p}|^2, \tag{27}$$

where $|\vec{p}|^2 \equiv (p^1)^2 + (p^2)^2 + (p^3)^2$. Comparing both expressions, we see that

$$E^2 = m^2 + |\vec{p}|^2. \tag{28}$$

In the rest frame of the particle, where $\vec{p} = 0$, this reduces to the familiar *mass-energy equivalence* equation $E = mc^2$ (with $c = 1$).

### 2.3.4 Massless Particles and the Speed of Light

For massless particles, the situation is simpler. We again take $p^\mu \equiv (E, \vec{p})$, so an observer at rest with $u^\mu = (1,0,0,0)$ will measure the energy $E$. Furthermore, since by definition $p^\mu \equiv \dot{x}^\mu$ and $\dot{x}^\mu$ is null, we have

$$|\mathbf{p}|^2 = |\dot{x}|^2 = 0, \tag{29}$$

and combining with (27) we find that $E^2 = |\vec{p}|^2$. Again, this is the familiar equation $E = pc$, with $c = 1$. We conclude that the relation (28) applies to all particles, whether massive or massless.

Even though a massless particle doesn't have a 4-velocity, we know that it **locally** (i.e. at the same point as the observer) always moves at the speed of light, $v = c = 1$. This is true both in special and general relativity. It is easy to see in a flat spacetime. The line element is

$$ds^2 = -dt^2 + dx^2 + dy^2 + dz^2. \tag{30}$$

Let us assume the massless particle is moving at this instant in the $x$ direction (if it isn't, then we can rotate our coordinate system until it is). Then, since the $y$ and $z$ coordinates remain

---

[18]We will use bold font, $\mathbf{v}$, for spacetime 4-vectors and an arrow, $\vec{v}$, for spatial 3-vectors.

unchanged, we have $\mathrm{d}y = \mathrm{d}z = 0$. Furthermore, since the particle is moving along a null path, it has $\mathrm{d}s^2 = 0$ as well. In conclusion, we have

$$0 = -\mathrm{d}t^2 + \mathrm{d}x^2, \tag{31}$$

which may be rearranged into

$$\frac{\mathrm{d}x}{\mathrm{d}y} = \pm 1. \tag{32}$$

In other words, the particle is moving at the speed of light 1, either in the positive or negative $x$ direction.

However, if the particle was massive, we would have $\mathrm{d}s^2 < 0$ since it is moving along a timelike path. Therefore we have

$$\mathrm{d}s^2 = -\mathrm{d}t^2 + \mathrm{d}x^2 < 0, \tag{33}$$

which may be rearranged into

$$\left| \frac{\mathrm{d}x}{\mathrm{d}t} \right| < 1. \tag{34}$$

Hence, the particle is necessarily moving locally at strictly slower than the speed of light.

So far, we have only considered special relativity. In general relativity we have an arbitrarily curved spacetime, and naively, it seems that massless particles may move at any speed. This is easy to see by considering, for example, the following metric:

$$\mathrm{d}s^2 = -V^2 \mathrm{d}t^2 + \mathrm{d}x^2, \tag{35}$$

where $V$ is some real number. Then for a massless particle we have $\mathrm{d}s^2 = 0$ and thus

$$\frac{\mathrm{d}x}{\mathrm{d}t} = \pm V, \tag{36}$$

so the speed of the particle is given by the arbitrary number $V$. The problem here is that we have calculated the **coordinate** speed of the particle, not its the **local** speed. General relativity works in any coordinate system – this is called *general covariance* or *diffeomorphism invariance* – and the coordinate speed will naturally depend on the choice of coordinate system.

The fact that massless particles always **locally** move at the speed of light easily follows from the well-known result that at a particular point $p$ it is always possible to transform to *locally inertial coordinates*[19], which have the property that $g_{\mu\nu}(p) = \eta_{\mu\nu}$ and $\partial_\sigma g_{\mu\nu}(p) = 0$. Then, as far as the observer at $p$ is concerned, spacetime is completely flat in their immediate vicinity, and thus they will see massless particles passing by at the speed of light.

Finally, it is important to note that if a particle starts moving along a timelike or null geodesic it can never suddenly switch to moving along a different type of geodesic or path; this is simply due to the mathematical fact that the parallel transport preserves the norm of the tangent vector to the path. Indeed, if the tangent vector being parallel transported is $v^\mu$, then $\dot{x}^\mu \nabla_\mu v^\nu = 0$ and thus

$$\dot{x}^\mu \nabla_\mu \left( |\mathbf{v}|^2 \right) = g_{\alpha\beta} \dot{x}^\mu \nabla_\mu \left( v^\alpha v^\beta \right) = g_{\alpha\beta} \left( v^\beta \left( \dot{x}^\mu \nabla_\mu v^\alpha \right) + v^\alpha \left( \dot{x}^\mu \nabla_\mu v^\beta \right) \right) = 0, \tag{37}$$

so $|\mathbf{v}|^2$ is constant along the path.

---

[19]See e.g. [4] for the details of how to construct such coordinates.

### 2.3.5 Inertial Motion, Maximization of Proper Time, and the Twin "Paradox"

It is important to clarify that particles follow geodesics only if they are in *inertial motion*, or *free-falling*. By this we mean that the only force acting on the particle is that of gravity, and there are no other forces which would influence the particle's trajectory. If non-gravitational forces act on the particle – or, for example, a spaceship uses its rockets – then it will no longer follow a geodesic.

Now, in Euclidean space, geodesics minimize distance. However, in a curved space with Lorentzian signature, geodesics instead **maximize** proper time – at least for massive particles, since for massless particles the proper time vanishes by definition. This can be seen from the fact that the geodesic equation may be derived by demanding that the variation of the proper time integral (9) vanishes[20].

The two facts we have just mentioned provide an elegant solution to the famous twin "paradox" of special relativity. In this "paradox" there are two twins, Alice and Bob. Alice stays on Earth while Bob goes on a round trip to a nearby planet, traveling at a significant fraction of the speed of light. When Bob returns to Earth, the twins discover that Alice is now older than Bob, due to relativistic time dilation.

The "paradox" lies in the naive assumption that, since from Bob's point of view he was the one who stayed in place while Alice was the one moving with respect to him, then Bob would expect himself to be the older twin. However, obviously both twins cannot be older than each other, which leads to a "paradox". The solution to the "paradox" lies in the fact that Alice remained in **inertial motion** for the entire time, while Bob was **accelerating** and thus not in inertial motion, hence there is an asymmetry between the twins.

Alternatively, if one does not wish to complicate things by involving acceleration, we may assume that all of the accelerations involved (speeding up, slowing down, and turning around) were instantaneous. Then Bob was also in inertial motion for the entire time, except for the moment of **turnaround**.

At that moment, the inertial frame for the outbound journey is replaced with a completely different inertial frame for the return journey. That one singular moment of turnaround is alone responsible for the asymmetry between the points of view of the twins. It is easy to see by drawing spacetime diagrams – as the reader is encouraged to do – that at the moment of turnaround, Bob's notion of simultaneity (that is, surfaces of constant $t$) changes dramatically. From Bob's point of view, Alice ages instantaneously at the moment!

This is the standard solution from the special relativistic point of view. However, now that we know about general relativity and geodesics, we may provide a much simpler solution. Since Alice remains in inertial motion, and the only forces that act on her are gravitational forces, she simply follows a timelike geodesic. On the other hand, since Bob is using non-gravitational forces (e.g. rockets) to accelerate himself, he will **not** follow a timelike geodesic. Of course, he will still follow a timelike **path**, but the path will not be a geodesic.

In other words, both Alice and Bob's paths in spacetime begin and end at the same point, but Alice follows a timelike geodesic while Bob follows a timelike path which is not a geodesic. Since timelike geodesics are exactly the timelike paths which maximize proper time, the proper time experienced by Alice must be larger than the proper time experienced by Bob. Therefore, Alice must be the older twin.

---

[20]See e.g. [4], chapter 3.3.

## 2.4 Curvature

### 2.4.1 The Riemann Curvature Tensor

The *Riemann curvature tensor* is defined as the commutator[21] of the action of two covariant derivatives on a vector:

$$R^{\rho}{}_{\sigma\mu\nu}v^{\sigma} = \left[\nabla_{\mu}, \nabla_{\nu}\right]v^{\rho}. \tag{38}$$

Since the covariant derivative facilitates parallel transport, this can – roughly speaking – be understood as taking the vector $v^{\rho}$ from some point $p$ along a path given by $\nabla_{\mu}\nabla_{\nu}v^{\rho}$ to a nearby point $q$, and then taking it back from $q$ along a path given by $-\nabla_{\nu}\nabla_{\mu}v^{\rho}$ to $p$. In other words, we take $v^{\rho}$ along a loop. If spacetime was flat, we would expect that the vector $v^{\rho}$ would remain unchanged after going around the loop. However, if spacetime is curved, there is a difference between the initial $v^{\rho}$ and the final $v^{\rho}$ (note that both are in the same tangent space $T_p M$, so we may compare them). This difference is encoded in the Riemann tensor $R^{\rho}{}_{\sigma\mu\nu}$.

The full coordinate expression for the Riemann tensor in terms of the connection coefficients may be calculated from the definition, and it is given by:

$$R^{\rho}{}_{\sigma\mu\nu} = \partial_{\mu}\Gamma^{\rho}_{\nu\sigma} - \partial_{\nu}\Gamma^{\rho}_{\mu\sigma} + \Gamma^{\rho}_{\mu\lambda}\Gamma^{\lambda}_{\nu\sigma} - \Gamma^{\rho}_{\nu\lambda}\Gamma^{\lambda}_{\mu\sigma}. \tag{39}$$

If we lower the first index with the metric ($R_{\rho\sigma\mu\nu} = g_{\lambda\rho}R^{\lambda}{}_{\sigma\mu\nu}$), the resulting tensor satisfies the following identities:

- Symmetry and anti-symmetry under exchange of indices: $R_{\rho\sigma\mu\nu} = -R_{\sigma\rho\mu\nu} = -R_{\rho\sigma\nu\mu} = R_{\mu\nu\rho\sigma}$.

- *First Bianchi identity*: $R_{\rho[\sigma\mu\nu]} = 0$ or more explicitly[22] $R_{\rho\sigma\mu\nu} + R_{\rho\mu\nu\sigma} + R_{\rho\nu\sigma\mu} = 0$.

- *Second Bianchi identity*: $\nabla_{[\lambda}R_{\rho\sigma]\mu\nu} = 0$.

### 2.4.2 Related Tensors: Ricci and Weyl

By contracting the first and third index of the Riemann tensor, we obtain the *Ricci tensor*:

$$R_{\mu\nu} \equiv R^{\lambda}{}_{\mu\lambda\nu}. \tag{40}$$

Note that it is symmetric, $R_{\mu\nu} = R_{\nu\mu}$. The trace of the Ricci tensor is the *Ricci scalar*:

$$R \equiv R^{\mu}{}_{\mu} \equiv g^{\mu\nu}R_{\mu\nu}. \tag{41}$$

We may also define (in 4 dimensions[23]) the *Weyl tensor*:

$$C_{\rho\sigma\mu\nu} \equiv R_{\rho\sigma\mu\nu} - g_{\rho[\mu}R_{\nu]\sigma} + g_{\sigma[\mu}R_{\nu]\rho} + \frac{1}{3}g_{\rho[\mu}g_{\nu]\sigma}R. \tag{43}$$

The Weyl tensor has all the symmetries of the Riemann tensor (including the first Bianchi identity), but it is completely traceless: it vanishes upon contraction of any pair of indices.

---

[21]Note that if there is torsion, that is $\Gamma^{\lambda}_{\mu\nu} \neq \Gamma^{\lambda}_{\nu\mu}$, then one must add a term $2\Gamma^{\lambda}_{[\mu\nu]}\nabla_{\lambda}v^{\rho}$ to the right-hand side of this equation, where $2\Gamma^{\lambda}_{[\mu\nu]} \equiv \Gamma^{\lambda}_{\mu\nu} - \Gamma^{\lambda}_{\nu\mu}$.

[22]Here $6R_{\rho[\sigma\mu\nu]} \equiv R_{\rho\sigma\mu\nu} + R_{\rho\mu\nu\sigma} + R_{\rho\nu\sigma\mu} - R_{\rho\mu\sigma\nu} - R_{\rho\nu\mu\sigma} - R_{\rho\sigma\nu\mu}$ is the usual anti-symmetrizer, and then we use the anti-symmetry in the last two indices.

[23]For a spacetime of dimension $d$ we have

$$C_{\rho\sigma\mu\nu} \equiv R_{\rho\sigma\mu\nu} - \frac{2}{d-2}\left(g_{\rho[\mu}R_{\nu]\sigma} - g_{\sigma[\mu}R_{\nu]\rho} - \frac{1}{d-1}g_{\rho[\mu}g_{\nu]\sigma}R\right). \tag{42}$$

## 2.5 Extrinsic Curvature

### 2.5.1 Intrinsic and Extrinsic Curvature

Let us now consider a surface embedded in a higher-dimensional space. The Riemann tensor describes the *intrinsic curvature* of the surface; this is the curvature within the surface itself, which exists intrinsically, regardless of any embedding (as long as the embedding is isometric, see below).

In contrast, *extrinsic curvature* is the curvature of the surface which comes from the way in which it is embedded, and from the particular space in which it is embedded. Intrinsic curvature comes from the parallel transport of vectors **tangent** to (a curve on) the surface, while extrinsic curvature comes from parallel transport of vectors **normal** to the surface.

For example, a flat piece of paper has no intrinsic curvature. If we draw a triangle on it, the angles will sum to $\pi$. However, if we roll that paper into a cylinder, then as seen from the (flat) higher-dimensional space in which we live, the angles of the triangle will no longer sum to $\pi$. Thus, the surface has acquired an extrinsic curvature.

However, the intrinsic curvature is still flat, since within the surface itself, the edges of the triangle are still geodesics – regardless of its embedding into the higher-dimensional space. Therefore, the angles still sum to $\pi$ when viewed from inside the surface. In other words, the intrinsic curvature is completely independent of any embedding of the surface.

Let us see this with a concrete calculation. We take a cylinder with circumference $2\pi L$. It can be obtained by "rolling up" $\mathbb{R}^2$, i.e. by performing the periodic identification

$$(x, y) \sim (x, y + 2\pi L). \tag{44}$$

Topologically, this is homeomorphic[24] to $\mathbb{R} \times S^1$. The metric is inherited from the original (unrolled) $\mathbb{R}^2$, and is thus flat. Now, let us embed the cylinder in $\mathbb{R}^3$. Introducing cylindrical coordinates $(r, \phi, z)$, the metric on $\mathbb{R}^3$ takes the form

$$ds^2 = dr^2 + r^2 d\phi^2 + dz^2. \tag{45}$$

Taking a constant $r = L$ and identifying the points

$$(L, \phi, z) \sim (L, \phi + 2\pi, z), \tag{46}$$

we get the same cylinder with circumference $2\pi L$, and it has the induced metric (with $dr = 0$ since $r$ is constant)

$$ds^2\big|_{r=L} = L^2 d\phi^2 + dz^2. \tag{47}$$

Since the components of this induced metric are constant, the intrinsic curvature of the cylinder is zero.

### 2.5.2 Isometric and Non-Isometric Embeddings

Let $\left(M, g_{\mu\nu}\right)$ and $\left(N, h_{\mu\nu}\right)$ be two Riemannian manifolds, with metrics $g_{\mu\nu}$ and $h_{\mu\nu}$. An *immersion* between the manifolds is a differentiable function $F : M \to N$ with an injective (one-to-one) derivative. An *embedding* of $M$ into $N$ is an injective immersion such that $M$ is homeomorphic to its image $f(M)$. The embedding is called *isometric* if it also preserves the metric, that is, $g_{\mu\nu} = f^*h_{\mu\nu}$ where $f^*h_{\mu\nu}$ is the pullback[25] of $h_{\mu\nu}$ by $f$.

---

[24]A *homeomorphism* between topological spaces $X$ and $Y$ is a continuous function that has a continuous inverse. The importance of homeomorphisms is that they preserve the topological properties of the space. If there is a homeomorphism between two spaces, we say that they are *homeomorphic* to each other.

[25]The *pullback* $f^*\mathbf{T}$ of a tensor $\mathbf{T}$ by the map $f : M \to N$ literally "pulls back" $\mathbf{T}$ from $N$ into the source manifold $M$. In the case of a metric $h_{\mu\nu}$, which is a rank $(0, 2)$ tensor, the pullback acts on its components as follows:

$$(f^*h(\mathbf{x}))_{\mu\nu} = \frac{\partial x^\alpha}{\partial y^\mu} \frac{\partial x^\beta}{\partial y^\nu} h_{\alpha\beta}(\mathbf{y}), \tag{48}$$

The embedding of the cylinder $\mathbb{R} \times S^1$ into $\mathbb{R}^3$ is isometric, since the induced metric on $\mathbb{R}^3$ is flat, and thus equal to the original metric on the cylinder. Therefore, the intrinsic curvature remains unchanged after the embedding. However, there are cases when an embedding forces us to change the intrinsic curvature of a manifold. This happens when we cannot embed our manifold in another one without stretching or bending it.

An illustrative example is given by the torus, which may be obtained from $\mathbb{R}^2$ in a similar manner to the cylinder, except that now **both** coordinates are identified, with periods $2\pi L_1$ and $2\pi L_2$:

$$(x, y) \sim (x + 2\pi L_1, y + 2\pi L_2). \tag{49}$$

Topologically, this is homeomorphic to $S^1 \times S^1$. The metric inherited from $\mathbb{R}^2$ is, of course, still flat. However, let us now embed the torus $\mathbb{R}^3$, with the same cylindrical coordinates as before. The surface will be defined as the set of points solving the equation

$$z^2 + (r - L_1)^2 = L_2^2. \tag{50}$$

Indeed, for each value of $\phi$, this equation defines a circle with radius $L_2$ centered at $(r, z) = (L_1, 0)$. The surface of revolution of a circle is a torus; it can also be seen as a cylinder whose top and bottom have been glued together. Thus $L_1$ is the *major radius*, or the distance from the $z$ axis to the center of the circle, while $L_2$ is the *minor radius*, that of the circle that is being revolved.

Let us now isolate $z$:

$$z = \pm\sqrt{L_2^2 - (r - L_1)^2}. \tag{51}$$

Then

$$dz = -\frac{(r - L_1)\,dr}{\sqrt{L_2^2 - (r - L_1)^2}}, \tag{52}$$

and by plugging this into the flat metric in cylindrical coordinates on $\mathbb{R}^3$, (45), we obtain the following induced metric:

$$ds^2\big|_{z^2+(r-L_1)^2=L_2^2} = \frac{L_2^2}{L_2^2 - (r - L_1)^2}dr^2 + r^2 d\phi^2, \tag{53}$$

where $L_1 - L_2 \leq r \leq L_1 + L_2$. This metric is **not** flat, as can be checked e.g. by calculating the Ricci scalar, which is

$$R = \frac{2(r - L_1)}{L_2^2 r}. \tag{54}$$

Thus, in this case the intrinsic curvature is not flat after the embedding.

In the case of the cylinder, all we did when we embedded it in $\mathbb{R}^3$ was to take a flat plane and glue two opposite ends of it together. This can be easily illustrated by taking a flat piece of paper and bending it to create a cylinder. However, for the torus, we started with a cylinder – which is still intrinsically flat – and glued its top and bottom together. This cannot be done without stretching the paper (try it!), resulting in an intrinsic curvature that is no longer flat. In other words, this embedding is not isometric[26].

### 2.5.3 Surfaces and Normal Vectors

Let $\Sigma$ be a surface embedded in a (higher-dimensional) manifold $M$. This surface may be defined, for example, by an equation of the form $f(\mathbf{x}) = 0$, as we did for the torus. Then the vector field $\xi^\mu \equiv \nabla^\mu f = \partial^\mu f$ is everywhere normal to the surface, meaning that it is

---

where $\mathbf{y}$ are coordinates on $N$ and $\mathbf{x}$ are coordinates on $M$.

[26]Note, however, that it is possible to isometrically embed the torus in $\mathbb{R}^4$.

orthogonal to every vector tangent to $\Sigma$ (we will not prove this here). The nature of the surface will have the opposite sign to that of the normal vector: if $\xi^\mu$ is timelike the surface is spacelike, if $\xi^\mu$ is spacelike the surface is timelike, and if $\xi^\mu$ is null the surface is also null.

If $\xi^\mu$ is timelike or spacelike, we may normalize it and define a unit normal vector:

$$n^\mu \equiv \frac{\xi^\mu}{\sqrt{|\xi_\lambda \xi^\lambda|}}. \tag{55}$$

If $\xi^\mu$ is null, then it is not only normal but also tangent to $\Sigma$, since it is orthogonal to itself. Thus the *integral curves* $x^\mu(\lambda)$ to the vector field $\xi^\mu$, which are curves such that their tangent vectors are equal to $\xi^\mu$ at every point along the curve:

$$\frac{\mathrm{d}x^\mu}{\mathrm{d}\lambda} = \xi^\mu, \tag{56}$$

are null curves contained in $\Sigma$. Now, we have

$$\xi^\mu \nabla_\mu \xi_\nu = \xi^\mu \nabla_\mu \partial_\nu f = \xi^\mu \left( \partial_\mu \partial_\nu f + \Gamma^\lambda_{\mu\nu} \partial_\lambda f \right) = \xi^\mu \nabla_\nu \partial_\mu f = \xi^\mu \nabla_\nu \xi_\mu = \frac{1}{2} \nabla_\nu \left( \xi^\mu \xi_\mu \right), \tag{57}$$

where we used the fact that $\Gamma^\lambda_{\mu\nu}$ is symmetric in its lower indices. Unfortunately, $\xi^\mu \xi_\mu$ does not necessarily vanish outside of $\Sigma$, so we cannot conclude that the last term vanishes. However, if it does not vanish, we simply redefine the defining equation $f(\mathbf{x}) = 0$ for the surface using $f(\mathbf{x}) \equiv \xi^\mu \xi_\mu$, and then the last term in (57) is simply $\frac{1}{2} \nabla_\nu f(\mathbf{x})$, which is normal to the surface and thus must be proportional to $\xi^\mu$! If the proportionality function is $\alpha$, then we get

$$\xi^\mu \nabla_\mu \xi_\nu = \alpha \xi_\nu, \tag{58}$$

which is the generalized geodesic equation (17). As we have seen when discussing that function, we may reparameterize the geodesic using an affine parameter $\lambda$ such that $\xi^\mu \nabla_\mu \xi_\nu = 0$. The null geodesics $x^\mu(\lambda)$ defined by (56) are called the *generators* of the null surface $\Sigma$, since the surface is the union of these geodesics.

### 2.5.4 The Projector on the Surface

Let $M$ be a manifold with metric $g_{\mu\nu}$. We define the *projector* on the surface $\Sigma$ with unit normal $n^\mu$ as follows:

$$P_{\mu\nu} \equiv g_{\mu\nu} - |\mathbf{n}|^2 n_\mu n_\nu. \tag{59}$$

Note that $|\mathbf{n}|^2 \equiv n_\lambda n^\lambda = \pm 1$ with $+$ for timelike and $-$ for spacelike surfaces. This symmetric tensor projects any vector field $v^\mu$ in $M$ to a tangent vector $P^\mu_\nu v^\nu$ on $\Sigma$. We can see this by showing that the projected vector is orthogonal to the normal vector:

$$n^\mu \left( P_{\mu\nu} v^\nu \right) = n^\mu v_\mu - |\mathbf{n}|^4 n_\nu v^\nu = 0, \tag{60}$$

since $|\mathbf{n}|^4 = 1$ whether $n^\mu$ is spacelike or timelike. Furthermore, the projector acts like a metric on vectors tangent to $\Sigma$:

$$P_{\mu\nu} v^\mu w^\nu = g_{\mu\nu} v^\mu w^\nu = \langle \mathbf{v}, \mathbf{w} \rangle, \tag{61}$$

since $n_\mu v^\mu = n_\mu w^\mu = 0$ for tangent vectors. Also, the projector remains unchanged under its own action:

$$P^\mu_\lambda P^\lambda_\nu = \left( \delta^\mu_\lambda - |\mathbf{n}|^2 n^\mu n_\lambda \right) \left( \delta^\lambda_\nu - |\mathbf{n}|^2 n^\lambda n_\nu \right) = \delta^\mu_\nu - 2|\mathbf{n}|^2 n^\mu n_\nu + |\mathbf{n}|^6 n^\mu n_\nu = P^\mu_\nu, \tag{62}$$

since $|\mathbf{n}|^6 = |\mathbf{n}|^2$.

### 2.5.5 Definition of Extrinsic Curvature

The *extrinsic curvature tensor*[27] $K_{\mu\nu}$ is a rank $(0,2)$ symmetric tensor defined as the Lie derivative[28] of the projector $P_{\mu\nu}$ in the direction of the normal vector $n^{\mu}$:

$$K_{\mu\nu} \equiv \frac{1}{2}\mathcal{L}_{\mathbf{n}}P_{\mu\nu}. \tag{64}$$

From the coordinate expression for the Lie derivative (see Footnote 28) it is relatively straightforward (try it!) to show that

$$K_{\mu\nu} = P_{\mu}^{\alpha}P_{\nu}^{\beta}\nabla_{(\alpha}n_{\beta)} = \frac{1}{2}P_{\mu}^{\alpha}P_{\nu}^{\beta}\mathcal{L}_{\mathbf{n}}g_{\alpha\beta}, \tag{65}$$

since by definition $\mathcal{L}_{\mathbf{n}}g_{\alpha\beta} = 2\nabla_{(\alpha}n_{\beta)}$.

Finally, let us assume that $n^{\mu}$ is tangent to a geodesic. This is generally possible by extending it off the surface $\Sigma$ using the geodesic equation, $n^{\lambda}\nabla_{\lambda}n^{\mu} = 0$. Then we have

$$\mathcal{L}_{\mathbf{n}}n_{\nu} = n^{\lambda}\nabla_{\lambda}n_{\nu} + n_{\lambda}\nabla_{\nu}n^{\lambda} = 0 + \frac{1}{2}\nabla_{\nu}\left(n_{\lambda}n^{\lambda}\right) = 0, \tag{66}$$

since $n_{\lambda}n^{\lambda}$ is a constant. Therefore we find that

$$K_{\mu\nu} = \frac{1}{2}\mathcal{L}_{\mathbf{n}}P_{\mu\nu} = \frac{1}{2}\mathcal{L}_{\mathbf{n}}\left(g_{\mu\nu} - |\mathbf{n}|^{2}n_{\mu}n_{\nu}\right) = \frac{1}{2}\mathcal{L}_{\mathbf{n}}g_{\mu\nu} = \nabla_{(\mu}n_{\nu)}. \tag{67}$$

## 2.6 Einstein's Equation

Let us define the *Einstein-Hilbert action*[29]:

$$S_{H} \equiv \frac{1}{16\pi}\int R\sqrt{-g}\,\mathrm{d}^{4}x, \tag{68}$$

where $g \equiv \det\mathbf{g}$. The factor[30] of $1/16\pi$ in front of the action is a convention chosen to produce the correct form of Newton's law of gravitation $F = m_{1}m_{2}/r^{2}$ in the Newtonian limit. By varying this action with respect to the metric, we obtain the *vacuum Einstein equation*:

$$\frac{1}{\sqrt{-g}}\frac{\delta S_{H}}{\delta g^{\mu\nu}} = \frac{1}{16\pi}\left(R_{\mu\nu} - \frac{1}{2}Rg_{\mu\nu}\right) = 0 \implies R_{\mu\nu} - \frac{1}{2}Rg_{\mu\nu} = 0. \tag{69}$$

To add matter, we simply add the appropriate action $S_{M}$ for the type of matter we would like to consider:

$$S \equiv S_{H} + S_{M}. \tag{70}$$

---

[27]Sometimes called the *second fundamental form* in the differential geometry literature.

[28]We will not go into the rigorous definition of the Lie derivative here, but just note that a general formula for its action on tensors is

$$\mathcal{L}_{\mathbf{n}}T^{\mu\cdots}{}_{\nu\cdots} = n^{\lambda}\partial_{\lambda}T^{\mu\cdots}{}_{\nu\cdots} - T^{\lambda\cdots}{}_{\nu\cdots}\partial_{\lambda}n^{\mu} - \ldots + T^{\mu\cdots}{}_{\lambda\cdots}\partial_{\nu}n^{\lambda} + \ldots, \tag{63}$$

where for each upper index of **T** we add a negative term with the index on **n** exchanged with that index, and for each lower index of **T** we add a positive term with the index on the partial derivative exchanged with that index. Note that the partial derivatives may be replaced with covariant derivatives, since the extra terms all cancel (show this).

[29]The *volume form* $\epsilon \equiv \frac{1}{4!}\epsilon_{\rho\sigma\mu\nu}\mathrm{d}x^{\rho}\wedge\mathrm{d}x^{\sigma}\wedge\mathrm{d}x^{\mu}\wedge\mathrm{d}x^{\nu} = \sqrt{-g}\,\mathrm{d}^{4}x$ is a 4-form also known as the *Levi-Civita tensor*, and its components $\epsilon_{\rho\sigma\mu\nu}$ are related to the familiar totally anti-symmetric *Levi-Civita symbol* $\tilde{\epsilon}_{\rho\sigma\mu\nu}$ by $\epsilon_{\rho\sigma\mu\nu} = \sqrt{-g}\,\tilde{\epsilon}_{\rho\sigma\mu\nu}$. Note that $\tilde{\epsilon}_{\rho\sigma\mu\nu}$ is not a tensor but a *tensor density* of weight 1, since it is related to a tensor by a factor of one power of $\sqrt{-g}$.

[30]If we did not use units where $G \equiv 1$, this would instead be $1/16\pi G$.

Whatever the form of $S_M$ is, by varying $S$ with respect to the metric we will get

$$\frac{1}{\sqrt{-g}}\frac{\delta S}{\delta g^{\mu\nu}} = \frac{1}{16\pi}\left(R_{\mu\nu} - \frac{1}{2}Rg_{\mu\nu}\right) + \frac{1}{\sqrt{-g}}\frac{\delta S_M}{\delta g^{\mu\nu}} = 0. \tag{71}$$

If we now define the *energy-momentum tensor* (or *stress-energy tensor*) to be the symmetric rank $(0,2)$ tensor with components

$$T_{\mu\nu} \equiv -\frac{2}{\sqrt{-g}}\frac{\delta S_M}{\delta g^{\mu\nu}}, \tag{72}$$

we get the *full Einstein equation* in the presence of matter:

$$R_{\mu\nu} - \frac{1}{2}Rg_{\mu\nu} = 8\pi T_{\mu\nu}. \tag{73}$$

The expression on the left-hand side is also known as the *Einstein tensor*[31]:

$$G_{\mu\nu} \equiv R_{\mu\nu} - \frac{1}{2}Rg_{\mu\nu}. \tag{74}$$

From the second Bianchi identity, one can show that $\nabla^\mu G_{\mu\nu} = 0$. This is good, since conservation of energy implies that the energy-momentum tensor must satisfy $\nabla^\mu T_{\mu\nu} = 0$.

One may also rewrite the Einstein equation as follows:

$$R_{\mu\nu} = 8\pi\left(T_{\mu\nu} - \frac{1}{2}Tg_{\mu\nu}\right), \tag{75}$$

where $T \equiv T^\mu{}_\mu$ is the trace of the energy-momentum tensor.

## 2.7 The Cosmological Constant

Consider the following action:

$$S_\Lambda \equiv -\frac{\Lambda}{8\pi}\int \sqrt{-g}\,\mathrm{d}^4x. \tag{76}$$

This is simply a constant times the volume of the entire spacetime! However, since the volume depends on the metric, the variation of this action with respect to the metric is not trivial; in fact, it is easy to see that

$$\frac{1}{\sqrt{-g}}\frac{\delta S_\Lambda}{\delta g^{\mu\nu}} = \frac{\Lambda}{16\pi}g_{\mu\nu}. \tag{77}$$

If we now take the full action to be

$$S \equiv S_H + S_\Lambda + S_M, \tag{78}$$

we get the *Einstein equation with a cosmological constant*:

$$R_{\mu\nu} - \frac{1}{2}Rg_{\mu\nu} + \Lambda g_{\mu\nu} = 8\pi T_{\mu\nu}. \tag{79}$$

---

[31]Notice from (69) and (72) that the Einstein tensor is, in fact, just the energy-momentum tensor of the metric field $g_{\mu\nu}$ itself, multiplied by $-8\pi$. Thus, if we treat the gravitational field as just another matter field, the Einstein equation reduces to the statement that the total energy-momentum tensor of all of the fields in the universe is zero: $T_{\mu\nu} = 0$!

The simplest way to interpret $\Lambda$ is as *vacuum energy*, that is, the energy density of empty spacetime. To see this, let us assume for simplicity there is no matter, so $T_{\mu\nu} = 0$:

$$R_{\mu\nu} - \frac{1}{2}Rg_{\mu\nu} + \Lambda g_{\mu\nu} = 0. \tag{80}$$

Then, we move the cosmological constant to the right-hand side of the equation:

$$R_{\mu\nu} - \frac{1}{2}Rg_{\mu\nu} = -\Lambda g_{\mu\nu}. \tag{81}$$

This may be interpreted as the usual Einstein equation with the energy-momentum tensor now given by

$$T_{\mu\nu} = -\frac{\Lambda}{8\pi}g_{\mu\nu}. \tag{82}$$

Now, let $t^\mu$ be a timelike vector normalized such that $|\mathbf{t}|^2 = g_{\mu\nu}t^\mu t^\nu = -1$. Then the energy density measured by an observer with 4-velocity $t^\mu$ is

$$T_{\mu\nu}t^\mu t^\nu = -\frac{\Lambda}{8\pi}g_{\mu\nu}t^\mu t^\nu = \frac{\Lambda}{8\pi}. \tag{83}$$

Hence, a cosmological constant corresponds to a uniform energy density everywhere in spacetime. Note that the sign of the energy density is the same as the sign of the cosmological constant; in our universe we have a positive cosmological constant, and thus positive energy density.

Finally, note that the trace of the energy-momentum tensor is

$$T \equiv T^\mu{}_\mu = -\frac{\Lambda}{2\pi}. \tag{84}$$

Thus

$$T_{\mu\nu} - \frac{1}{2}Tg_{\mu\nu} = \frac{\Lambda}{8\pi}g_{\mu\nu}, \tag{85}$$

and so

$$\left(T_{\mu\nu} - \frac{1}{2}Tg_{\mu\nu}\right)t^\mu t^\nu = -\frac{\Lambda}{8\pi}. \tag{86}$$

We will see the importance of this result in Sec. 3.3, when we will talk about violations of the energy conditions.

## 3 Advanced Topics in General Relativity

### 3.1 Causal Structure

Light cones are sufficient to understand causality in special relativity, where spacetime is flat. Whatever is inside the past light cone of a point can affect it, and whatever is in its future light cone can be affected by it. In a curved spacetime, things are not so simple anymore. Let us discuss some important concepts now.

#### 3.1.1 Chronological and Causal Relations

In general relativity, as we have discussed, it is assumed that massive particles follow timelike paths, and massless particles follows null paths. If we want to transmit information, we must use some kind of particle, either massive or massless. Therefore, in a sense, information itself must travel along either a timelike or null path. We thus define a *causal curve* to be a curve

whose tangent is non-spacelike everywhere. Particles following causal curves always travel locally at or below the speed of light.

Note that the fact that particles must follow causal curves does not mean they cannot exceed the the speed of light **globally**, and indeed in these lecture notes we will see some examples where this in fact happens! However, it is still true that massless and massive particles particles must **locally** travel at or below the speed of light respectively; the faster-than-light travel we will consider here is a global phenomenon.

At each point $p \in M$ we have a tangent space $T_p M$, which is by definition isomorphic to Minkowski spacetime. Thus, in the tangent space we may define a light cone passing through the origin, in the same way that we do in special relativity; it is the region bounded by null curves emanating from the origin. We may divide this light cone into a *future light cone* and a *past light cone*.

If there is a choice of past and future that is continuous throughout spacetime, we say that $M$ is *time orientable*. More precisely, a spacetime is time orientable if and only if it admits an everywhere non-vanishing continuous timelike vector field (which points e.g. to the future). We will assume that all spacetimes are time orientable from now on.

Let $p$ be a point in our spacetime manifold $M$. We define the *causal future* $J^+(p)$ to be the set of points that are connected to $p$ by a future-directed causal curve, and similarly define the *causal past* $J^-(p)$ to be the set of points that are connected to $p$ by a past-directed causal curve, where the future and past directions are determined by our particular choice of time orientation. The *chronological future* $I^+(p)$ and *chronological past* $I^-(p)$ are defined similarly, except that we replace causal curves by timelike curves[32]. Finally, if $S \subset M$ is a set of points in $M$, then $J^+(S)$ is defined to be the set of points that are connected by a future-directed causal curve to any point in $S$, and we similarly define $J^-(S)$ and $I^\pm(S)$.

Note that $I^+(p)$ is an open[33] set; $J^+(p)$ is often a closed set, but not always. Furthermore, $\overline{I^+(p)} = \overline{J^+(p)}$, where the bar denotes closure, and $\partial I^+(p) = \partial J^+(p)$. The proofs of these statements may be found in Hawking and Ellis [6], pages 182-183.

### 3.1.2 Achronal Sets and Cauchy Horizons

A set $S \subset M$ is called *achronal* if no two points in it are chronologically connected, that is, no two points can be connected by a timelike curve. Similarly, a set is *acausal* if no two points in it are causally connected. We can similarly define achronal and acausal surfaces. A *spacelike surface* is a surface such that all tangent vectors to the surface are spacelike.

Acausal sets and spacelike surfaces are completely independent notions. For example, the surface $t = x$ in flat Minkowski spacetime is achronal, but it is not spacelike, since its tangent vectors are null. Conversely, in 2+1-dimensional Minkowski spacetime with polar coordinates $(t, r, \phi)$, consider the surface $S$ defined by

$$\left\{ t = \frac{\phi}{2}, r \in (1, 3) \right\}. \tag{87}$$

This surface is a narrow strip (of width 2) winding around the $t$ axis. Two linearly independent tangent vectors to this 2-dimensional surface are $\partial_r$ (which points across the short side of the strip) and $\partial_t + 2\partial_\phi$ (which points up the long side of the strip), and they are both space-

---

[32] A useful mnemonic is that the extra curve in the bottom of the letter $J$ compared to $I$ corresponds to the extra type of curves (null ones) allowed in $J^\pm(p)$ compared to $I^\pm(p)$.

[33] An *open set* in a metric space is a set which contains a neighborhood of (that is, a ball around) each of its points. For example, $(0, 1)$ is an open set (in $\mathbb{R}$) since for each $x \in (0, 1)$ there is a neighborhood $(x - \varepsilon, x + \varepsilon) \subseteq (0, 1)$ for some $\varepsilon > 0$. A *closed set* is a set whose complement is an open set. For example, $[0, 1]$ is a closed set since its complement is $(-\infty, 0) \cup (1, +\infty)$, which is open.

like[34]. Thus, the surface is spacelike. However, the point $(t = 0, r = 2, \phi = 0)$ and the point $(t = \pi, r = 2, \phi = 0)$ may be connected by a timelike curve (with tangent vector $\partial_t$), so the surface is not achronal.

Now, consider a closed achronal set $S$. The *future domain of dependence* $D^+(S)$ of $S$ is the set of all points $p$ such that every inextendible[35], past-directed, causal curve through $p$ intersects $S$. In other words, every point in $D^+(S)$ is causally related to some point in $S$, so $D^+(S)$ is the region of spacetime which may be causally influenced by events happening in $S$. The requirement for the curve to be inextendible is necessary since we could always have short extendible curves which pass through $p$ and do not intersect $S$, but would intersect it if we extended them.

We similarly define the *past domain of dependence* $D^-(S)$ by replacing "future" with "past" in the definition above. The *domain of dependence* is then defined as $D(S) \equiv D^+(S) \cup D^-(S)$, and it is the set of all points in spacetime causally connected to $S$ from either the future or the past. Note that $S$ itself is inside both $D^+(S)$ and $D^-(S)$.

### 3.1.3 Cauchy Surfaces and Globally Hyperbolic Spacetimes

The *Cauchy horizon* $H(S)$ separates the points in $D(S)$, which are causally connected to $S$, from the points which are not causally connected to $S$. Thus, the Cauchy horizon is simply the boundary of $D(S)$, i.e. $H(S) = \partial D(S)$. In particular, we have the *future Cauchy horizon* $H^+(S) = \partial D^+(S)$ and the *past Cauchy horizon* $H^-(S) = \partial D^-(S)$, and $H(S) = H^+(S) \cup H^-(S)$. Both $H^+(S)$ and $H^-(S)$ are null surfaces.

Note that $D^+(S) \subseteq J^+(S)$, since by definition $J^+(S)$ is the set of points that are connected to $S$ by a causal curve. However, there can be points in $J^+(S)$ which are not in $D^+(S)$, since a point in $J^+(S)$ only needs to have **one** causal curve connected to $S$, while a point in $D^+(S)$ must have **every** causal curve connected to $S$. Similarly, $D^-(S) \subseteq J^-(S)$. Also, the closure[36] $\bar{D}^+(S)$ of $D^+(S)$ is[37] the set of all points $p$ such that every inextendible, past-directed, **timelike** curve through $p$ intersects $S$.

We are finally ready to define a *Cauchy surface*. It is a closed achronal[38] surface $S$ such that its domain of dependence $D(S)$ is the **entire spacetime** $M$. Thus, by knowing the information (e.g. the values of some field) on a Cauchy surface, we are able to predict (and postdict) what happens throughout all of spacetime. Unfortunately, not all spacetimes possess Cauchy surfaces; but if a spacetime has a Cauchy surface, we call it *globally hyperbolic*.

A simple example is any surface of constant $t$ in Minkowski spacetime. It is the entire universe, "frozen" in a single moment in time. If we know what is happening everywhere in the universe in that particular moment, then it is straightforward to use the equations of motion to know what will happen in any other moment.

### 3.1.4 Closed Causal Curves and Time Machines

When a causal (non-spacelike) curve forms a loop, we call it a *closed causal curve*. If spacetime possesses such a curve, and it passes through a Cauchy surface, then the information on the

---

[34]The 2+1D Minkowski metric in polar coordinates is $g_{\mu\nu} = \text{diag}(-1, 1, r^2)$. Thus $|\partial_r|^2 = 1$ and $\left|\partial_t + 2\partial_\phi\right|^2 = -1 + 4r^2 > 3$ since $r > 1$ inside the surface.

[35]An *inextendible causal curve* has no ends; it either goes on forever, or closes in on itself to create a *closed causal curve*. More precisely, a curve is inextendible if it is not a strict subset of another curve. For example, consider a curve $C : (0, 1] \to M$, which has an endpoint: $C(1)$. Obviously, we may extend this curve to $(0, 1 + \varepsilon)$ for some $\varepsilon > 0$, therefore it is not inextendible.

[36]i.e. the set $D^+(S)$ together with all of its limit points.

[37]The proof is a bit too technical for our purposes, so we will omit it.

[38]Note that some authors require a Cauchy surface to be spacelike, instead of achronal.

surface determines not only the future, but also – following the loop – the information on the surface itself. A *closed timelike curve* (or *closed chronological curve*) is defined similarly.

The *causality-violating region* of a spacetime is the set of all points $p$ which are connected to themselves by a closed causal curve. Each such point is in its own future (and past), and it is unclear how causality should work in this situation. We may define the causality-violating region associated with a point $p$ as:

$$J^0(p) \equiv J^+(p) \cap J^-(p), \tag{88}$$

and from this, the causality-violating region for the entire spacetime manifold $M$:

$$J^0(M) \equiv \bigcup_{p \in M} J^0(p). \tag{89}$$

If this region is empty, we say that $M$ is a *causal spacetime*. Of course, a *chronology-violating region* is defined similarly with $I$ instead of $J$. If this region is non-empty, we say that $M$ contains a *time machine*. We will discuss time machines in detail later in these notes.

## 3.2 The Energy Conditions

### 3.2.1 Introduction

We have seen that the Einstein equation takes the form $G_{\mu\nu} = 8\pi T_{\mu\nu}$ where the left-hand side depends on the metric and the right-hand side depends on the matter fields. Note that if the energy-momentum tensor is allowed to be arbitrary, then literally any metric trivially solves this equation! Simply calculate the Einstein tensor $G_{\mu\nu}$ for that metric, and then set $T_{\mu\nu} = G_{\mu\nu}/8\pi$.

Of course, we do know how to write down $T_{\mu\nu}$ for familiar forms of matter, such as a scalar field or the electromagnetic field. Using these known expressions, for specific distributions of the matter fields, definitely limits the metrics which can solve Einstein's equation, but now it limits them **too much**. For example, sometimes we want to derive general results or prove theorems which should be true in general, and not just for a particular choice of matter fields.

As a compromise, we often do not specify exactly what kind of matter we have, and only require that it satisfies some conditions that we consider "realistic" or "physically reasonable". These are called the *energy conditions*. Roughly speaking, these conditions restrict energy density to be non-negative and gravity to be attractive[39].

As we will see later, the strategy taken with the exotic metrics – which allow faster-than-light travel and/or time travel – is the one we described in the beginning; one first writes down the desired metric, and then calculates the form that $T_{\mu\nu}$ must take in order to solve the Einstein equation. In almost every single case, exotic metrics require a type of matter called *exotic matter*, which violates one or more of the energy conditions.

However, this may not necessarily be a problem, since the energy conditions are just assumptions; so far they have not been proven, in the most general case, to follow from fundamental principles[40]. In fact, some known realistic of matter – in particular, quantum fields –

---

[39]Not only as a field of study and research, but also as a force.

[40]The energy conditions cannot be proven from general relativity coupled to classical fields. However, in the context of quantum field theory (QFT), one may prove that some of the conditions must hold in specific cases. The averaged null energy condition (ANEC) – the weakest (and thus most important) of the conditions we will present in this chapter – has been proven in [9] to be satisfied by a Lorentz-invariant QFT on a **flat** spacetime. (Another proof is [10], but since it uses causality as an assumption for the proof, its results cannot be used to disallow causality violations.) However, quantum fields on a **curved** spacetime are known to violate even the ANEC. Nevertheless, as discussed in [11], there is an even weaker energy condition, known as the *achronal* ANEC, which is only required to hold on achronal null geodesics (see Subsection 3.1.2) and of which there currently exist

are known to violate some or all of the energy conditions to some extent. We shall see several examples below.

We will begin this section with some important definitions and derivations, and then proceed to motivate and state the most commonly used energy conditions.

### 3.2.2 Geodesic Deviation

Consider a *congruence* of timelike geodesics. This is a collection of geodesics in some (open) region of spacetime such that every point in that region is on precisely one geodesic. Note that this means the geodesics do not intersect each other.

More precisely, we consider a 1-parameter family of geodesics $x^\mu(\sigma, \tau)$ such that $\tau$ is the proper time along each geodesic and the parameter $\sigma$ continuously enumerates the different geodesics. This 1-parameter family thus defines a 2-dimensional surface with coordinates $\tau, \sigma$. In addition to the vector $t^\mu(\sigma, \tau) \equiv \partial x^\mu / \partial \tau$ tangent to each geodesic, we may also define a *separation vector*:

$$s^\mu \equiv \frac{\partial x^\mu}{\partial \sigma}. \tag{90}$$

The vector with components $V^\mu \equiv t^\nu \nabla_\nu s^\mu$, that is, the covariant derivative of $s^\mu$ in the direction of $t^\nu$, describes the *"relative velocity"* of the geodesics, that is, how the separation between them changes as one moves along the $\tau$ direction. We can furthermore define a *"relative acceleration"*:

$$A^\mu \equiv t^\mu \nabla_\mu V^\mu, \tag{91}$$

and one can derive (try it!) the following relation, called the *geodesic deviation equation*:

$$A^\mu = R^\mu{}_{\nu\rho\lambda} t^\nu t^\rho s^\lambda. \tag{92}$$

As expected, we see that the behavior of the separation between our geodesics is determined by the curvature. Furthermore, one can show that the "relative velocity" satisfies

$$V^\mu = t^\nu \nabla_\nu s^\mu = s^\nu \nabla_\nu t^\mu. \tag{93}$$

The equation for parallel transporting a vector $t^\mu$ along a curve with tangent vector $s^\nu$ is given by $s^\nu \nabla_\nu t^\mu = 0$. Therefore, the amount by which the relative velocity differs from zero measures how much the tangent vector $t^\mu$ fails to be parallel-transported in the direction of the neighboring geodesics, that is, along $s^\nu$.

### 3.2.3 The Raychaudhuri Equation

In the following we will assume that $t^\mu$ is a 4-velocity, so it is normalized as usual:

$$|\mathbf{t}|^2 = t^\mu t_\mu = -1. \tag{94}$$

Let us consider the rank $(0, 2)$ tensor $\nabla_\mu t_\nu$, the covariant derivative of the tangent vector to the geodesics. Note that, if we contract this tensor with $s^\mu$, we get $V^\nu$. We may decompose it into a trace, a symmetric traceless component, and an anti-symmetric component:

$$\nabla_\mu t_\nu = \frac{1}{3} \theta P_{\mu\nu} + \sigma_{\mu\nu} + \omega_{\mu\nu}, \tag{95}$$

where:

---

no known violations. So far the achronal ANEC has not actually been proven in the general case, but such a proof could be sufficient to completely rule out many (although not all) forms of causality violation. Since QFT is beyond the scope of these notes, we will not go into any more details here.

- $P_{\mu\nu} \equiv g_{\mu\nu} + t_\mu t_\nu$ is a *projector* into the subspace (of the tangent space) corresponding to vectors normal to $t^\mu$, defined similarly to (59). Note that $P^{\mu\nu}P_{\mu\nu} = 3$.

- $\theta \equiv \nabla_\mu t^\mu$, the trace, is called the *expansion*. It is the fractional change of volume per unit time, that is, $\theta = \Delta V/(V \Delta \tau)$ where $V$ is the volume. If $\theta > 0$ then the geodesics move away from each other and the congruence diverges; if $\theta < 0$ then the geodesics get closer together and the congruence converges. Alternatively, if we start with a sphere of particles, the expansion measures the change in the volume of the sphere.

- $\sigma_{\mu\nu} \equiv \nabla_{(\mu} t_{\nu)} - \frac{1}{3}\theta P_{\mu\nu}$, the symmetric traceless component (such that $P^{\mu\nu}\sigma_{\mu\nu} = 0$), is called the *shear*. If we start out with a sphere of particles, the shear measures the distortion in its shape into an ellipsoid.

- $\omega_{\mu\nu} \equiv \nabla_{[\mu} t_{\nu]}$, the anti-symmetric component, is called the *rotation* or *vorticity*. It measures how the sphere of particles rotates around itself. If $\omega_{\mu\nu} = 0$ then, by *Frobenius' theorem*, the congruence is orthogonal to a family of hypersurfaces foliating it.

Let us now look at the expression $t^\lambda \nabla_\lambda \nabla_\mu t_\nu$. Using the definition of the Riemann tensor as a commutator of covariant derivatives, (38), we find that

$$\nabla_\lambda \nabla_\mu t_\nu = \nabla_\mu \nabla_\lambda t_\nu + R_{\nu\sigma\lambda\mu} t^\sigma = \nabla_\mu \nabla_\lambda t_\nu - R_{\sigma\nu\lambda\mu} t^\sigma, \tag{96}$$

and thus

$$t^\lambda \nabla_\lambda \nabla_\mu t_\nu = t^\lambda \nabla_\mu \nabla_\lambda t_\nu - R_{\sigma\nu\lambda\mu} t^\sigma t^\lambda. \tag{97}$$

Taking the trace of this expression (with respect to the indices $\mu$, $\nu$), we get after relabeling indices

$$t^\lambda \nabla_\lambda \theta = t^\nu \nabla_\mu \nabla_\nu t^\mu - R_{\mu\nu} t^\mu t^\nu. \tag{98}$$

Next, we note that

$$t^\nu \nabla_\mu \nabla_\nu t^\mu = \nabla_\mu (t^\nu \nabla_\nu t^\mu) - (\nabla_\mu t_\nu)(\nabla^\nu t^\mu), \tag{99}$$

and furthermore, using (95) we have

$$(\nabla_\mu t_\nu)(\nabla^\nu t^\mu) = \frac{1}{3}\theta^2 + \sigma_{\mu\nu}\sigma^{\mu\nu} - \omega_{\mu\nu}\omega^{\mu\nu}. \tag{100}$$

Thus, we obtain

$$t^\lambda \nabla_\lambda \theta = \nabla_\mu (t^\nu \nabla_\nu t^\mu) + \omega_{\mu\nu}\omega^{\mu\nu} - \sigma_{\mu\nu}\sigma^{\mu\nu} - \frac{1}{3}\theta^2 - R_{\mu\nu} t^\mu t^\nu. \tag{101}$$

Finally, since $t^\mu$ is a tangent vector to a timelike geodesic, it satisfies the geodesic equation, $t^\nu \nabla_\nu t^\mu$, so the first term on the right-hand side vanishes. Moreover, $t^\lambda \nabla_\lambda = \mathrm{d}/\mathrm{d}\tau$ by the chain rule. Therefore, our equation becomes

$$\frac{\mathrm{d}\theta}{\mathrm{d}\tau} = \omega_{\mu\nu}\omega^{\mu\nu} - \sigma_{\mu\nu}\sigma^{\mu\nu} - \frac{1}{3}\theta^2 - R_{\mu\nu} t^\mu t^\nu. \tag{102}$$

This is called the *Raychaudhuri equation*. It has many uses, among them proving the Penrose-Hawking singularity theorems. Below we will see that it is used to impose the condition that gravity is always attractive. Note that this equation applies to a congruence of timelike geodesics; a similar equation may be derived for a congruence of null geodesics:

$$\frac{\mathrm{d}\theta}{\mathrm{d}\lambda} = \omega_{\mu\nu}\omega^{\mu\nu} - \sigma_{\mu\nu}\sigma^{\mu\nu} - \frac{1}{2}\theta^2 - R_{\mu\nu} n^\mu n^\nu, \tag{103}$$

where $\lambda$ is an affine parameter and $n^\mu$ is a null vector. The derivation of this equation is a bit more involved, and we will not do it here.

Finally, recall from (67) that the extrinsic curvature to a surface $\Sigma$ may be written as $K_{\mu\nu} = \nabla_{(\mu} t_{\nu)}$ where $t^\mu$ is a vector tangent to a geodesic passing through $\Sigma$ and normal to it. Thus, the expansion is none other than the trace of the extrinsic curvature:

$$\theta = K^\mu_\mu = \nabla_\mu t^\mu. \tag{104}$$

### 3.2.4 The Strong Energy Condition

Let us first find how to impose the condition that gravity should always be attractive. If this condition is satisfied, then a collection of (free-falling) massive particles should get closer to each other over time. Since the particles follow timelike geodesics, they may be collectively described as a congruence of such geodesics, and thus they satisfy the Raychaudhuri equation, (102).

If gravity is attractive, then we expect $d\theta/d\tau$ to be negative, signifying that the expansion $\theta$ decreases as time passes, that is, the geodesics get closer together[41]. Let us assume that the congruence has no vorticity, so $\omega_{\mu\nu} = 0$. Since the terms $\sigma_{\mu\nu}\sigma^{\mu\nu}$ and $\theta^2$ are always positive, the only term on the right-hand side of (102) which might prevent $d\theta/d\tau$ from being negative is the last term, $R_{\mu\nu}t^\mu t^\nu$. It is thus reasonable to demand

$$R_{\mu\nu}t^\mu t^\nu \geq 0, \qquad \text{for any timelike vector } t^\mu. \tag{105}$$

This is a sufficient, but not necessary, condition for gravity to always be attractive.

Now, using the Einstein equation in the form (75), we see that this condition is equivalent[42] to

The Strong Energy Condition (SEC):

$$\left(T_{\mu\nu} - \frac{1}{2}T g_{\mu\nu}\right) t^\mu t^\nu \geq 0, \qquad \text{for any timelike vector } t^\mu. \tag{106}$$

Imposing the SEC thus guarantees that gravity is attractive for massive particles. The statement that gravity is attractive if the SEC is satisfied is called the *focusing theorem*.

### 3.2.5 The Null, Weak and Dominant Energy Conditions

In the SEC, we can take timelike vectors which are arbitrarily close to being null. Since by definition $g_{\mu\nu}n^\mu n^\nu = 0$ for a null vector $n^\mu$, in the limit of null vectors $t^\mu \to n^\mu$ we obtain

The Null Energy Condition (NEC):

$$T_{\mu\nu}n^\mu n^\nu \geq 0, \qquad \text{for any null vector } n^\mu. \tag{107}$$

The NEC thus imposes that gravity is attractive also for massless particles – rays of light also converge under the influence of gravity. If it is violated, then the SEC will be violated as well.

Another reasonable assumption is that the energy density measured by any observer should be non-negative, which leads us to

The Weak Energy Condition (WEC):

$$T_{\mu\nu}t^\mu t^\nu \geq 0, \qquad \text{for any timelike vector } t^\mu. \tag{108}$$

---

[41]If initially $\theta > 0$, then $d\theta/d\tau < 0$ means that the congruence will diverge less rapidly until it eventually starts converging when $\theta < 0$.

[42]Note that if the Einstein equation is altered, as in some modified gravity theories, then this equivalence does not hold anymore.

Here, $t^\mu$ is the 4-velocity of the observer performing the measurement, and $T_{\mu\nu}t^\mu t^\nu$ is the energy density. Despite the naming, the WEC isn't actually a weaker form of the SEC; they are in fact independent. Note that the NEC may also be derived from the WEC in the limit of null vectors. Thus, if the NEC is violated, then so is the WEC.

One may furthermore impose that the energy density of massive particles propagates **in a causal way**. To do this, we define the *energy flux vector* $F^\mu \equiv -T^\mu{}_\nu t^\nu$, which encodes the energy and momentum density of the matter as measured by an observer moving at 4-velocity $t^\mu$. We require that it is causal, that is, $|\mathbf{F}|^2 \equiv F^\mu F_\mu \leq 0$. Adding this condition to the WEC, we get

The Dominant Energy Condition (DEC):

$$T_{\mu\nu}t^\mu t^\nu \geq 0 \text{ and } T^\mu{}_\nu t^\nu \text{ is not spacelike,} \qquad \text{for any timelike vector } t^\mu. \tag{109}$$

If the WEC is violated, then the DEC is violated as well. Note that this means if the NEC is violated, then so will all the other energy conditions we have defined!

### 3.2.6 Energy Density and Pressure

Let us now assume that the energy-momentum tensor $T_{\mu\nu}$ is of the *Hawking-Ellis Type I* form, which means there is some orthonormal basis[43] such that its components take the diagonal form

$$T^{\mu\nu} = \text{diag}(\rho, p_1, p_2, p_3), \tag{111}$$

where $\rho$ is the *energy density* and $p_i$ are the three *principle pressures*. An example of a type of matter which has an energy-momentum tensor of this form is a *perfect fluid*[44], which has

$$T^{\mu\nu} = (\rho + p)u^\mu u^\nu + p g^{\mu\nu}, \tag{112}$$

where $u^\mu$ is the 4-velocity of the fluid.

In the orthonormal frame where the energy-momentum tensor is of the form (111), we find that the four energy conditions we have considered above imply the following:

- Null Energy Condition: $\rho + p_i \geq 0$ for all $i$.

- Weak Energy Condition: $\rho + p_i \geq 0$ for all $i$ and $\rho \geq 0$.

- Strong Energy Condition: $\rho + p_i \geq 0$ for all $i$ and $\rho + \sum_i p_i \geq 0$.

- Dominant Energy Condition: $|p_i| \leq \rho$ for all $i$ and $\rho \geq 0$.

---

[43]An orthonormal basis is a set of four 4-vectors $\mathbf{e}_A$, referred to as *frame fields*, *vierbeins* or *tetrads*, with components $e_A^\mu$ where $\mu$ is the usual spacetime index and $A \in \{0, 1, 2, 3\}$ is called an *internal index*, such that $\langle \mathbf{e}_A, \mathbf{e}_B \rangle \equiv g_{\mu\nu} e_A^\mu e_B^\nu = \eta_{AB}$. The energy-momentum tensor is then decomposed as

$$T^{\mu\nu} = \rho e_0^\mu e_0^\nu + \sum_{i=1}^3 p_i e_i^\mu e_i^\nu. \tag{110}$$

Then $\rho$ and $p_i$ are the eigenvalues of $T_{\mu\nu}$, while $\mathbf{e}_A$ are its eigenvectors. The spacetime components $v^\mu$ of an abstract vector $\mathbf{v}$ are related to its components $v^A$ in the orthonormal basis by $v^\mu = v^A e_A^\mu$.

[44]This is a special case of (110) where we identify $\mathbf{e}_0$ with the 4-velocity $\mathbf{u}$ and take $p_i = p$. Using the fact that $g^{\mu\nu} = \eta^{AB} e_A^\mu e_B^\nu = -u^\mu u^\nu + \sum_{i=1}^3 e_i^\mu e_i^\nu$, we can then write $\sum_{i=1}^3 p_i e_i^\mu e_i^\nu = p(u^\mu u^\nu + g^{\mu\nu})$.

### 3.2.7 The Averaged Energy Conditions

If we integrate the Raychaudhuri equation (102) along a timelike geodesic $\Gamma$ with tangent vector $t^\mu$, we get

$$\Delta\theta = \int_\Gamma \left( \omega_{\mu\nu}\omega^{\mu\nu} - \sigma_{\mu\nu}\sigma^{\mu\nu} - \frac{1}{3}\theta^2 - R_{\mu\nu}t^\mu t^\nu \right) d\tau, \tag{113}$$

where $\Delta\theta$ is the difference in the expansion between the initial and final points of the geodesic. If we assume $\omega_{\mu\nu} = 0$ as before, we get that

$$\Delta\theta \leq -\int_\Gamma R_{\mu\nu}t^\mu t^\nu d\tau. \tag{114}$$

Therefore, for the geodesics to converge (i.e. for $\theta$ to decrease) it is in fact enough to require that

$$\int_\Gamma R_{\mu\nu}t^\mu t^\nu d\tau \geq 0, \qquad \text{for any timelike geodesic } \Gamma \text{ with tangent } t^\mu. \tag{115}$$

As before, using the Einstein equation, we obtain

$$\text{The Averaged Strong Energy Condition (ASEC):} \tag{116}$$

$$\int_\Gamma \left( T_{\mu\nu} - \frac{1}{2}T g_{\mu\nu} \right) t^\mu t^\nu d\tau \geq 0, \qquad \text{for any timelike geodesic } \Gamma \text{ with tangent } t^\mu. \tag{117}$$

In the null limit we have

$$\text{The Averaged Null Energy Condition (ANEC):} \tag{118}$$

$$\int_\Gamma T_{\mu\nu}n^\mu n^\nu d\lambda \geq 0, \qquad \text{for any null geodesic } \Gamma \text{ with tangent } n^\mu. \tag{119}$$

Here $\lambda$ is an affine parameter[45] along the null geodesic. The ANEC is the weakest energy condition we have – but unfortunately, even this condition is violated in some cases.

Finally, we also define

$$\text{The Averaged Weak Energy Condition (AWEC):} \tag{120}$$

$$\int_\Gamma T_{\mu\nu}t^\mu t^\nu d\tau \geq 0, \qquad \text{for any timelike geodesic } \Gamma \text{ with tangent } t^\mu. \tag{121}$$

The idea behind the averaged energy conditions is that the (non-averaged) energy conditions are allowed to be violated in some regions along the geodesics, as long as they are balanced by positive contributions from other regions; they are thus satisfied "on average".

---

[45]If we allow the parameter $\lambda$ to be arbitrary, the ANEC will be equivalent to the (non-averaged) NEC.

### 3.3 Violations of the Energy Conditions

Despite the fact that the energy conditions are supposed to "reasonable", some types of matter are known to violate some or all of these conditions. These include both classical and quantum matter. A comprehensive list (with references) may be found in [12]. We will not discuss examples from quantum field theory here, since they are beyond the scope of these notes; instead, we will focus on classical examples.

A particularly simple example is the cosmological constant; in (86) we saw that

$$\left(T_{\mu\nu} - \frac{1}{2}T g_{\mu\nu}\right)t^{\mu}t^{\nu} = -\frac{\Lambda}{8\pi}. \tag{122}$$

Thus, a positive[46] cosmological constant (as is the case in our universe) violates the SEC. Since the SEC was derived by requiring gravity to be attractive, a positive cosmological constant may potentially make gravity repulsive instead. This makes sense, given that the cosmological constant accelerates the expansion of the universe, so in a way it makes galaxies "repel" each other .

#### 3.3.1 Basic Facts about Scalar Fields

We now move to a more complicated example, that of a classical scalar field. The discussion is based largely on [13].

The action for a classical scalar field $\phi$ coupled to gravity is

$$S = -\int d^4x \sqrt{-g}\left(\frac{1}{2}\left(g^{\mu\nu}\nabla_\mu\phi\nabla_\nu\phi + \xi R\phi^2\right) + V(\phi)\right), \tag{123}$$

where $R$ is the Ricci scalar, $V(\phi)$ is the field's potential, and $\xi$ is a coupling constant. If $\xi = 0$, the field is referred to as *minimally coupled* to gravity. Scalar field theories describe, for example, the Higgs field, scalar mesons (pions, etc.), and some hypothesized fields such as axions and inflatons.

The equation of motion may be found by varying the action with respect to $\phi$:

$$\delta S = -\int d^4x \sqrt{-g}\left(g^{\mu\nu}\nabla_\mu\phi\nabla_\nu\delta\phi + \xi R\phi\,\delta\phi + \frac{dV}{d\phi}\delta\phi\right). \tag{124}$$

To isolate $\delta\phi$, we need to integrate the first term by parts. This is easy because the connection is metric-compatible, $\nabla_\lambda g_{\mu\nu} = 0$, so

$$\sqrt{-g}\,g^{\mu\nu}\nabla_\mu\phi\nabla_\nu\delta\phi = -\sqrt{-g}\left(g^{\mu\nu}\nabla_\nu\nabla_\mu\phi\right)\delta\phi, \tag{125}$$

where we ignored boundary terms since we assume the variation vanishes at infinity. It is customary to define the *covariant d'Alembertian* or *"box" operator*:

$$\Box \equiv g^{\mu\nu}\nabla_\nu\nabla_\mu. \tag{126}$$

Then we get

$$\delta S = \int d^4x \sqrt{-g}\left(\Box\phi - \xi R\phi - \frac{dV}{d\phi}\right)\delta\phi, \tag{127}$$

---

[46]In (83) we found that $T_{\mu\nu}t^{\mu}t^{\nu} = \Lambda/8\pi$ and thus a negative cosmological constant would violate the WEC (and, trivially, also the AWEC) since it corresponds to a negative energy density everywhere in spacetime. However, our universe happens to have a positive $\Lambda$, and therefore we already know that the cosmological constant in our universe in fact does satisfy the WEC.

so the equation of motion is

$$\Box\phi - \xi R\phi - \frac{dV}{d\phi} = 0. \tag{128}$$

The corresponding energy-momentum tensor may be calculated using (72). For this we must vary the action with respect to the inverse metric $g^{\mu\nu}$. For this we will use the relation[47]

$$\delta\sqrt{-g} = -\frac{1}{2}\sqrt{-g}\,g_{\mu\nu}\delta g^{\mu\nu}. \tag{133}$$

We will also need the variation of $R$, which turns out to be (prove this!)

$$\delta R = R_{\mu\nu}\delta g^{\mu\nu} + g_{\mu\nu}\Box\delta g^{\mu\nu} - \nabla_\mu\nabla_\nu\delta g^{\mu\nu}. \tag{134}$$

Taking the variation of the action with respect to $g^{\mu\nu}$, we get after some manipulations, using (72):

$$T_{\mu\nu} = \nabla_\mu\phi\nabla_\nu\phi - g_{\mu\nu}\left(\frac{1}{2}g^{\rho\sigma}\nabla_\rho\phi\nabla_\sigma\phi + V(\phi)\right) + \xi\left(g_{\mu\nu}\Box + G_{\mu\nu} - \nabla_\mu\nabla_\nu\right)\phi^2, \tag{135}$$

where $G_{\mu\nu} \equiv R_{\mu\nu} - \frac{1}{2}Rg_{\mu\nu}$ is the Einstein tensor (74). Now, let us plug this into the Einstein equation (73):

$$G_{\mu\nu} = 8\pi\left(\nabla_\mu\phi\nabla_\nu\phi - g_{\mu\nu}\left(\frac{1}{2}g^{\rho\sigma}\nabla_\rho\phi\nabla_\sigma\phi + V(\phi)\right) + \xi\left(g_{\mu\nu}\Box + G_{\mu\nu} - \nabla_\mu\nabla_\nu\right)\phi^2\right). \tag{136}$$

Note that the Einstein tensor $G_{\mu\nu}$ appears on both sides. If we isolate it, we get

$$G_{\mu\nu} = \frac{8\pi}{1-8\pi\xi\phi^2}\left(\nabla_\mu\phi\nabla_\nu\phi - g_{\mu\nu}\left(\frac{1}{2}g^{\rho\sigma}\nabla_\rho\phi\nabla_\sigma\phi + V(\phi)\right) + \xi\left(g_{\mu\nu}\Box\phi^2 - \nabla_\mu\nabla_\nu\phi^2\right)\right). \tag{137}$$

Therefore we may define a simpler *effective energy-momentum tensor*:

$$\tilde{T}_{\mu\nu} = \frac{1}{1-8\pi\xi\phi^2}\left(\nabla_\mu\phi\nabla_\nu\phi - g_{\mu\nu}\left(\frac{1}{2}g^{\rho\sigma}\nabla_\rho\phi\nabla_\sigma\phi + V(\phi)\right) + \xi\left(g_{\mu\nu}\Box\phi^2 - \nabla_\mu\nabla_\nu\phi^2\right)\right). \tag{138}$$

This tensor then appears on the right-hand side of the Einstein equation, $R_{\mu\nu} - \frac{1}{2}Rg_{\mu\nu} = 8\pi\tilde{T}_{\mu\nu}$, such that now the right-hand side does **not** depend directly on the Ricci tensor $R_{\mu\nu}$. Therefore it is the energy-momentum tensor of interest with respect to the energy conditions. Indeed, recall that we derived the SEC by first demanding that gravity attracts, thus finding a condition on $R_{\mu\nu}$, and then used Einstein's equation to convert it into a condition on $T_{\mu\nu}$. Hence, in that derivation we must use $\tilde{T}_{\mu\nu}$ and not the original $T_{\mu\nu}$. From now on we will just write $T_{\mu\nu}$ for the effective energy-momentum tensor, for brevity.

---

[47]To derive this, recall the familiar relation $\ln(\det\mathbf{M}) = \mathrm{tr}(\ln\mathbf{M})$, where $\mathbf{M}$ is any matrix. Taking the variation, we find

$$\frac{\delta(\det\mathbf{M})}{\det\mathbf{M}} = \mathrm{tr}(\delta\mathbf{M}\mathbf{M}^{-1}). \tag{129}$$

Note that the order of $\delta\mathbf{M}$ and $\mathbf{M}^{-1}$ inside the trace does not matter, since it is cyclic. Applying this relation to the metric $\mathbf{g}$, we find

$$\frac{\delta g}{g} = \mathrm{tr}(\delta\mathbf{g}\mathbf{g}^{-1}) = \delta g_{\mu\nu}g^{\mu\nu}. \tag{130}$$

Now, since $g_{\mu\nu}g^{\mu\nu} = 4$, we have

$$\delta(g_{\mu\nu}g^{\mu\nu}) = \delta g_{\mu\nu}g^{\mu\nu} + g_{\mu\nu}\delta g^{\mu\nu} = 0, \tag{131}$$

so $\delta g_{\mu\nu}g^{\mu\nu} = -g_{\mu\nu}\delta g^{\mu\nu}$, and we can write $\delta g = -g\,g_{\mu\nu}\delta g^{\mu\nu}$. Finally, we have

$$\delta\sqrt{-g} = -\frac{\delta g}{2\sqrt{-g}} = -\frac{(-g)g_{\mu\nu}\delta g^{\mu\nu}}{2\sqrt{-g}} = -\frac{1}{2}\sqrt{-g}\,g_{\mu\nu}\delta g^{\mu\nu}. \tag{132}$$

### 3.3.2 The Scalar Field and the Energy Conditions

We are now ready to test whether the scalar field satisfies the energy conditions. Already from the overall coefficient $1/\left(1-8\pi\xi\phi^2\right)$ in (138) we can see that something fishy is going on, since for $\xi \neq 0$ the sign of $\tilde{T}_{\mu\nu}$ depends on $\phi$!

Let us begin by testing the SEC:

$$\left(T_{\mu\nu} - \frac{1}{2}Tg_{\mu\nu}\right)t^\mu t^\nu \geq 0, \tag{139}$$

where $t^\mu$ is a timelike vector. In fact, this energy condition is already violated for a minimally-coupled scalar field, that is, with $\xi = 0$, where the energy-momentum tensor (138) simplifies to

$$T_{\mu\nu} = \nabla_\mu\phi\nabla_\nu\phi - g_{\mu\nu}\left(\frac{1}{2}g^{\rho\sigma}\nabla_\rho\phi\nabla_\sigma\phi + V(\phi)\right), \tag{140}$$

and its trace is

$$T = g^{\mu\nu}T_{\mu\nu} = -g^{\rho\sigma}\nabla_\rho\phi\nabla_\sigma\phi - 4V(\phi). \tag{141}$$

For a timelike vector $t^\mu$ normalized as usual such that $|\mathbf{t}|^2 = -1$, we get

$$\left(T_{\mu\nu} - \frac{1}{2}Tg_{\mu\nu}\right)t^\mu t^\nu = \left(t^\mu\nabla_\mu\phi\right)^2 - V(\phi). \tag{142}$$

If $t^\mu\nabla_\mu\phi$ is small (the field changes slowly in the direction of $t^\mu$) and the potential is positive (e.g. $V(\phi) = \frac{1}{2}m^2\phi^2$ for a free massive field), we see that the SEC is easily violated!

Next we move to the NEC:

$$T_{\mu\nu}n^\mu n^\nu \geq 0, \tag{143}$$

where $n^\mu$ is a null vector. Since $g_{\mu\nu}n^\mu n^\nu = 0$, we get

$$T_{\mu\nu}n^\mu n^\nu = \frac{1}{1-8\pi\xi\phi^2}\left(\left(n^\mu\nabla_\mu\phi\right)^2 - \xi n^\mu n^\nu\nabla_\mu\nabla_\nu\phi^2\right). \tag{144}$$

For minimal coupling, $\xi = 0$, this simplifies to

$$T_{\mu\nu}n^\mu n^\nu = \left(n^\mu\nabla_\mu\phi\right)^2, \tag{145}$$

which is obviously non-negative. So a minimally-coupled scalar field satisfies the NEC. However, for $\xi \neq 0$, let us assume that $n^\mu$ is tangent to a null geodesic, so that $n^\mu\nabla_\mu n^\nu = 0$. If $\lambda$ is an affine parameter, such that $n^\mu\nabla_\mu = \mathrm{d}/\mathrm{d}\lambda$, then we may write (144) as

$$T_{\mu\nu}n^\mu n^\nu = \frac{1}{1-8\pi\xi\phi^2}\left(\left(\frac{\mathrm{d}\phi}{\mathrm{d}\lambda}\right)^2 - \xi\frac{\mathrm{d}^2\left(\phi^2\right)}{\mathrm{d}\lambda^2}\right). \tag{146}$$

At a local extremum of $\phi^2$ along the null geodesic we have

$$\frac{\mathrm{d}\left(\phi^2\right)}{\mathrm{d}\lambda} = 2\phi\frac{\mathrm{d}\phi}{\mathrm{d}\lambda} = 0, \tag{147}$$

and thus as long as $\phi \neq 0$, we have $\mathrm{d}\phi/\mathrm{d}\lambda = 0$ there. Hence

$$T_{\mu\nu}n^\mu n^\nu = \frac{1}{8\pi\phi^2 - 1/\xi}\frac{\mathrm{d}^2\left(\phi^2\right)}{\mathrm{d}\lambda^2}. \tag{148}$$

Now, if $\xi < 0$, or $\xi > 0$ and $\phi^2 > 1/8\pi\xi$ at the extremum, then the denominator will be positive and thus any local maximum of $\phi^2$, which has $\mathrm{d}^2\left(\phi^2\right)/\mathrm{d}\lambda^2 < 1$, will violate the NEC.

Similarly, if $\xi > 0$ and $\phi^2 < 1/8\pi\xi$ then the denominator will be negative and thus any local minimum of $\phi^2$ will violate the NEC. Recall that the NEC is the weakest energy condition, in the sense that if the NEC is violated, then so will all the other energy conditions. Therefore, a classical scalar field which is not minimally coupled violates all of the energy conditions!

In fact, we can go even weaker and show the the ANEC is violated, which means all of the averaged energy conditions are violated. We have

$$\int T_{\mu\nu}n^\mu n^\nu \mathrm{d}\lambda = \int \frac{1}{1-8\pi\xi\phi^2}\left(\left(\frac{\mathrm{d}\phi}{\mathrm{d}\lambda}\right)^2 - \xi\frac{\mathrm{d}^2\left(\phi^2\right)}{\mathrm{d}\lambda^2}\right)\mathrm{d}\lambda. \tag{149}$$

Integrating the second term by parts, discarding the boundary term, and collecting terms, we get

$$\int T_{\mu\nu}n^\mu n^\nu \mathrm{d}\lambda = \int \left(\frac{1-8\pi\xi\left(1-4\xi\right)\phi^2}{\left(1-8\pi\xi\phi^2\right)^2}\right)\left(\frac{\mathrm{d}\phi}{\mathrm{d}\lambda}\right)^2\mathrm{d}\lambda. \tag{150}$$

Thus, the ANEC is violated when

$$\xi\left(1-4\xi\right) > \frac{1}{8\pi\phi^2}. \tag{151}$$

This is only possible for $\xi \in (0, 1/4)$, since otherwise the left-hand side will be negative. In fact, the value $\xi = 1/6$, which is within this range, corresponds to *conformal coupling* which is thought to be the most natural way in which quantum scalar fields should be coupled to gravity. In terms of $\phi$ we may write this as

$$\phi^2 > \frac{1}{8\pi\xi\left(1-4\xi\right)} > \frac{2}{\pi}, \tag{152}$$

where in the last step we used the fact that $\xi(1-4\xi)$ has a local maximum at $\xi = 1/8$, where $\xi(1-4\xi) = 1/16$. Since we are working in Planck units, this means that $\phi$ must take a value at the Planck scale in order for the ANEC to be violated.

### 3.3.3 The Jordan and Einstein Frames

Let us combine the scalar field action (123) with the Einstein-Hilbert action (68):

$$S = -\int \mathrm{d}^4x \sqrt{-g}\left(\frac{1}{2}\left(g^{\mu\nu}\nabla_\mu\phi\nabla_\nu\phi + \left(\xi\phi^2 - \frac{1}{8\pi}\right)R\right) + V\left(\phi\right)\right). \tag{153}$$

If our theory is described by such an action, where there is a term in which the Ricci scalar $R$ is multiplied by the scalar field $\phi$, we say that it is in the *Jordan frame*. If we take a conformal transformation [14]

$$g_{\mu\nu} \mapsto \Omega^2 g_{\mu\nu}, \qquad \Omega^2 = 1-8\pi\xi\phi^2, \tag{154}$$

and redefine the scalar field

$$\mathrm{d}\phi \mapsto \frac{\sqrt{1-8\pi\xi\left(1-6\xi\right)\phi^2}}{1-8\pi\xi\phi^2}\mathrm{d}\phi, \tag{155}$$

we obtain gravity with a **minimally-coupled** scalar field:

$$S = -\int \mathrm{d}^4x \sqrt{-g}\left(\frac{1}{2}\left(g^{\mu\nu}\nabla_\mu\phi\nabla_\nu\phi - \frac{1}{8\pi}R\right) + V\left(\phi\right)\right). \tag{156}$$

We call this the *Einstein frame*. This is still the same action as before, only written in different variables. Does this mean that a minimally-coupled scalar field may also violate the ANEC, as

we have shown above for the action (123) with $\xi \in (0, 1/4)$? In fact, that is not the case. Note that this transformation is only possible if

$$\Omega^2 = 1 - 8\pi\xi\phi^2 > 0, \qquad 1 - 8\pi\xi(1 - 6\xi)\phi^2 > 0. \tag{157}$$

Let us only consider conformal coupling, $\xi = 1/6$, for simplicity. Then the second condition is always satisfied, and the first condition reduces to

$$\phi^2 < \frac{3}{4\pi}. \tag{158}$$

However, in order to violate the ANEC, (152) must also be satisfied. For $\xi = 1/6$, (152) becomes

$$\phi^2 > \frac{9}{4\pi}. \tag{159}$$

Clearly, we cannot have both (158) and (159) at the same time. Thus, we cannot use this trick to make a minimally-coupled scalar field violate the ANEC.

### 3.3.4 Conclusions

To summarize, we have found that a positive cosmological constant and a minimally-coupled scalar field violate the SEC, while a conformally-coupled scalar field violates the NEC and thus all the other energy conditions, and can even violate the ANEC, which is the weakest energy condition we have defined.

As we commented in the beginning, many other examples exist, some classical and some quantum. However, the few simple examples we have considered here should already make it clear that the energy conditions are definitely not fundamental axioms or results of general relativity, but rather additional assumptions that do not necessarily apply to all conceivable forms of matter.

As we will see below, it is possible to define spacetime geometries in general relativity which potentially allow for faster-than-light travel and/or time travel. These spacetimes almost universally require, through the Einstein equation, matter which violates the energy conditions, also known as *exotic matter*. It is currently unknown whether exotic matter may realistically be used to facilitate these exotic spacetime geometries. We will discuss this in more detail below.

## 4 Faster-than-Light Travel and Time Travel in Special Relativity

### 4.1 The Nature of Velocity

In this section we will work in special relativity, in which spacetime is flat and described by the Minkowski metric $\eta_{\mu\nu} \equiv \mathrm{diag}(-1, 1, 1, 1)$. These results also work in general relativity, but only locally, if one uses locally inertial coordinates as described in Sec. 2.3.4. Indeed, one of the major differences between special and general relativity is that in the former the inertial coordinates are global, while in the latter one can generally only define them locally, that is, at one particular point in spacetime.

### 4.1.1 Movement in the Time Direction

Recall that massive particles move along timelike paths. Such paths, when parametrized as usual by the proper time $\tau$, have tangent vectors $\dot{x}^\mu$ with $|\dot{x}|^2 = -1$. The tangent vector is the 4-velocity of the particle. If the particle is at rest, then it has

$$\dot{x}^\mu = (1, 0, 0, 0). \tag{160}$$

In other words, it has no velocity along any of the spatial coordinates, but it "moves at the speed of light[48]" along the time coordinate. Indeed, a massive particle must **always** move in time, since if we take $\dot{t} = 0$ then there is no way to satisfy the condition

$$|\dot{\mathbf{x}}|^2 = -\dot{t}^2 + \dot{x}^2 + \dot{y}^2 + \dot{z}^2 < 0. \tag{161}$$

Thus, we learn that the 4-velocity for a massive particle must have a non-zero value in the time component.

Note that we said "non-zero", not "positive"; even if $\dot{t} < 0$, we still have $\dot{t}^2 > 0$. In other words, $\dot{x}^\mu = (-1, 0, 0, 0)$ is also a perfectly good 4-velocity, except that it is past-directed instead of future-directed. Unfortunately, time travel is not as easy as taking $\dot{t} < 0$; in practice we always assume[49] that the 4-velocity is future-directed, $\dot{t} > 0$.

### 4.1.2  The Lorentz Factor

Let us now imagine a particle moving at some constant spatial 3-velocity $v$ along, say, the $x^1$ direction:

$$v \equiv \frac{\mathrm{d}x^1}{\mathrm{d}t} = \frac{\mathrm{d}x^1}{\mathrm{d}x^0}. \tag{162}$$

Since the 4-velocity is defined as the derivative with respect to the proper time $\tau$, **not** the coordinate time $t = x^0$, the corresponding spatial component $\dot{x}^1$ of the 4-velocity will be

$$\dot{x}^1 \equiv \frac{\mathrm{d}x^1}{\mathrm{d}\tau} = \frac{\mathrm{d}t}{\mathrm{d}\tau}\frac{\mathrm{d}x^1}{\mathrm{d}t} = \gamma v, \tag{163}$$

where $\gamma \equiv \mathrm{d}t/\mathrm{d}\tau$ is called the *Lorentz factor*, and it measures the relation between coordinate time and proper time. In other words, $\gamma$ measures the amount of *time dilation*, since $\mathrm{d}t = \gamma\,\mathrm{d}\tau$.

Similarly, the time component $\dot{x}^0 = \dot{t}$ of the 4-velocity will be

$$\dot{t} \equiv \frac{\mathrm{d}t}{\mathrm{d}\tau} = \gamma. \tag{164}$$

In conclusion, the 4-velocity is

$$\dot{x}^\mu = \gamma\,(1, v, 0, 0). \tag{165}$$

To derive an explicit value for $\gamma$, we use the normalization condition:

$$|\dot{\mathbf{x}}|^2 = \gamma^2\left(-1 + v^2\right) = -1 \quad \Longrightarrow \quad \gamma = \frac{1}{\sqrt{1 - v^2}}. \tag{166}$$

We now see that there is nothing mysterious about the Lorentz factor in special relativity; it is just a normalization factor!

In (37) we showed that parallel transport preserves the norm of the tangent vector to the path. Thus, a particle which starts moving along a timelike path will forever stay along a timelike path, and so a massive particle can never reach or exceed the speed of light (locally), since that would mean changing to a null path. Another way to see this is to note that $\gamma \to \infty$ as $v \to 1$. However, the energy of a massive particle with mass $m$ moving at speed $v$ is $E = \gamma m$. Therefore, it would require infinite energy to accelerate the particle to the speed of light.

---

[48]Recall that we are using units where $c \equiv 1$.

[49]Another way to phrase this assumption is that the *"arrow of time"* points to the future. This result **cannot** be derived from relativity; in fact, relativity is completely invariant under *time reversal*, $t \mapsto -t$. However, we nonetheless know that physics is in general not time-reversal-invariant; entropy increases when going forward in time. Understanding this discrepancy is a major open problem in physics.

### 4.1.3 The Speed of Light

In Sec. 2.3.4 we proved that particles moving along null paths always locally move at the speed of light. Let us now show the opposite – that a particle moving locally at the speed of light always moves along a null path. Notice that in the discussion of the previous section we did not actually use the fact that the path is timelike until (166). If $v = 1$, then (166) becomes

$$|\dot{\boldsymbol{x}}|^2 = \gamma^2\left(-1 + v^2\right) = 0, \tag{167}$$

and thus the path must be null. In that case $\gamma$ can take any non-zero value, and it is conventional to set it to 1, so the 4-velocity will be

$$\dot{x}^\mu = (1, 1, 0, 0). \tag{168}$$

From (37), we see that a particle which starts on the null path must stay on a null path. Thus, a particle moving at the speed of light can never decelerate or accelerate to a different speed.

Since the norm $|\dot{\boldsymbol{x}}|^2$ is by definition invariant under a Lorentz transformation, the path will remain null when transforming to any other frame of reference. We deduce that the speed of light is the same for all observers, which is of course one of the two fundamental postulates of relativity.

### 4.1.4 Tachyons

If $v > 1$, that is, the particle is moving faster than the speed of light, then the norm of the 4-velocity (165) will necessarily be $|\dot{\boldsymbol{x}}|^2 > 0$ and thus the path must be spacelike. In that case we can normalize to $|\dot{\boldsymbol{x}}|^2 = 1$, which gives[50]

$$\gamma = \frac{1}{\sqrt{v^2 - 1}}. \tag{169}$$

Particles moving faster than light, and along spacelike paths, are called *tachyons*. Just as normal massive particles moving along timelike paths can never accelerate to the speed of light or beyond, so can tachyons never decelerate to the speed of light or below. This follows from the fact that a particle which starts on a spacelike path will stay on a spacelike path.

Another way to see this is to note that, just for normal particles, we have $\gamma \to \infty$ as $v \to 1$ (except now $v$ approaches 1 from above). Therefore, it would require infinite energy $E = \gamma m$ to decelerate a tachyon to the speed of light. Paradoxically, a tachyon in fact has **less** energy at higher velocities, since $\gamma \to 0$ as $v \to \infty$! Indeed, a tachyon is at rest when its velocity is **infinite**, $v \to \infty$, since that's when its energy is minimized (and *proper distance* $\mathrm{d}s^2 = g_{\mu\nu}\,\mathrm{d}x^\mu \otimes \mathrm{d}x^\nu$ is maximized). Taking $\gamma = 1/\sqrt{v^2 - 1}$, we see that $\lim_{v\to\infty} \gamma = 0$, but $\lim_{v\to\infty} \gamma v = 1$. Therefore a tachyon at rest has

$$\dot{x}^\mu = \lim_{v\to\infty} \gamma (1, v, 0, 0) = (0, 1, 0, 0), \tag{170}$$

so it is moving only along a spacelike direction and not along the timelike direction, just as a normal massive particle at rest moves only along a timelike direction and not along a spacelike direction.

---

[50]Note that the Lorentz factor is real, as it should be. Naively, if one used the definition of $\gamma$ which was derived by assuming $v < 1$, then $\gamma$ will be imaginary for $v > 1$; but that is simply not the right normalization factor to use. Furthermore, since $\gamma$ is real, so is the energy $E = \gamma m$. A tachyon need not have imaginary energy or mass, as is sometimes naively claimed.

## 4.2   Why is There a Universal Speed Limit?

A question which often arises when one learns that particles cannot move faster than light is: Why is there a speed limit in the first place? It seems quite arbitrary. One possible, and perhaps unexpected, answer to this question is that we are using the **wrong variable** to measure velocity. The correct variable is called *rapidity*.

### 4.2.1   The Velocity Addition Formula

Let us work in 1+1 spacetime dimensions for simplicity. A Lorentz transformation to a frame with relative velocity $v$ takes the form:

$$\begin{pmatrix} t' \\ x' \end{pmatrix} = \gamma \begin{pmatrix} 1 & -v \\ -v & 1 \end{pmatrix} \begin{pmatrix} t \\ x \end{pmatrix}. \tag{171}$$

Combining it with another transformation of velocity $v'$, we get

$$\begin{pmatrix} t'' \\ x'' \end{pmatrix} = \gamma' \begin{pmatrix} 1 & -v' \\ -v' & 1 \end{pmatrix} \begin{pmatrix} t' \\ x' \end{pmatrix} = \gamma\gamma' \begin{pmatrix} 1+vv' & -v-v' \\ -v-v' & 1+vv' \end{pmatrix} \begin{pmatrix} t \\ x \end{pmatrix}. \tag{172}$$

We may write the product of Lorentz factors as follows:

$$\gamma\gamma' = \frac{1}{\sqrt{1-v^2}} \frac{1}{\sqrt{1-v'^2}} = \frac{1}{(1+vv')\sqrt{1-\left(\frac{v+v'}{1+vv'}\right)^2}}, \tag{173}$$

which then allows us to simplify the combined transformation to:

$$\begin{pmatrix} t'' \\ x'' \end{pmatrix} = \frac{1}{\sqrt{1-\left(\frac{v+v'}{1+vv'}\right)^2}} \begin{pmatrix} 1 & -\frac{v+v'}{1+vv'} \\ -\frac{v+v'}{1+vv'} & 1 \end{pmatrix} \begin{pmatrix} t \\ x \end{pmatrix}. \tag{174}$$

Clearly, the product of two Lorentz transformations with velocities $v$ and $v'$ is another Lorentz transformation with velocity

$$v \oplus v' \equiv \frac{v+v'}{1+vv'}. \tag{175}$$

This is the familiar *velocity addition formula* from special relativity. For small velocities, $v, v' \ll 1$, the denominator is approximately 1 and thus velocities add normally. However, for relativistic velocities, it turns out that we must use this counter-intuitive formula.

### 4.2.2   Rapidity

The velocity addition formula comes from the fact that a Lorentz transformation is actually a type of rotation – more precisely, a *hyperbolic rotation*. This is simply a version of rotation where one of the dimensions involved in the rotation has a negative signature in the metric. Like any rotation, a hyperbolic rotation also has an angle – which is appropriately called a *hyperbolic angle*. The correct variable to parametrize a Lorentz transformation is **not** velocity, but rather this hyperbolic angle – which is sometimes called *rapidity*.

A usual rotation in Euclidean space with angle $\theta$ is given by

$$\begin{pmatrix} x' \\ y' \end{pmatrix} = \begin{pmatrix} \cos\theta & -\sin\theta \\ \sin\theta & \cos\theta \end{pmatrix} \begin{pmatrix} x \\ y \end{pmatrix}. \tag{176}$$

The determinant of the matrix is $\cos^2\theta + \sin^2\theta = 1$, so we know that it only rotates vectors and does not change their magnitude. Also, it is easy to calculate that the product of two rotations with angles $\theta$ and $\theta'$ is a rotation with angle $\theta + \theta'$.

Analogously, a hyperbolic rotation in Minkowski space with hyperbolic angle $\psi$ is given by

$$\begin{pmatrix} t' \\ x' \end{pmatrix} = \begin{pmatrix} \cosh\psi & -\sinh\psi \\ -\sinh\psi & \cosh\psi \end{pmatrix} \begin{pmatrix} t \\ x \end{pmatrix}. \tag{177}$$

The determinant of the matrix is $\cosh^2\psi - \sinh^2\psi = 1$, and it is easy to calculate that the product of two rotations with angles $\psi$ and $\psi'$ is a rotation with angle $\psi + \psi'$. Hence, we learn that we may simply replace the Lorentz transformation matrix with a hyperbolic rotation matrix of hyperbolic angle (or rapidity)

$$\psi = \sinh^{-1}(\gamma v) = \tanh^{-1} v. \tag{178}$$

Indeed, we then have

$$\cosh\psi = \cosh\left(\tanh^{-1} v\right) = \frac{1}{\sqrt{1 - v^2}} = \gamma, \tag{179}$$

and thus

$$\begin{pmatrix} \cosh\psi & -\sinh\psi \\ -\sinh\psi & \cosh\psi \end{pmatrix} = \gamma \begin{pmatrix} 1 & -v \\ -v & 1 \end{pmatrix}. \tag{180}$$

Now, since $v = \tanh\psi$, and using the fact that $\tanh 0 = 0$ and $\lim_{\psi \to \pm\infty} \tanh\psi = \pm 1$, we see that $\psi$ can take any real value, but $v$ will be limited to $|v| \leq 1$. We conclude that there is no "universal rapidity limit"! If rapidity is the variable with which you measure your movement, then you can accelerate as much as you want, never reaching any upper bound. Perhaps, in a relativistic setting, rapidity is the natural variable to use, and velocity is an unnatural variable, which only makes sense in the non-relativistic limit $\psi \to 0$[51].

## 4.3 Tachyons and Time Travel

In the next section, we will present exotic spacetime geometries in general relativity which allow faster-than-light travel globally while lawfully maintaining the speed of light limit locally – by which we mean that massless or massive particles always locally follow null or timelike paths respectively, and no particles **locally** follow spacelike paths, but there is nonetheless some notion in which the particles may **globally** follow spacelike paths.

However, even in the much simpler setting of special relativity, we have seen that it is possible to mathematically describe particles called tachyons, which follow spacelike paths even **locally**, and always travel faster than light. If tachyons exist, then they also allow time travel, or at the very least, communication to the past. Let us see how this works.

Consider a spacetime which is flat and completely empty aside from from two space stations, $S$ and $S'$. Both stations are in inertial motion, and they are moving with respect to each other with constant relative speed $u < 1$. The rest frame of station $S$ has coordinates $(t, x)$ and the rest frame of station $S'$ has coordinates $(t', x')$. Hence, the worldline of station $S$ is along the $t$ axis and the worldline of station $S'$ is along the $t'$ axis.

When station $S$ reached the origin of its coordinate system, $(t, x) = (0, 0)$, it sends a tachyon to station $S'$ at a speed $v > 1$. The tachyon arrives at station $S'$ whenever its worldline

---

[51]A similar case is encountered with temperature. The variable $T$ that we use to measure temperature is unnatural, since negative temperatures (i.e. below absolute zero) exist, and they are in fact **hotter** than any positive temperature! A variable which makes more sense as a fundamental variable is the reciprocal of the temperature, $\beta \equiv 1/T$ (in units where $k_B \equiv 1$), also known as the *coldness*. One finds that $\beta \to +\infty$ corresponds to absolute zero, $T = 0$, and as $\beta$ decreases towards zero (coldness increases, so things get hotter), $T$ approaches $+\infty$. As $\beta$ crosses zero, $T$ suddenly jumps down to $-\infty$, and as $\beta$ decreases towards $-\infty$, $T$ increases to zero from below. Thus negative values of $T$ are indeed hotter than any positive value of $T$, since they correspond to lower values of coldness!

intersects the $t'$ axis. This will necessarily be in the future, since while the tachyon is superluminal, it is still going forward in time. So no time travel has occurred yet. For simplicity, we assume that this point of contact is the origin of the $(t', x')$ coordinate system[52].

When station $S'$ receives the tachyon, which is by assumption at time $t' = 0$, it sends another tachyon back to station $S$ at speed $v' > 1$. Now, from the point of view of $S'$, the tachyon it emits is again going forward in time. This means that its worldline must be above the $x'$ axis. However, by drawing a spacetime diagram as in Fig. 1, one can easily see that, for sufficiently large $u$, the $x'$ axis intersects the $t$ axis at a negative value. Therefore, all $S'$ has to do in order to send a message to the past of $S$ is to make the velocity $v'$ fast enough so that the tachyon ends up at a point on the $t$ axis corresponding to a negative $t$ value (while still having a positive $t'$ value).

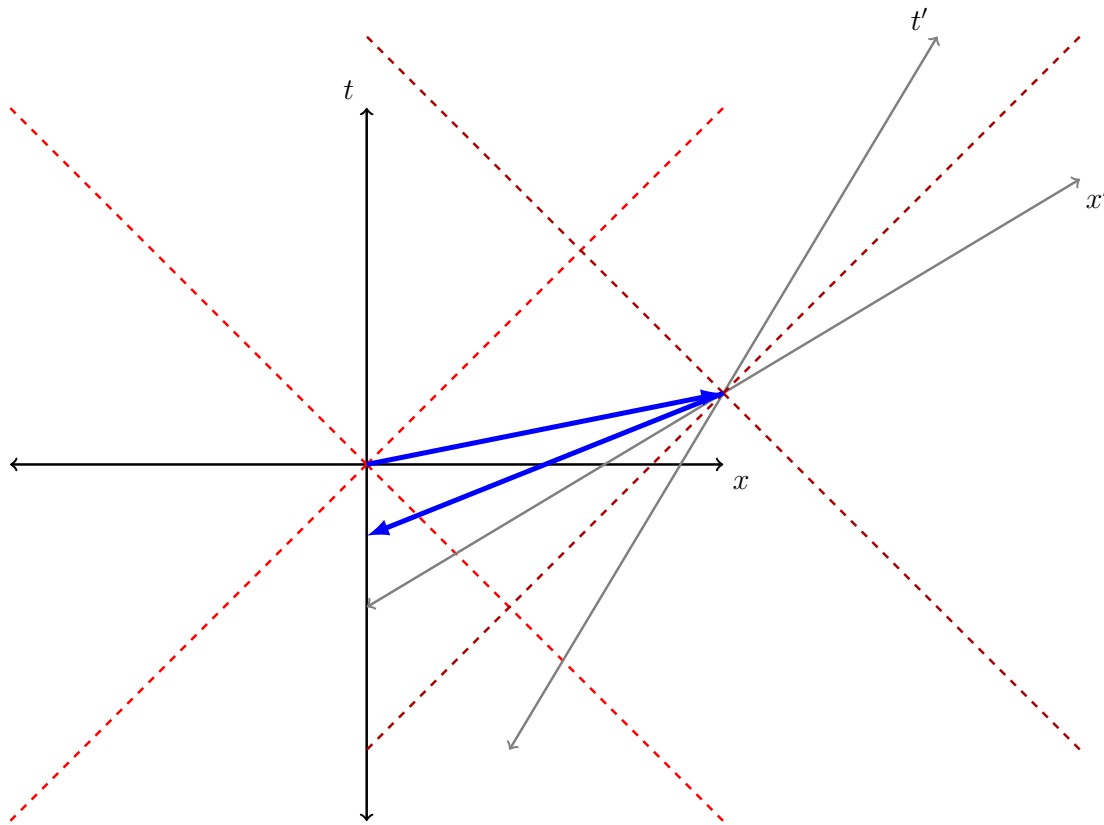

Figure 1: Using tachyons to send a message back in time. The black lines are the axes for the $(t, x)$ system, the gray lines are the axes for the $(t', x')$ system, the bright red lines are the light cones for the $(t, x)$ system, the dark red lines are the light cones for the $(t', x')$ system, and the blue lines are the (spacelike) worldlines of the tachyons.

Now, if the tachyon is used to encode a message[53], a person at space station $S$ may use the tachyon to send a message to their past selves. This will inevitably create a paradox! This kind of paradox is common to all forms of time travel and/or communication to the past, and it resembles the *grandfather paradox,* which we will discuss in detail later. For the tachyon, the paradox may be formulated as follows.

Let us assume that station $S$ sends a tachyon at time $t = 0$ only if it did **not** receive a tachyon at any time $t < 0$. Furthermore, station $S'$ sends a tachyon at time $t' = 0$ only if it

---

[52]This means the coordinate systems are related by both a Lorentz transformation and a spacetime translation, or in other words, a é transformation.

[53]The message could even simply be the detection of the tachyon itself.

**did** receive a tachyon at that time; in other words, it simply acts as a tachyon "mirror". So if $S$ sent a tachyon at $t = 0$, this means that it did **not** receive a tachyon at an earlier time; but in that case, since the sent tachyon was relayed by $S'$, it in fact must have received a tachyon at an earlier time, which means it couldn't have sent the initial tachyon in the first place.

In other words, station $S$ sends a tachyon at $t = 0$ **if and only if** it does not send a tachyon at $t = 0$! This is a paradox, since an event cannot both happen and not happen at the same time.

## 5 Warp Drives

Let us now discuss how, in general relativity, it is in principle possible to travel faster than light globally while still not breaking the speed of light barrier locally. We begin with the warp drive. To gain an intuitive understanding of how it works, we first present the expanding universe as motivation.

### 5.1 Motivation: The Expanding Universe

Our expanding universe is described by the Friedmann–Lemaitre–Robertson–Walker (FLRW) metric, which has the line element:

$$ds^2 = -dt^2 + a(t)^2 d\Sigma^2, \tag{181}$$

where $d\Sigma^2$ is the line element for the 3-dimensional spatial hypersurfaces of constant $t$, which are assumed to have uniform curvature. Usually we take them to be flat, so

$$d\Sigma^2 = dx^2 + dy^2 + dz^2 = dr^2 + r^2 \left( d\theta^2 + \sin^2\theta \, d\phi^2 \right), \tag{182}$$

depending on the coordinate system we use. The function $a(t)$ is called the *scale factor*, and it simply scales the spatial distances measured within the spatial hypersurfaces. At the present time, $a(t)$ is defined to be equal to 1 so that the whole spacetime metric is flat.

We define *proper distance* to be the spatial distance measured with this metric, which is modified by the scale factor. In an expanding universe like ours, $a(t)$ increases with time, and thus if two galaxies are at rest, the proper distance between them nevertheless increases with time. However, we also define *comoving distance*, which factors out the scale factor so that the comoving distance between two galaxies at rest is constant.

A galaxy that is currently a proper distance $D_0$ from us will be a distance

$$D(t) = a(t) D_0 \tag{183}$$

from us at a later time $t$. From this we can calculate the *recession velocity* of the galaxy with respect to us:

$$\dot{D}(t) = \dot{a}(t) D_0 = \frac{\dot{a}(t)}{a(t)} D(t) \equiv H(t) D(t), \tag{184}$$

where $H(t) \equiv \dot{a}(t)/a(t)$ is called the *Hubble parameter*. At the present time we have

$$\dot{D} = H_0 D, \qquad H_0 \approx 70 \, (\text{km/s})/\text{Mpc}. \tag{185}$$

This is *Hubble's law*: the recession velocity of a galaxy from us is proportional to its distance from us.

As a numerical example, a galaxy at a distance $1 \, \text{Mpc} \approx 3 \, \text{Mly} \approx 3 \times 10^{19} \, \text{km}$ away from us has a recession velocity of $\approx 70 \, \text{km/s}$. However, it's easy to see that a galaxy further away than approximately $4.5 \, \text{Gpc} \approx 14 \, \text{Gly}$ is receding faster than light!

Importantly, the recession velocity $\dot{D}$ is only a **global** velocity due to space itself expanding. The galaxy's **local** velocity (sometimes called *peculiar velocity*) in space, relative to nearby galaxies, is independent from $\dot{D}$ and always slower than light, since it locally follows a timelike path. Nonetheless, it seems that we've found a **loophole** in the universal speed limit: While the local velocity of an object **within space** cannot exceed the speed of light, the global velocity due to the expansion of **space itself** is unlimited! We will now see how this loophole is exploited by the warp drive.

## 5.2 The Warp Drive Metric

The Alcubierre warp drive metric, presented by Alcubierre in 1994 [15], is given by the following line element:

$$\mathrm{d}s^2 = -\mathrm{d}t^2 + \mathrm{d}x^2 + \mathrm{d}y^2 + (\mathrm{d}z - v(t)f(r(t))\mathrm{d}t)^2, \tag{186}$$

corresponding to the metric

$$g_{\mu\nu} = \begin{pmatrix} -1 + v^2 f^2 & 0 & 0 & -vf \\ 0 & 1 & 0 & 0 \\ 0 & 0 & 1 & 0 \\ -vf & 0 & 0 & 1 \end{pmatrix}, \tag{187}$$

where:

- $r(t) \equiv \sqrt{x^2 + y^2 + (z - \zeta(t))^2}$ is the spatial distance at time $t$ to the center of the spherical *warp bubble*, located at position $\zeta(t)$ along the $z$ axis at time $t$ and moving at (global) velocity $v(t) \equiv \mathrm{d}\zeta(t)/\mathrm{d}t$. The radius of the bubble is $R$ and the thickness of its wall is $\varepsilon$. The center of the bubble is where our spaceship will be located.

- $f(r(t))$ is the *form function*; its exact form does not matter, as long as it has the value $f(0) = 1$ at the center of the bubble and $f \to 0$ as $r \to \infty$ outside the bubble, and drops sharply from $\approx 1$ to $\approx 0$ at the wall of the bubble, at $r = R$. How sharp the transition is depends on the wall thickness $\varepsilon$. This means spacetime is approximately flat everywhere outside the bubble. An example of a possible form function is

$$f(r) \equiv \frac{\tanh \frac{r+R}{\varepsilon} - \tanh \frac{r-R}{\varepsilon}}{2 \tanh \frac{R}{\varepsilon}}. \tag{188}$$

In the limit $\varepsilon \to 0$, this function approaches a step function:

$$\lim_{\varepsilon \to 0} f(r) = \begin{cases} 1 & \text{if } r \in [0, R], \\ 0 & \text{if } r \in (R, \infty). \end{cases} \tag{189}$$

However, for positive $\varepsilon$ there is a smooth transition, as can be seen in Fig. 2.

- For constant $t$ hypersurfaces, we have $\mathrm{d}t = 0$ and thus the induced metric is simply the flat Euclidean metric.

The crucial fact about this metric is that the velocity $v$ of the bubble can be faster than light! We will now study how this is possible.

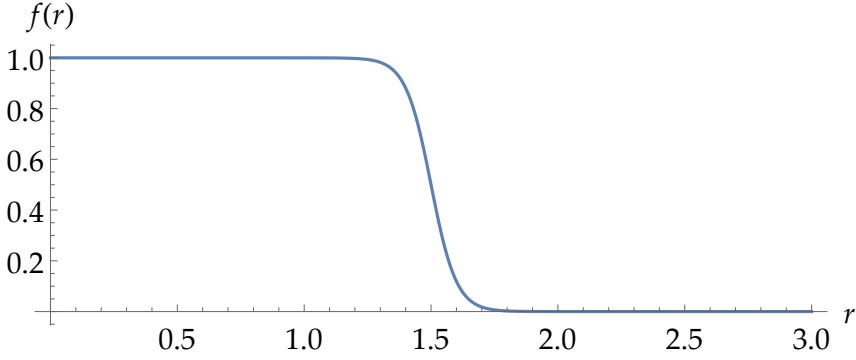

Figure 2: The function $f(r)$ with $R = 1.5$ and $\varepsilon = 0.1$.

## 5.3 Properties of the Metric

### 5.3.1 (No) Time Dilation

For any $v$, even $v > 1$, if we substitute the location of the center of the bubble $(x, y, z) = (0, 0, \zeta)$ into the metric (186) we get

$$\mathrm{d}s^2 = -\mathrm{d}t^2, \tag{190}$$

since $f(0) = 1$, $\mathrm{d}x = \mathrm{d}y = 0$ and

$$\mathrm{d}z = \mathrm{d}\left(\zeta(t)\right) = \frac{\mathrm{d}\zeta}{\mathrm{d}t}\mathrm{d}t = v(t)\,\mathrm{d}t. \tag{191}$$

Since the proper time is given by $\mathrm{d}t = \sqrt{-\mathrm{d}s^2}$, we see that

$$\mathrm{d}\tau = \mathrm{d}t. \tag{192}$$

Hence, the proper time of the spaceship inside the bubble is the same as the coordinate time. The spaceship will experience no time dilation during the journey.

This is very different from the case of a spaceship traveling normally (e.g. using rockets) at close to the speed of light, using just the usual laws of special relativity. The journey to a distant star at close to light speed might take a very short proper time, but will correspond to a very long coordinate time.

As a concrete example, let us assume the ship travels at $v = 0.999999$, which corresponds to a time dilation factor $\gamma = 1000$. The crew travels to a star 100 light years away and back, and the entire round trip journey only lasts about 10 weeks. However, when they arrive back on Earth, they see that 200 years have passed, and everyone they know is long dead. The warp drive solves this problem; since there is no time dilation, only 10 weeks will have passed on Earth during the journey[54].

### 5.3.2 Timelike Paths and Tilted Light Cones

Globally, the path taken by the bubble is going to be spacelike if $v > 1$. This does not violate anything in relativity, since the bubble is not a massive object obeying the geodesic equation; it is just a specific choice of the curvature of spacetime itself. Furthermore, from the fact that $\mathrm{d}s^2 < 0$ we learn that, even for $v > 1$, the spaceship travels along a **timelike** path **inside** the bubble, as expected from a massive object. (Below we will show it is, in fact, at rest!)

---

[54]We have, of course, neglected issues of acceleration in this example.

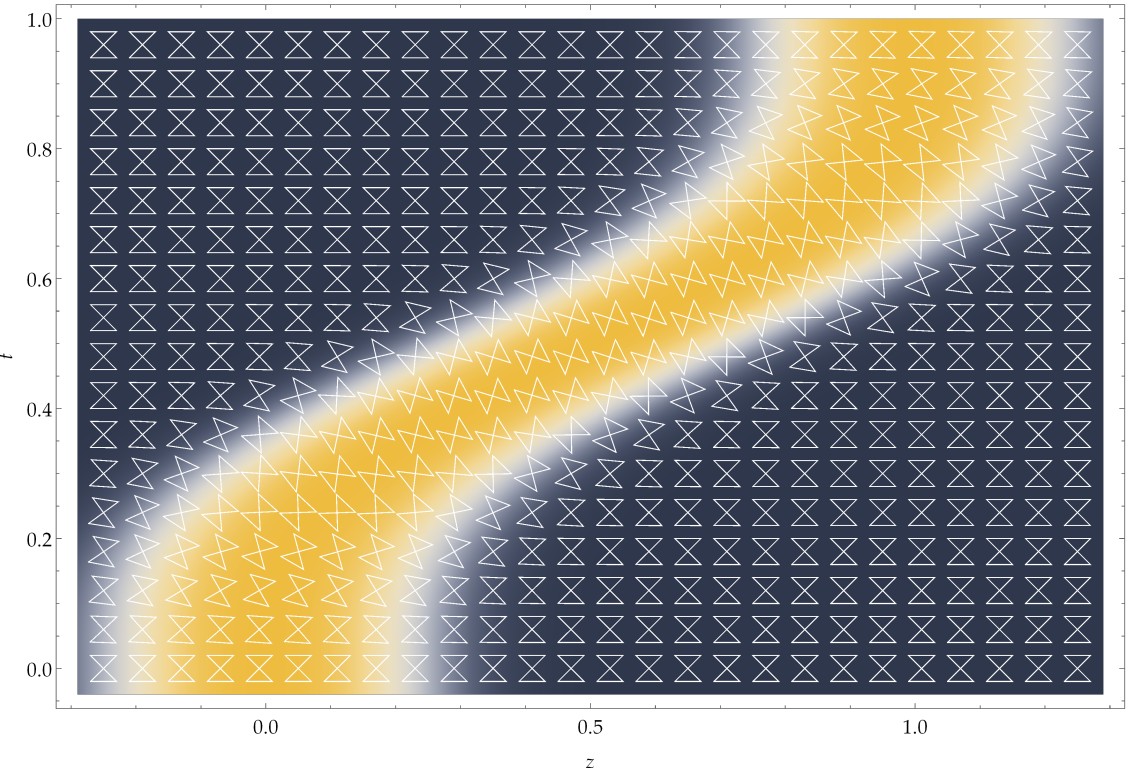

Figure 3: A spacetime diagram for a warp bubble (in yellow) with $R = 0.25$, $\varepsilon = 0.1$, and $v(t) = 2\sin^2(\pi t)$. It starts at rest at $t = 0$, accelerates up to a maximum velocity of $v = 2$ at $t = 0.5$, and then decelerates to rest at $t = 1$. The light cones have slopes $\pm 1 + v f$, so when $v > 0$ they are "tipped over" inside the bubble such that the spaceship locally remains within its own light cone at all times.

To see this more clearly, let us calculate the light cones of this metric, which correspond to $\mathrm{d}s^2 = 0$. We will only look at the light cones in the $z$ direction, so $\mathrm{d}x = \mathrm{d}y = 0$. We find

$$\frac{\mathrm{d}z}{\mathrm{d}t} = \pm 1 + v(t)f(r(t)). \tag{193}$$

Outside the bubble, where $f \approx 0$, we have $\mathrm{d}z/\mathrm{d}t \approx \pm 1$ which is the usual light cone for a flat spacetime. However, inside the bubble, where $f \approx 1$, we have a "tilt" which is proportional to $v$. This is illustrated in Fig. 3. It is now clear that, even though the spaceship travels faster than light globally, it is locally still within its own light cone along the path, so no relativistic traffic laws are broken.

### 5.3.3 Geodesics and Free Fall

Let us calculate the geodesics of the warp drive metric. Recall that a geodesic is calculated by finding an extremum (which will, in fact, be a maximum) of the proper time, given by (9):

$$\tau = \int \sqrt{-g_{\mu\nu}\dot{x}^\mu \dot{x}^\nu}\,\mathrm{d}\lambda. \tag{194}$$

In our case, for the metric (186), we have

$$\tau = \int \sqrt{-\dot{t}^2 + \dot{x}^2 + \dot{y}^2 + (\dot{z} - v(t)f(r(t))\dot{t})^2}\,\mathrm{d}\lambda. \tag{195}$$

Since the square root is a monotonically increasing function, we may instead maximize the following Lagrangian:

$$L = \frac{1}{2}\left(-\dot{t}^2 + \dot{x}^2 + \dot{y}^2 + (\dot{z} - V(\mathbf{x})\,\dot{t})^2\right), \tag{196}$$

where we defined $V(\mathbf{x}) \equiv v(t)f(r(t))$ for simplicity; we treat it as a general function which depends on all 4 coordinates (but not their derivatives). Furthermore, we parameterize the geodesic using the proper time $\tau$ itself, which automatically normalizes it as a 4-velocity, $|\dot{\mathbf{x}}|^2 = -1$. Thus, the dot is now a derivative with respect to $\tau$.

To maximize this Lagrangian, we use the Euler-Lagrange equations:

$$\frac{\mathrm{d}}{\mathrm{d}\tau}\left(\frac{\partial L}{\partial \dot{x}^\mu}\right) - \frac{\partial L}{\partial x^\mu} = 0. \tag{197}$$

In total we have 4 equations, one for each value of $\mu$. We leave the $\tau$ derivatives implicit[55]:

$$\frac{\mathrm{d}}{\mathrm{d}\tau}\left(-\dot{t} - V(\mathbf{x})(\dot{z} - V(\mathbf{x})\,\dot{t})\right) + \frac{\partial V(\mathbf{x})}{\partial t}\dot{t}\,(\dot{z} - V(\mathbf{x})\,\dot{t}) = 0, \tag{198}$$

$$\frac{\mathrm{d}\dot{x}}{\mathrm{d}\tau} + \frac{\partial V(\mathbf{x})}{\partial x}\dot{t}\,(\dot{z} - V(\mathbf{x})\,\dot{t}) = 0, \qquad \frac{\mathrm{d}\dot{y}}{\mathrm{d}\tau} + \frac{\partial V(\mathbf{x})}{\partial y}\dot{t}\,(\dot{z} - V(\mathbf{x})\,\dot{t}) = 0, \tag{199}$$

$$\frac{\mathrm{d}}{\mathrm{d}\tau}(\dot{z} - V(\mathbf{x})\,\dot{t}) + \frac{\partial V(\mathbf{x})}{\partial z}\dot{t}\,(\dot{z} - V(\mathbf{x})\,\dot{t}) = 0. \tag{200}$$

It's easy to see that

$$\dot{x}^\mu = (1, 0, 0, v(t)f(r(t))) \tag{201}$$

solves all 4 equations, since then we have $\dot{z} - V(\mathbf{x})\,\dot{t} = 0$.

Observers with the 4-velocity (201) are called *Eulerian observers*. They are following timelike geodesics, even though, at $r = 0$ where $f = 1$, their spatial velocity is $v$ which may be faster than light! Note that

$$\dot{x}_\mu = g_{\mu\nu}\dot{x}^\mu = (-1, 0, 0, 0), \tag{202}$$

which is the covector normal to the hypersurfaces given by $\mathrm{d}t = 0$, which as we said above, have a flat Euclidean metric. (Of course, $\dot{x}_\mu\dot{x}^\mu = -1$ as expected.)

### 5.3.4 Expansion and Contraction of Space

How exactly does the warp bubble move? Just like with the expansion of the universe, where the galaxies move faster than light with respect to each other due to the expansion of space itself, here also the spaceship moves faster than light with respect to its point of origin (or the coordinate system) due to the expansion and contraction of space around the warp bubble.

We are essentially using the same "loophole". Space expands behind the warp bubble and contracts in front of it, thus pushing the bubble forward at velocity $v$. The ship, which is **at rest** inside the bubble, moves along with the bubble at an arbitrarily large **global** velocity.

To see how this works, recall that if the unit normal vector $n^\mu$ is tangent to a geodesic, then from (67) the expansion $\theta$ is given by the trace of the extrinsic curvature:

$$\theta = K^\mu_\mu = \nabla_\mu n^\mu. \tag{203}$$

---

[55]Leaving the derivative implicit simplifies things, since we always end up taking the derivative of a constant (or zero) anyway.

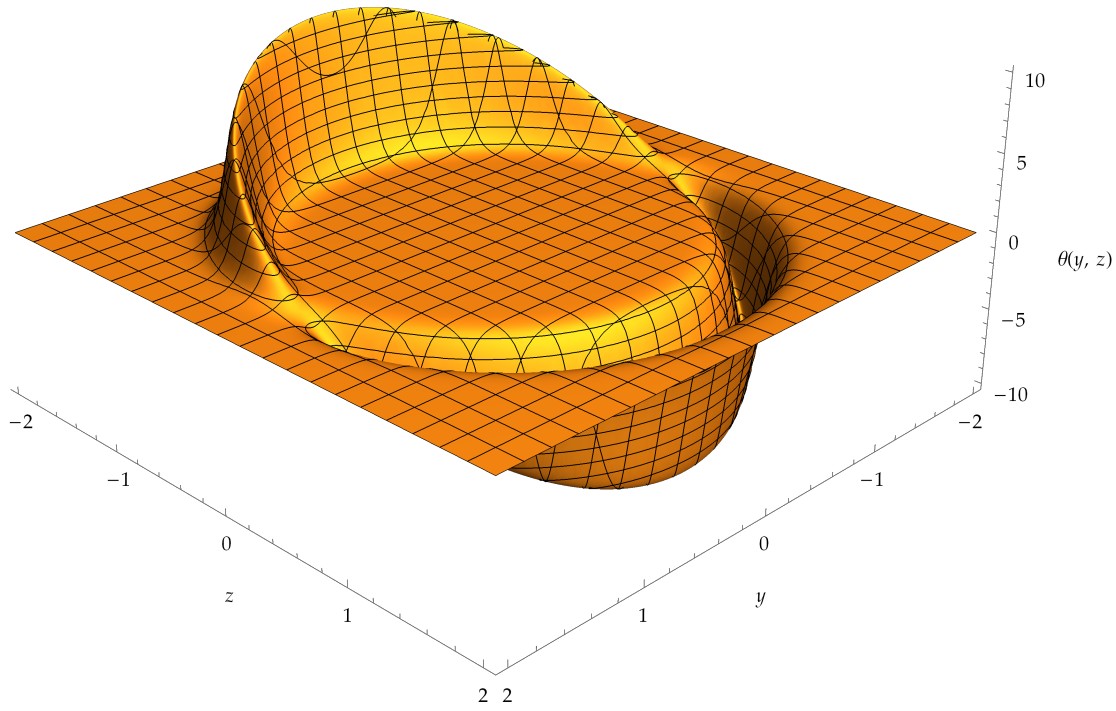

Figure 4: A plot of the expansion $\theta$ of a warp drive bubble with $v = 2$, $R = 1.5$, and $\varepsilon = 0.1$ moving along the positive $z$ axis.

In this case, as we have seen above, the unit normal covector is $n_\mu = (-1, 0, 0, 0)$, which is dual to the vector $n^\mu = (1, 0, 0, vf)$, which is indeed tangent to a geodesic. Thus

$$\theta = \nabla_z n^z = \partial_z \left(v(t) f(r(t))\right) = \frac{z - \zeta}{r(t)} v(t) f'(r(t)), \tag{204}$$

where we replaced the covariant derivative with a partial derivative since the metric on the surface is flat.

Now, since $f \approx 1$ inside the bubble and $f \approx 0$ outside the bubble, we have that $\theta \approx 0$ in those regions. The only place where we would have $\theta \neq 0$ is on the wall of the bubble, where, as $r$ (the distance from the center of the bubble) increases, $f$ sharply changes from 1 to 0, and thus has a negative derivative, $f'(r(t)) < 0$. Behind the bubble we have $z < \zeta$ and thus $\theta > 0$, so space is expanding; in front of the bubble we have $z > \zeta$ and thus $\theta < 0$, so space is contracting[56]. This is illustrated in Fig. 4.

## 5.4 Violations of the Energy Conditions

Let us plug the warp drive metric into the left-hand side of the Einstein equation (73):

$$R_{\mu\nu} - \frac{1}{2} R g_{\mu\nu} = 8\pi T_{\mu\nu}. \tag{205}$$

---

[56]Interestingly, Natario [17] has generalized the warp drive metric and shown that it is in fact possible to construct it in such a way that $\theta = 0$ everywhere, so space does not expand nor contract! Instead, his interpretation is that the warp bubble simply "slides" through space, taking the spaceship with it, and the expansion and contraction are not necessary for the warp drive to function.

Let the timelike vector $t^\mu$, normalized such that $|\mathbf{t}|^2 = -1$, be the 4-velocity of an Eulerian observer. We have

$$T_{\mu\nu}t^\mu t^\nu = \frac{1}{8\pi}\left(R_{\mu\nu} - \frac{1}{2}Rg_{\mu\nu}\right)t^\mu t^\nu. \tag{206}$$

The calculation of the right-hand side is tedious, but it has been done (see e.g. [18]), and the result is

$$T_{\mu\nu}t^\mu t^\nu = -\frac{v^2}{32\pi}\left(\left(\frac{\partial f}{\partial x}\right)^2 + \left(\frac{\partial f}{\partial y}\right)^2\right) \le 0, \tag{207}$$

which is always non-positive, and zero only if $v = 0$ and/or $f = 0$. Thus, the weak energy condition is violated!

In particular, since $f$ is a function of $r(t) \equiv \sqrt{x^2 + y^2 + (z - \zeta(t))^2}$, we may write this as

$$T_{\mu\nu}t^\mu t^\nu = -\frac{v^2}{32\pi}\frac{x^2 + y^2}{r^2}\left(\frac{\mathrm{d}f}{\mathrm{d}r}\right)^2 \le 0. \tag{208}$$

Just like with the expansion, the dependence on the derivative $f'$ means that the violation of the energy conditions is negligible inside and outside the bubble, and becomes noticeable only at the wall of the bubble, where $f'$ sharply changes from 1 to 0. This is illustrated in Fig. 5.

The expression $T_{\mu\nu}t^\mu t^\nu$ is the energy density measured by an observer with 4-velocity $t^\mu$, and we may integrate it over a spatial slice to find the total energy:

$$E = \int T_{\mu\nu}t^\mu t^\nu \mathrm{d}^3x = -\frac{v^2}{32\pi}\int \frac{x^2 + y^2}{r^2}\left(\frac{\mathrm{d}f}{\mathrm{d}r}\right)^2 r^2\,\mathrm{d}r\,\mathrm{d}^2\Omega = -\frac{v^2}{12}\int \left(\frac{\mathrm{d}f}{\mathrm{d}r}\right)^2 r^2\,\mathrm{d}r. \tag{209}$$

Given (188), we may estimate

$$E \approx -\frac{v^2 R^2}{\varepsilon}. \tag{210}$$

Thus, we need a larger amount of negative energy for higher bubble velocities, larger bubble sizes, and thinner bubble walls, which makes sense.

A slightly better estimate is given by Pfenning and Ford [19]. They use a piecewise-continuous function instead of (188):

$$f(r) = \begin{cases} 1 & r \le R - \frac{\varepsilon}{2}, \\ \frac{1}{2} + \frac{R-r}{\varepsilon} & r \in \left(R - \frac{\varepsilon}{2}, R + \frac{\varepsilon}{2}\right), \\ 0 & r \ge R + \frac{\varepsilon}{2}, \end{cases} \tag{211}$$

where again $R$ is the radius of the bubble and $\varepsilon$ is its wall thickness. Plugging this into (209), we get

$$E = -\frac{v^2}{12}\int_{R-\varepsilon/2}^{R+\varepsilon/2}\frac{1}{\varepsilon^2}r^2\mathrm{d}r = -\frac{v^2}{12}\left(\frac{R^2}{\varepsilon} + \frac{\varepsilon}{12}\right). \tag{212}$$

Note that this is assuming $v$ is constant. Taking $R = 100\,\mathrm{m}$ and assuming $\varepsilon \le 10^2 v$ in Planck units we get

$$E \le -6.2 \times 10^{70}v \approx 1.3 \times 10^{63}v\,\mathrm{kg}. \tag{213}$$

This is 10 orders of magnitude larger than the mass of the observable universe, which is roughly estimated at $10^{53}\,\mathrm{kg}$. This by itself is already bad news, but on top of that, the energy must also be negative!

However, it is unclear how meaningful this result is. One issue comes from the requirement that $\varepsilon \le 10^2 v$, which is only relevant if our source of exotic matter is a quantum field (and even then, relies on some assumptions which may or may not be applicable). The reason is that quantum fields must obey the *quantum energy inequality* [20], which is a limitation on

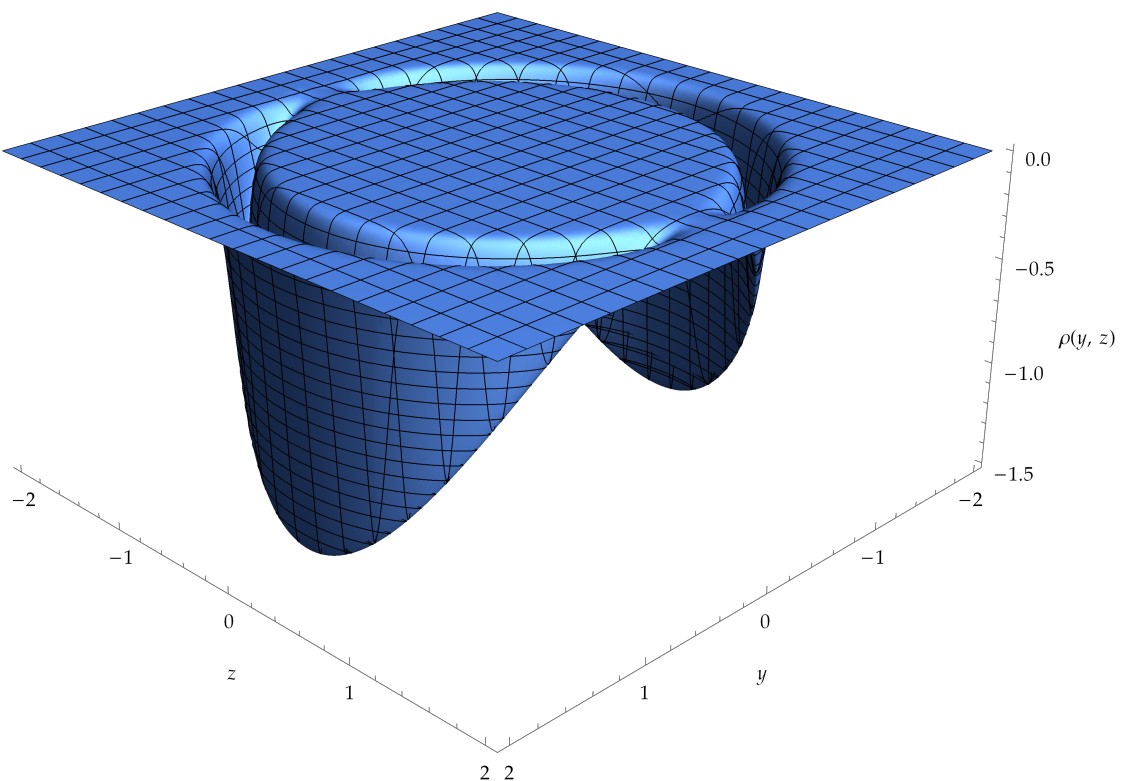

Figure 5: The distribution of energy density around a warp drive bubble with $v = 2$, $R = 1.5$, and $\varepsilon = 0.1$ moving along the positive $z$ axis.

how much negative energy may be contributed by the field over some period of time – and it is a pretty small amount[57].

If the negative energy instead comes from a classical field, then this restriction does not apply, and we may allow a thicker wall, for example with $\varepsilon \approx 1\,\mathrm{m}$. In this case, we would only need

$$E \leq -5.0 \times 10^{29}\,\mathrm{kg} \tag{214}$$

of exotic matter, which is roughly of the order of the mass of the sun, and thus perhaps a bit more realistic – provided, of course, that a suitable form of exotic matter exists in this amount.

## 5.5 The Event Horizon

Let us write down the warp drive metric in the reference frame of an observer on the spaceship. Their $z$ coordinate becomes $z' \equiv z - \zeta(t)$, so (186) becomes

$$ds^2 = -dt^2 + dx^2 + dy^2 + \left(dz' + v(t)(1 - f(r(t)))\,dt\right)^2. \tag{215}$$

Now, suppose the engineers on the spaceship want to modify the geometry of the warp bubble – for example, in order to accelerate or to stop moving. In order to do that, they must be able to causally influence the bubble wall in front of them. Let us assume they try to do so by sending a photon forward towards the bubble wall. The photon has $ds^2 = 0$, and it is going in the $z'$ direction, so $dx = dy = 0$. Thus, we obtain from the metric that

$$\frac{dz'}{dt} = 1 - v(1 - f), \tag{216}$$

---

[57]We will not go into the details of the quantum energy inequality here, since that would require knowledge of quantum field theory.

where we chose the positive square root so that the photons moves along the positive direction of the $z'$ axis. At the center of the bubble we have $f = 1$, so $dz'/dt = 1$ and the photon moves at the speed of light with respect to the spaceship (in the original frame, where $dz/dt = 1+vf$, it moves at speed $1+v$).

Now, if $v < 1$, then since $f \in [0,1]$, $dz'/dt$ will always remain positive. However, if $v > 1$ then, at some point $z = z_h$ where $f = 1-1/v$, the photon's speed will become zero and it will therefore remain forever at that point. Thus, there is a *horizon* at $z_h$.

Since $f = 0$ outside the bubble, this means the photon will be stuck somewhere inside the bubble wall. Therefore, the outer edge of the bubble wall is outside the forward light cone of the spaceship, or more precisely, outside of its causal future $J^+$. From this, we conclude that the crew on the spaceship cannot modify the bubble, and thus cannot control their journey, which severely limits the feasibility of the warp drive! However, perhaps it could be possible, in principle, that the entire trajectory of the bubble may be constructed beforehand.

Let us see the existence of the horizon more precisely. We will follow a paper by Hiscock [21]. For simplicity, we work in 2 dimensions $(t,x)$, so the Alcubierre metric is

$$ds^2 = -\left(1 - v^2 f^2\right)dt^2 - 2vf\,dt \otimes dx + dx^2, \tag{217}$$

and we will assume that $v$ is constant. Let us now transform to coordinates $(t,r)$ by taking $dx = dr + v\,dt$:

$$ds^2 = -A(r)\left(dt - \frac{v(1-f)}{A(r)}dr\right)^2 + \frac{dr^2}{A(r)}, \tag{218}$$

where

$$A(r) \equiv 1 - v^2(1-f)^2. \tag{219}$$

We can then make the obvious transformation, $d\tau = dt - \frac{v(1-f)}{A(r)}dr$, to diagonalize the metric:

$$ds^2 = -A(r)d\tau^2 + \frac{dr^2}{A(r)}. \tag{220}$$

Note that this metric is static, since $A$ does not depend on $\tau$. Looking at this metric, we see that we run into a problem. We have $(1-f)^2 \in [0,1]$, and therefore, if $v < 1$ we have $A > 0$. However, for $v > 1$ there is a point $r = r_h$ where $A = 0$, given implicitly by $f(r_h) = 1-1/v$. At this point, there is a coordinate singularity, and an event horizon forms[58].

## 5.6 Using a Warp Drive for Time Travel

Since a warp drive allows one to travel along paths which globally seem spacelike, the discussion of tachyons in Sec. 4.3 applies to a warp drive as well. Of course, here the spacetime is not entirely flat, but it is essentially flat everywhere except inside the bubble, so this should not invalidate the results of the tachyon discussion.

---

[58]This situation is very similar to the event horizon of a Schwarzschild black hole, which has the metric:

$$ds^2 = -A(r)dt^2 + \frac{dr^2}{A(r)} + r^2\left(d\theta^2 + \sin^2\theta\,d\phi^2\right), \qquad A(r) \equiv 1 - \frac{2M}{r}, \tag{221}$$

where $M$ is the mass of the black hole. Since $A = 0$ when $r = 2M$, there is a coordinate singularity at that point. This is where the event horizon forms. However, it is not an actual singularity – only an artifact of the particular system of coordinates we chose; the curvature is in fact finite there. The point at $r = 0$, where $A$ diverges, is an actual singularity since there is a scalar quantity, called the *Kretschmann scalar*, which diverges there: $R^{\mu\nu\alpha\beta}R_{\mu\nu\alpha\beta} = 48M^2/r^6 \to \infty$. Since this is a scalar, it is independent of any choice of coordinates.

Intuitively, when $A$ changes sign at the event horizon $r = 2M$, the $t$ and $r$ directions switch places: $t$ becomes spacelike and $r$ becomes timelike. In a typical spacetime, where $t$ is the timelike coordinate, we must always travel along the $t$ direction whether we want to or not. Inside the event horizon, where $r < 2M$, we must always travel along the negative $r$ direction – and thus into the singularity at $r = 0$ – whether we want to or not.

However, it is important to clarify several key differences between a warp drive and a tachyon:

- As we have seen, a tachyon must always move faster than light; it can never slow down to the speed of light or slower. A warp drive, in principle, can travel at any speed, and should be able to accelerate all the way from zero to arbitrarily high speeds, with the only difference between $v < 1$ and $v > 1$ being the appearance of a horizon.

- A tachyon does not depend on the spacetime geometry to move faster than light. A warp drive requires a very specific spacetime geometry.

- A particle inside a warp drive always moves along a locally timelike path; in fact, it's at rest. A tachyon always moves along a locally (and globally) spacelike path. Usually the expression "closed timelike curve" is synonymous with "time machine", but we see that while warp drives can be used to create a closed timelike curve, tachyons allow a different form time travel, which does not involve closed timelike curves.

There is one important similarity, though. We have never seen any tachyons in reality; and warp drives may be impossible to construct, due to the absence of exotic matter. Both concepts seem more like science fiction than science; but unlike science function, they may be represented using concrete mathematical models, within the context of physical theories that are known to describe the real universe. Therefore, whether or not tachyons or warp drives can actually exist in reality, studying them might teach us a lot about how our universe works.

# 6 Wormholes

## 6.1 The Traversable Wormhole Metric

The *Morris-Thorne wormhole* [22], a traversable, static, spherically symmetric wormhole[59] may be described, in spherical coordinates $(t, r, \theta, \phi)$, by the following line element:

$$ds^2 = -e^{2\Phi(r)} dt^2 + \frac{dr^2}{1 - b(r)/r} + r^2 \left( d\theta^2 + \sin^2\theta \, d\phi^2 \right), \tag{222}$$

where:

- There are two coordinate patches, both covering the range $r \in [r_0, \infty)$.

    - The two patches corresponds to two different universes, or two different locations in the same universe, each on one side of the wormhole.

    - The value $r_0$ satisfies $b(r_0) = r_0$, and it is where the wormhole's "throat" is located; it is the passage between the two patches.

    - Note that there is a coordinate singularity at $r_0$, but it turns out to not be a problem; the Riemann tensor is continuous across the throat.

- $\Phi(r)$ is called the *redshift function*, since it is related to gravitational redshift.

    - It must be finite everywhere, including at $r \to \infty$, since if we have $g_{tt} = 0$, an event horizon will form and the wormhole won't be traversable.

---

[59]There are many other types of wormholes studied in the literature, although many of them are only of historical significance. Visser's book [1] provides a very thorough introduction to virtually all known types of wormholes.

- For simplicity we assume that $\Phi(r)$ is the same function in both coordinate patches; if it's not, then time flows at different rates on each side.

- $b(r)$ is called the *shape function*, and it determines the wormhole's shape[60].

    - For simplicity we assume that $b(r)$ is the same function in both coordinate patches; if not, then the wormhole has a different shape on each side.

    - Away from the throat, we have $b(r) < r$, so that $1 - \frac{b(r)}{r} > 0$.

- The *proper radial distance* $\ell$ is given by:

$$\frac{\mathrm{d}\ell}{\mathrm{d}r} = \pm \frac{1}{\sqrt{1 - \frac{b(r)}{r}}} \quad \Longrightarrow \quad \ell(r) \equiv \pm \int_{r_0}^{r} \frac{1}{\sqrt{1 - \frac{b(r)}{r}}} \mathrm{d}r, \tag{223}$$

and it is required to be finite everywhere[61].

    - The two signs correspond to the two separate coordinate patches. The range $\ell \in (-\infty, 0)$ corresponds to one coordinate patch and one universe, while the range $\ell \in (0, +\infty)$ corresponds to the other coordinate patch and either another universe, or another location in the same universe.

    - The throat is located at $\ell = 0$, where $r(\ell = 0) = r_0 = \min[r(\ell)]$.

    - The two regions at the limits $\ell \to \pm\infty$ are asymptotically flat, so we must have $\lim_{\ell \to \pm\infty} \Phi = 0$ and $\lim_{\ell \to \pm\infty} b/r = 0$ in order to get the Minkowski metric there.

The coordinate $r$ is a bit confusing. Let us embed the wormhole in a 3-dimensional space, as in Fig. 6. Then $r$ is the radius of each circle that we see in the figure. At the top universe, at $\ell \to +\infty$, we have $r \to +\infty$. Then, as we go south towards the throat, $r$ decreases monotonically until it reaches the minimum value $r_0$, which is the radius of the throat – the smallest circle in the figure. As we keep going south into the bottom universe, $r$ now increases monotonically until, at $\ell \to -\infty$, we have $r \to +\infty$ again. Note that $r$ is always positive (and larger than $r_0$), even though $\ell$ can take any real value.

## 6.2 Violations of the Energy Conditions

Now, in an orthonormal basis (find it!), the non-zero components of the Einstein tensor are:

$$G_{tt} = \frac{b'}{r^2}, \qquad G_{rr} = 2\left(1 - \frac{b}{r}\right)\frac{\Phi'}{r} - \frac{b}{r^3}, \tag{224}$$

$$G_{\theta\theta} = G_{\phi\phi} = \left(1 - \frac{b}{r}\right)\left(\Phi'' + \Phi'\left(\Phi' + \frac{1}{r}\right)\right) - \frac{b'r - b}{2r^2}\left(\Phi' + \frac{1}{r}\right). \tag{225}$$

Using the Einstein equation, $G_{\mu\nu} = 8\pi T_{\mu\nu}$, we see that the energy-momentum tensor must have the same components (divided by $8\pi$). In particular, the energy density is

$$T_{tt} = \rho(r) = \frac{1}{8\pi}\frac{b'}{r^2}, \tag{226}$$

and it is negative whenever $b' < 0$. We also find that the pressure along the radial direction is

$$T_{rr} = p(r) = \frac{1}{8\pi}\left(2\left(1 - \frac{b}{r}\right)\frac{\Phi'}{r} - \frac{b}{r^3}\right). \tag{227}$$

---

[60]By comparison with the Schwarzschild metric (see Footnote 58), we see that the mass of the wormhole is given by $M = \lim_{r \to \infty} b(r)/2$.

[61]Recall that an integral can converge even if the integrand diverges at the boundary of the integration region

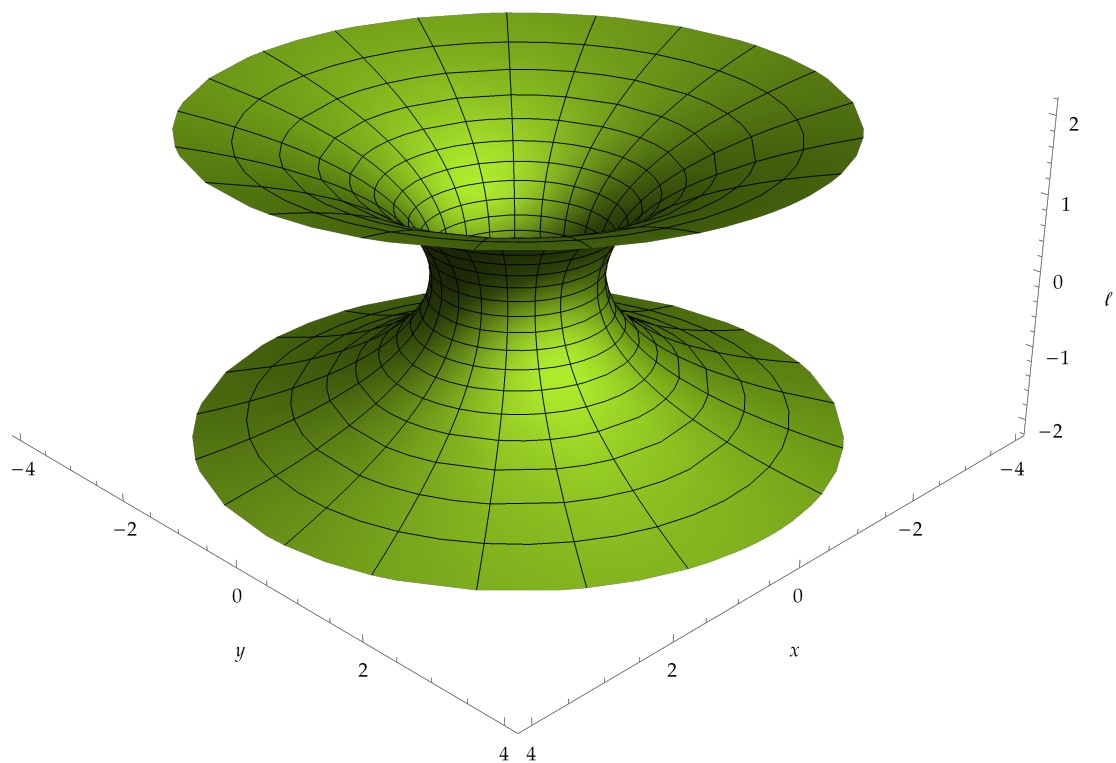

Figure 6: A slice of the wormhole geometry at the equator, $\theta = \pi/2$, and at a constant value of $t$. The upper universe corresponds to $\ell \to +\infty$, while the lower universe (which we will assume is another part of the same universe) corresponds to $\ell \to -\infty$. Note that each circle is actually a slice of a sphere; we are simply suppressing other values of $\theta$ on the sphere so that we can plot the wormhole in three dimensions.

Recalling from Sec. 3.2.6 that the null energy condition is equivalent to $\rho + p \geq 0$, we calculate this quantity and find:

$$
\begin{aligned}
\rho(r) + p(r) &= -\frac{1}{8\pi}\frac{1}{r}\left(\frac{1}{r}\left(\frac{b}{r} - b'\right) - 2\Phi'\left(1 - \frac{b}{r}\right)\right) \\
&= -\frac{1}{8\pi}\frac{1}{r}\left(\frac{\mathrm{d}}{\mathrm{d}r}\left(1 - \frac{b}{r}\right) - 2\Phi'\left(1 - \frac{b}{r}\right)\right) \\
&= -\frac{1}{8\pi}\frac{e^{2\Phi}}{r}\frac{\mathrm{d}}{\mathrm{d}r}\left(e^{-2\Phi}\left(1 - \frac{b}{r}\right)\right).
\end{aligned}
$$

Now, at the throat, $r = r_0$, we have $b(r_0) = r_0$, so

$$
\left. e^{-2\Phi}\left(1 - \frac{b}{r}\right)\right|_{r=r_0} = 0. \tag{228}
$$

Away from the throat we have $b(r) < r$, so

$$
\left. e^{-2\Phi}\left(1 - \frac{b(r)}{r}\right)\right|_{r>r_0} > 0. \tag{229}
$$

Thus, there must be some point $r_*$ such that

$$
\left. \frac{\mathrm{d}}{\mathrm{d}r}\left(e^{-2\Phi}\left(1 - \frac{b(r)}{r}\right)\right)\right|_{r\in(r_0,r_*)} > 0. \tag{230}
$$

In conclusion, we have

$$\rho(r) + p(r)\bigg|_{r \in (r_0, r_*)} < 0, \tag{231}$$

and thus there is a non-zero region where the null energy condition is violated. This implies a violation of the weak, strong and dominant energy conditions.

Exactly how much exotic matter is needed depends on many different factors, and we won't go into that here. However, there are some indications that the required amount of negative energy can be made arbitrarily small; see, for example, the paper by Visser, Kar, and Dadhich [23]. Furthermore, Barceló and Visser [24] managed to create a traversable wormhole from a classical conformally-coupled scalar field by employing the violations of the energy conditions we discussed in Subsection 3.3.2.

There are many more things one can study about wormholes, including questions of traversability, horizons, singularities, tidal forces, geodesics, and more. These discussions are beyond the scope of our lecture notes; here we just wanted to give the basic details. The reader is referred to the vast literature on wormholes for more information; [1] and [2] are good places to start.

### 6.3 Using a Wormhole for Time Travel

Creating a time machine using a wormhole is fairly easy. Consider a wormhole with its two mouths located in the same universe, a distance $L$ from each other. Normally, if you enter one mouth, you exit through the other mouth at the same value of the time coordinate. Now, assuming it is possible to move one of the mouths (presumably by moving the exotic matter used to construct it?), we may use special- or general-relativistic time dilation in order to introduce a relative time shift $T$ between the two mouths.

For special-relativistic time dilation, we move one of the mouths at close to light speed; this will be a version of the twin "paradox", where the mouths play the roles of the twins. For general-relativistic time dilation, we place the mouth near a strong gravitational source. Finally, we move the mouths closer together.

Once the distance $L$ is less than the time shift $T$, a closed timelike curve forms. Indeed, using coordinates $(t, x)$, the first wormhole mouth follows the path $(t - T, 0)$, while the other one follows the path $(t, L)$, both paths parameterized by $t$. The existence of the wormhole means that the following points are identified along the paths, for any $t$:

$$(t - T, 0) \sim (t, L). \tag{232}$$

Now, consider a spaceship at the first mouth, at $(0, 0)$. The ship moves at close to the speed of light until it covers a distance $L$, which would roughly take time $L$. Therefore, its location is now $(L, L)$. At this location, it enters the second mouth, and arrives almost instantaneously back at the first mouth, at $(L - T, 0)$. Thus, if $L < T$, the ship has followed a closed timelike curve and arrived at the first wormhole before it left!

# 7 Time Travel Paradoxes and Their Resolutions

## 7.1 The Paradoxes

We have seen that it is theoretically possible to create time machines, within the framework of general relativity, using special spacetime geometries such as warp drives and wormholes. It is unclear if these exotic geometries can exist in reality, especially due to the requirement of exotic matter. However, let us now assume that some sufficiently advanced civilization

has succeeded in building some kind of time machine. This seems to inevitably give rise to paradoxes, which we will now describe[62].

### 7.1.1  Consistency Paradoxes

A *consistency paradox* is any situation where changing the past removes the conditions which allowed (or required) changing the past in the first place, or when an event happens if and only if it does not happen. The most familiar example is called the *grandfather paradox*. In this paradox, the time traveler goes into a time machine in 2019 and travels back to a time before his or her grandparents met, let's say 1939. The time traveler then prevents this meeting, thus ensuring that one of his or her parents will never be born[63]. But if the time traveler was never born, then he or she could not have gone back in time and prevent his or her grandparents from meeting when they did... In other words, the meeting happened **if and only if** it did not happen, which doesn't make any sense.

Here is a more concrete example of a consistency paradox. The advantage of this example is that it uses inanimate objects instead of people, and thus leaves "free will"[64] out of the equation. In the grandfather paradox as we formulated it above, one could always suggest that the time traveler simply changes his or her mind, but it is impossible for an inanimate object to "change its mind".

Consider a uranium lump with mass $m = \frac{2}{3}m_{\text{crit}}$, where $m_{\text{crit}}$ is the critical mass – which, if exceeded, will cause a nuclear explosion. The lump follows a closed timelike curve: it goes into a time machine, comes out a short time earlier, and then collides with its younger itself. At that instant the total mass of both lumps together will exceed the critical mass, creating an explosion. This, of course, means that the younger lump will not reach the time machine (or perhaps even the time machine itself will be annihilated by the explosion), and thus the collision will, in fact, not take place. Again, this is a situation where the explosion happens **if and only if** it does not happen.

### 7.1.2  Bootstrap Paradoxes

A *bootstrap paradox* is any situation where something (information, an object, etc.) is "created from nothing", or when an effect is its own cause. For example, just as I began writing these lecture notes, a future version of myself appeared and gave me a USB stick with the LaTeX file for the final version of the lecture notes. In the future, when I invent my time machine, I will make sure to go back to that same day and give my past self the same USB drive with the same lecture notes. Everything in this scenario is perfectly consistent[65]. However, one might

---

[62]For an extensive and detailed analysis of time travel paradoxes from the philosophical point of view, the reader is referred to the book by Wasserman [25].

[63]In most versions of the paradox the time traveler kills his or her grandfather, but that is not necessary in order to create a paradox. In fact, even if the meeting was not prevented but just delayed a bit, the Butterfly Effect still guarantees that even if the prospective grandparents do have a child, it will not actually be the time traveler's parent, due to different initial conditions.

[64]Whatever that means...

[65]It should be noted, though, that avoiding a consistency paradox in this situation is in fact basically impossible. No matter how hard I try to look and act the same as my future self did when I met him, there is no way I could remember all the details perfectly; and even if I did, there is absolutely no way I can guarantee that **every single atom** in my body is exactly the same as it is supposed to be. Furthermore, the USB drive will inevitably wear down; even if the data stored on it has not changed, it cannot possibly be true that every single atom in the USB drive is exactly the same as it was when I received it. Consistency paradoxes seem to be completely inevitable. Still, for the purpose of this example we may assume that the Novikov conjecture (see below) holds true and automatically guarantees that everything will be consistent. This conjecture cannot, however, prevent a bootstrap paradox; the only known way of avoiding both types of paradoxes, while still allowing time travel, is using multiple timelines (see below).

still wonder: who actually wrote the notes? It seems that both information (the lecture notes) and a physical object (the USB drive) were created from nothing.

As another example [26], consider a wormhole time machine. A billiard ball comes out of the past mouth and travels directly into the future mouth. From the point of view of an external observer, the billiard ball only exists for the short period of time where it travels between the two mouths. It never existed before that, and will never exist after that; it was effectively created from nothing. From the point of view of the ball, it keeps moving in an endless loop between the two mouths. Neither point of view is compatible with what we are used to in the absence of time travel[66].

Bootstrap paradoxes are not necessarily "true" paradoxes; Krasnikov [3] refers to them as "pseudo-paradoxes". They are not as disturbing as consistency paradoxes, since they do not seem to pose logical inconsistencies, only mild discomfort. Furthermore, bootstrap paradoxes seem to require assumptions which may not themselves make sense. In both examples provided here, we never specified how such a loop would exist in the first place, so perhaps there is no reason to assume these scenarios are actually possible to create.

If time travel is possible, the paradoxes described above must not be true paradoxes, since the universe, after all, has to make sense. Let us discuss some options suggested in the literature in attempt to resolve these paradoxes.

## 7.2 The Hawking Chronology Protection Conjecture

The *Hawking chronology protection conjecture* [27] suggests that time travel is simply impossible. For example, while wormholes might in principle exist, perhaps they cannot be used to create a time machine. As we get closer to building a time machine by creating a time shift between the mouths, something will inevitably happen that will cause everything to break down before the time machine is constructed. Note that this also means we shouldn't be able to (globally) travel faster than light using wormholes, since then we could potentially exploit the same scenario described in Fig. 1 in order to time travel[67].

In quantum field theory, divergences appear everywhere, but usually it is possible to "fix" them through a method called *renormalization*. However, if a quantity still diverges even after renormalization, there is no known way to get rid of that divergence. Kim and Thorne [34] have shown that the renormalized energy-momentum tensor diverges when approaching the Cauchy horizon. This could prevent time machines from forming.

However, Kim and Thorne also conjectured that such divergences should get cut off by quantum gravity effects. Furthermore, other authors, such as Krasnikov [35], have found spacetimes containing time machines where the energy-momentum tensor is in fact bounded near the Cauchy horizon. Therefore, Kim and Thorne's result is not universal.

Grant [36] studied a particular causality-violating spacetime, originally by Gott [37], and found a similar divergence. Furthermore, Kay, Radzikowski, and Wald [38] proved theorems which show that the renormalized expectation value of a scalar field and its energy-momentum tensor are ill-defined or singular in the presence of time machines; however, Krasnikov [39] showed that this can be avoided by replacing some assumptions.

---

[66]Again, it seems impossible to avoid a consistency paradox even in this case. Much like the USB drive wearing down, the ball also changes between cycles – for example, it radiates heat, so it will have less energy in each cycle.

[67]It is interesting to note that recently, Maldacena, Milekhin, and Popov [28] found a traversable wormhole solution which seems realistic – it does not require more negative energy than is possible to realistically generate using quantum fields – but it is a "long" wormhole, which means that its throat is at least as long as the distance between the mouths in the space in which the wormhole is embedded. Therefore it does not provide a shortcut through space, and cannot lead to violations of causality. Perhaps it is a general result that only "long" wormholes are realistic, while "short" wormholes, which may allow time travel, are forbidden. However, at the moment there is no proof of this. The interested reader is referred to [29–33] for further information.

The chronology protection conjecture definitely seems plausible. However, even if it is true, we are nowhere near close to proving it, in part because we do not yet have the right tools to do so. A full theory of quantum gravity, which will tell us exactly how spacetime behaves at the quantum level, would most likely be necessary to prove this conjecture. Currently, the most we can do is semi-classical gravity – which involves quantum matter, but classical gravity.

Many papers have been published on Hawking's conjecture. However, they all make use of quantum field theory, which is beyond the scope of these notes. Therefore, we will not discuss them further here. For more information, the reader is referred to the reviews by Visser [40] and by Earman, Smeenk, and Wüthrich [41].

## 7.3 Multiple Timelines

### 7.3.1 Introduction

Another possible solution – which is perhaps the most exciting one – is to assume that the universe may have more than one history or timeline. Whenever time travel occurs, the universe *branches* into two independent timelines. The time traveler leaves from one timeline and arrives at a different timeline. Both timelines have the same past, up to that point in time, but their futures may differ[68].

If the traveler prevents his or her grandparents from meeting, this will happen in the **new** timeline – which does **not** causally influence the original timeline. It is perfectly consistent for the time traveler to not be born in this new timeline, since he or she entered the time machine in the **original** timeline, not in the new one. Hence, there are no paradoxes. Similarly, when the lump of uranium travels into the time machine, it emerges at an earlier point in time in a new timeline. In this new timeline, it meets a copy of itself and explodes. This again creates no paradox, since the explosion in the new timeline does not prevent the lump from going into the time machine in the original timeline. In the same way, multiple timelines can solve all known consistency paradoxes.

In the case of the bootstrap paradox, in timeline 1 I sat down for weeks and worked day and night on these lecture notes. Exhausted but satisfied, I copied them onto a USB drive and got into my time machine. I arrived in timeline 2, at the exact moment when I began writing the notes, and gave my past self (actually, a **copy** of myself) the finished notes. It is clear, in this case, that information did **not** come from nothing; it was created by the original me, who lived in timeline 1. In other words, while information is not conserved in each timeline individually, it is conserved in the union of both timelines[69].

In order to avoid any possibility of a paradox, a new timeline must be created whenever a time machine is used, and one may never return to their original timeline, unless they return to a moment after they left. This can be understood by noticing that the way multiple timelines solve paradoxes is by "opening up" closed timelike curves; a curve which starts at $t = 0$, goes to $t = 1$ and then loops back to $t = 0$ will inevitably create a paradox, but if it goes back to $t = 0$ in a different timeline (and hence a different point on the manifold), no loop will be formed. Therefore, to avoid paradoxes there should be at least enough timelines to "open up" every possible CTC.

### 7.3.2 Non-Hausdorff Manifolds

It is obvious that models of multiple timelines would solve time travel paradoxes. However, it's one thing to describe this solution in words; it's another thing entirely to actually write

---

[68]If this reminds you of the Everett ("many-worlds") interpretation of quantum mechanics, you're in good company; see below how this interpretation was used by Deutsch as a potential solution to time travel paradoxes.

[69]Furthermore, my copy in timeline 2 doesn't have to go back to the past and continue the cycle in order to avoid a consistency paradox.

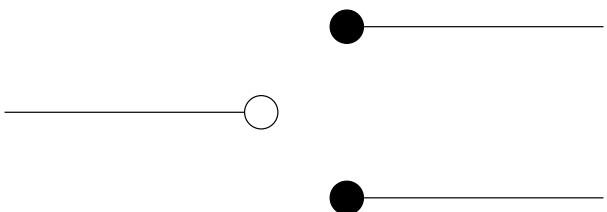

Figure 7: A 1-dimensional manifold which is locally Euclidean, but not Hausdorff. The line on the left is $(-\infty, 0)$, and it does not include the point 0 (the white disk). The two lines on the right are $[0_1, +\infty_1)$ and $[0_2, +\infty_2)$, and they do include the points $0_1$ and $0_2$ respectively (the black disks). Both of these points are in the closure of $(-\infty, 0)$.

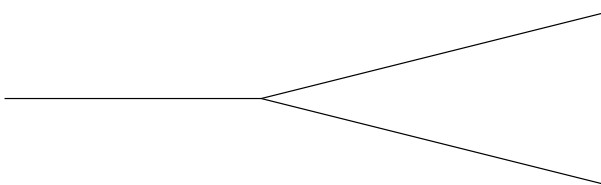

Figure 8: A 1-dimensional manifold which is Hausdorff, but not locally Euclidean at the point of branching.

down a well-defined mathematical model.

In the literature, multiple timelines are often said to be possible if one relaxes the *Hausdorff condition* in the definition of the spacetime manifold. A topology satisfies the Hausdorff condition (or "is Hausdorff") if and only if for any two distinct points $x_1 \neq x_2$ there exist two open neighborhoods $\mathcal{O}_1 \ni x_1$ and $\mathcal{O}_2 \ni x_2$ such that $\mathcal{O}_1 \cap \mathcal{O}_2 = \emptyset$. This property is useful since it allows us to separate any pair of points by finding neighborhoods that are sufficiently small to not intersect each other. In a non-Hausdorff manifold, it is possible that no two such neighborhoods exist for particular pairs of points.

A well-known example is the *branching line*, which is obtained by taking the real line $\mathbb{R}$, removing the interval $[0, +\infty)$, and replacing it with two copies of itself, $[0_1, +\infty_1)$ and $[0_2, +\infty_2)$. The topology of this manifold is Hausdorff everywhere, except for the pair of points $0_1, 0_2$. Indeed, since any possible neighborhood of each of these two points must contain at least some portion of the interval $(-\infty, 0)$, one cannot find two neighborhoods which do not intersect. This example is illustrated in Fig. 7.

Another option, discussed by McCabe [42], is to use a manifold which is Hausdorff, but not homeomorphic to Euclidean space at the point of branching. In other words, the manifold is not *locally Euclidean* at that point. A manifold in the shape of a "Y" is Hausdorff but not locally Euclidean at the point of branching, since there is no neighborhood of that point which is homeomorphic to $\mathbb{R}$. This example is illustrated in Fig. 8.

It is unclear which option is better for defining branching timelines, a non-Hausdorff manifold or a non-locally-Euclidean manifold; it might also turn out that a completely different mathematical structure is required. In fact, at the time of writing there do not exist any actual models in the literature which use either type of manifold in order to resolve paradoxes in any well-defined mathematical way.

Recently, Placek [43] developed a detailed theory of branching spacetime which employs non-Hausdorff (but locally Euclidean) manifolds in order to take multiple histories into account, although he did not discuss time travel paradoxes in this context. Luc [44] (see also [45] with Placek) discussed the issue of non-Hausdorff manifolds and concluded that they can, in

fact, be used as fundamental mathematical objects describing spacetime in general relativity.

The basic notion of using such manifolds to describe branching universes with multiple timelines seems reasonable, but there are many conceptual and mathematical issues which need to be resolved first. For example:

1. What exactly is the physical mechanism which causes the branching?

2. Realistic time machines (unlike those we usually see in science fiction) don't allow one to jump discretely from one point in spacetime to another. Instead, as we have seen, the time traveler moves continuously along a closed timelike curve which doesn't generally have a well-defined beginning and end. At which point along this curve does the branching actually happen?

Clearly, much more work is needed in order to construct a well-defined multiple-timeline solution to the paradoxes, and it seems that this will inevitably require some major modifications to our current theories of physics.

### 7.3.3 Deutsch's Quantum Time Travel Model

A consistency paradox occurs when something has to both happen and not happen at the same time. Classically, this is impossible; however, in quantum mechanics, we know perfectly well that a system can be in a superposition of two (or more) different states at once. Perhaps, then, we could solve paradoxes by invoking this and/or other properties of quantum mechanics?

It turns out that it's not as easy as just using superposition. However, Deutsch [46] famously showed that consistency paradoxes can be avoided if one uses mixed states instead of pure states. Let us recall what that means. A quantum system is said to be in a *pure state* if its state, $|\psi\rangle$, is known exactly. However, if we only know that it could be in the states $|\psi_i\rangle$ with probabilities $p_i$ (where $i$ indexes the possible states), then the system is said to be in a *mixed state*, which is given by a *density operator* (or *density matrix*):

$$\rho \equiv \sum_i p_i |\psi_i\rangle \langle \psi_i|. \tag{233}$$

Note that we always have $\text{tr}(\rho) = 1$, since probabilities sum to 1. However, one can check that a pure state satisfies $\text{tr}(\rho^2) = 1$ while a mixed state has $\text{tr}(\rho^2) < 1$.

We also need to define the partial trace. Given two physical systems 1 and 2, with density operators $\rho_1$ and $\rho_2$, we can describe their *joint state* as another density operator $\rho_{12}$. Then we define the *reduced density operator* for system 1 by

$$\rho_1 \equiv \text{tr}_2(\rho_{12}), \tag{234}$$

where $\text{tr}_2$ is the *partial trace* with respect to system 2. In other words, we "trace out" the information about system 2, which means we are left with system 1 alone. Let $|A_1\rangle$, $|B_1\rangle$ be states of system 1 and $|A_2\rangle$, $|B_2\rangle$ states of system 2. Then we define the partial trace by

$$\text{tr}_2(|A_1\rangle \langle B_1| \otimes |A_2\rangle \langle B_2|) = |A_1\rangle \langle B_1| \langle B_2|A_2\rangle. \tag{235}$$

Since $\langle B_2|A_2\rangle = \text{tr}(|A_2\rangle \langle B_2|)$ is just a number, we are left with a density matrix for system 1 alone. By demanding that $\text{tr}_2$ is linear, we are able to calculate the partial trace of any joint mixed state by looking at it term by term.

Now, consider the following paradox given by Deutsch. First of all, a *qubit* is a generalization of a classical bit, which is described by a superposition of two basis states: $|0\rangle$ and $|1\rangle$. This paradox involves a gate such that two qubits enter it and two leave it. Since this system involves time travel, this might get a bit confusing, so let us label the two input qubits and two output qubits as follows:

- $I_1$ = first input qubit,

- $I_2$ = second input qubit,

- $O_1$ = first output qubit ($I_1$ after undergoing a transformation),

- $O_2$ = second output qubit ($I_2$ after undergoing a transformation).

$I_1$ is the only part we have control of – it is the qubit sent by the experimenter into the gate, and we assume it's in a pure state $|x\rangle$, where $x \in \{0, 1\}$. $I_2$ is another qubit, in the pure state $|y\rangle$, where $y \in \{0, 1\}$. Both $I_1$ and $I_2$ pass through the gate, which imposes the following interaction:

$$I_1 \otimes I_2 \Longrightarrow O_1 \otimes O_2, \tag{236}$$

given by

$$|x\rangle \otimes |y\rangle \Longrightarrow |1 - y\rangle \otimes |x\rangle. \tag{237}$$

Now, to create a paradox, we assume that after passing through the gate, $O_1$ goes through a time machine – and **becomes** $I_2$:

$$O_1 = I_2. \tag{238}$$

In other words, the experimenter's qubit $I_1$ goes through the gate, leaves as $O_1$, and then travels back in time as $I_2$ and interacts with itself. After going through the gate once more, it becomes $O_2$. Exactly one qubit, $I_1$, enters the lab and exactly one qubit, $O_2$, leaves the lab. By looking at the interaction, we see that this means we must have

$$|y\rangle = |1 - y\rangle. \tag{239}$$

This is obviously inconsistent, and it is something we have no control of, since it does not depend on the initial condition $|x\rangle$. Therefore, we have created a consistency paradox.

To resolve this paradox, let us now generalize the system by allowing the qubits to be described by mixed states. We assume that $I_1$ is initially in a pure state $|\psi\rangle$, which is some superposition[70] of $|0\rangle$ and $|1\rangle$. Then its density operator is $|\psi\rangle \langle\psi|$. Let $I_2$ be described by a mixed state with some density operator $\rho$. Then, at the gate entrance, the two qubits $I_1$ and $I_2$ have the joint density operator

$$\rho_{\text{input}} = |\psi\rangle \langle\psi| \otimes \rho. \tag{240}$$

The gate is described by the following unitary operator (check this!):

$$U \equiv \sum_{x,y \in \{0,1\}} |1 - y\rangle \otimes |x\rangle \langle x| \otimes \langle y|, \tag{241}$$

such that the two qubits $O_1$ and $O_2$ leave the gate with the density operator

$$\rho_{\text{output}} = U \left( |\psi\rangle \langle\psi| \otimes \rho \right) U^\dagger. \tag{242}$$

Now, we want the density operator of $O_1$ to be the same as that of $I_2$. Let us therefore take the partial trace of $\rho_{\text{output}}$ with respect to $O_2$, which then leaves us with the density operator of $O_1$ alone, and demand that it is the same as $\rho$, the density operator of $I_2$:

$$\rho = \text{tr}_2 \left( U \left( |\psi\rangle \langle\psi| \otimes \rho \right) U^\dagger \right). \tag{243}$$

---

[70]That is, $|\psi\rangle = \alpha |0\rangle + \beta |1\rangle$ where $|\alpha|^2 + |\beta|^2 = 1$, but the values of $\alpha$ and $\beta$ are irrelevant to the discussion.

The reader should use (235) to calculate the partial trace. It turns out that there is a family of solutions to this equation (i.e. fixed points of the operator on the right), given by:

$$\rho = \frac{1}{2}\left(1 + \lambda\left(|0\rangle\langle 1| + |1\rangle\langle 0|\right)\right), \qquad \lambda \in [0,1]. \tag{244}$$

Therefore, regardless of the initial state $|\psi\rangle$, there is always a consistent solution to the evolution of the system, and we have avoided a paradox. Furthermore, Deutsch proves that there is always a solution for **any** $U$, which makes this solution apply to any other paradoxes one could define in this particular setting.

How is all this related to multiple timelines? For that, we need to employ the *Everett* or *"many-worlds" interpretation* of quantum mechanics [47]. Consider a qubit in the state $\alpha|0\rangle + \beta|1\rangle$ and an observer. After the observer measures the qubit, the joint state of the qubit and the observer is

$$\alpha|0\rangle \otimes |\text{Observer measured "0"}\rangle + \beta|1\rangle \otimes |\text{Observer measured "1"}\rangle. \tag{245}$$

Without going into details, we can interpret this as if the observer (and eventually, after interacting with the environment, the entire universe) "branched" into two independent (and non-interacting) histories or timelines, one in which the measurement yielded 0 and another in which it yielded 1. Usually, it is impossible to travel between these timelines. However, in Deutsch's model, one can interpret the CTC as connecting one timeline to another and allowing the qubit to travel between them. This provides a natural mechanism for creating multiple timelines [48][71].

There are several problems with Deutsch's solution, as pointed out by Deutsch himself. One of the main problems is that, while unitarity and linearity are two crucial properties of quantum mechanics, the equation of motion (243) is neither unitary (due to the partial trace) nor linear (since we are only allowing the specific states who solve this equation to exist). This has some strange effects, such as allowing both classical and quantum computers to do incredibly powerful computations very easily (as shown by Aaronson and Watrous [49]), allowing the violation of the no-cloning theorem, and so on. Accepting this model as a solution to time travel paradoxes thus requires modifying quantum mechanics in a very significant way, at least in the vicinity of CTCs[72].

Moreover, while (243) always has a solution, it does not, in general, have a **unique** solution, and it is thus unclear which one of the solutions will be realized in nature. Indeed, in the case described above there is a solution for any value of $\lambda \in [0,1]$. This is a version of the bootstrap paradox, since the information about the value of $\lambda$ seems to be created out of nothing and is completely independent of anything else in the system. Deutsch suggests that the state with the maximal entropy should be the unique solution (in this case it is the maximally mixed state, corresponding to $\lambda = 0$) but it is unclear why this must be so.

In addition, one may wonder if Deutsch's solution works for general objects that are not test particles, that is, objects which have enough mass or energy to noticeably influence the curvature of spacetime. This would allow, for example, "turning off" the time machine in one of the timelines, which seems to break the model. It seems that a theory of quantum gravity, which should allow the geometry of spacetime itself to be in a superposition, would be necessary for Deutsch's model to work in this case.

---

[71]Note that we secretly assumed the initial state (240) is a tensor product of the states of $I_1$ and $I_2$. This means that those two states are not correlated, and this is only possible if $I_1$ and $I_2$ are completely independent of each other. Hence, $I_2$ must have come from a different timeline.

[72]Note that, as Deutsch himself mentions in his paper, using this model does not actually allow us to avoid the mathematical issues regarding branching spacetime manifolds discussed in the previous subsection.

## 7.4 The Novikov Self-Consistency Conjecture

### 7.4.1 Classical Treatment

The *Novikov self-consistency conjecture* [50] states that "the only solutions to the laws of physics that can occur locally in the real universe are those which are globally self-consistent". Alternatively, it may be understood as the statement that the universe has only one history (or timeline), and it must be consistent, no matter what.

For example, the time traveler who tries to prevent his or her grandparents from meeting will simply find that the universe has conspired to make him or her automatically fail in this task, since anything else would lead to an inconsistency. In fact, the time traveler **already** tried it and **already** failed... Perhaps even his or her attempt at meddling is what **caused** their grandparents to meet in the first place[73]!

A simple toy model, known as *Polchinski's paradox*, is given by a billiard ball which goes into a time machine, travels to the past, and knocks its past self out of its original path – thus preventing itself from going into the time machine in the first place. Although this seems like a paradox, Echeverria, Klinkhammer, and Thorne [51] found that it has self-consistent solutions, where the ball collides with itself and knocks itself into the time machine in just the right angle to product the former collision[74].

Note that such solutions cannot exist for the uranium lump paradox, since it will always explode independently of the trajectories or angles of collision. Nevertheless, Novikov [52] described a similar problem with an exploding ball and showed that there could still be a self-consistent evolution by keeping track of the fragments of the ball after the explosion. It was also conjectured by Frolov, Novikov, and others [53] that the principle of self-consistency could be a consequence of the *principle of least action*, and this was demonstrated using the billiard ball scenario. Rama and Sen [54] attempted to generalize the billiard ball model in order to create a paradox that cannot be solved; however, Krasnikov [55] showed that their model does in fact have self-consistent solutions.

From these results, one may deduce that the there might be self-consistent solution for any possible system. However, this turns out not to be the case; it is possible to construct paradoxical scenarios for which there are no known self-consistent solutions. One such example is given by Krasnikov [56]. It generalizes the billiard ball scenario, in 1+1 dimensions, by considering the trajectories of particles which have specific properties and follow specific rules of interactions. This is an example of a **true** paradox, to which the Novikov conjecture does not seem to apply, as far as we know.

Still, the Novikov conjecture seems to make sense. Indeed, if the Hawking conjecture is incorrect and time travel exists, and if there is only one possible timeline, then it's hard to see any other way to avoid time travel paradoxes. However, if it is true, it may lead to very peculiar scenarios.

For example, let us assume that the Novikov conjecture is correct. Then, if I travel 10 years into the past and try really hard to create a paradox by killing my past self, I will find that my murder attempts always fail for some reason. For instance, if I use a gun, then I will inevitably find that it gets jammed, or that the bullet misses the target. At first, I might consider this a temporary fluke. However, over the next 10 years, every single day, I try to kill my past self 100 times per day. If every single one of those 365,000 assassination attempts fails, no matter what the circumstances are (weapon used, location, time of day, etc.), this might start to seem just a tiny bit ridiculous!

In fact, in this class of thought experiments, the number of failed attempts may be made arbitrarily large. Consider some measurement which may be performed in a very short time

---

[73]This is a common trope in science fiction.

[74]In fact, there is an **infinite** number of self-consistent solutions for each initial condition.

and has two outcomes of equal probability – e.g., the measurement of a qubit prepared in the quantum[75] state $(|0\rangle + |1\rangle)/\sqrt{2}$. If we measure $|0\rangle$ nothing happens, but if we measure $|1\rangle$, something happens which creates a paradox. If the Novikov conjecture is true, then we will find that the outcome of the measurement is **always** $|0\rangle$. We repeat this process over and over, producing $N$ outcomes of $|0\rangle$ and exactly zero outcomes of $|1\rangle$. As $N \to \infty$, the probability for this sequence of outcomes approaches zero.

The fact that the Novikov conjecture implies bizarre situations such as the ones we described here does not mean that it is wrong; indeed, we have seen above that Deutsch's multiple-timeline model has very weird implications of its own. However, our discussion does imply that the Novikov conjecture would require modifying quantum mechanics such that it works differently when time travel is involved, similar to Deutsch's modification of quantum mechanics in the case of multiple timelines. Let us now see one example of such a modification.

### 7.4.2 The Postselected Quantum Time Travel Model[76]

Deutsch's model represents one possible way of making quantum mechanics consistent in the presence of CTCs. In addition to his model, which is referred to as *D-CTCs*, there is another model called "postselected" CTCs or *P-CTCs*, described by Svetlichny [57] and by Lloyd et al. [58, 59]. In this model, one **simulates** time travel using a *quantum teleportation* protocol. Let us recall how that works [60].

At time $t = 0$, a *Bell state* (or *EPR pair*) of entangled qubits is created:

$$|\beta_{00}\rangle \equiv \frac{1}{\sqrt{2}}(|0\rangle \otimes |0\rangle + |1\rangle \otimes |1\rangle) \equiv \frac{1}{\sqrt{2}}(|00\rangle + |11\rangle), \tag{246}$$

where we used the shorthand notation $|xy\rangle \equiv |x\rangle \otimes |y\rangle$. Alice takes the first qubit, Bob takes the second, and they go their separate ways. Later, at time $t = 1$, Alice receives some qubit $|\psi\rangle$ which she knows nothing about, and she needs to transfer it to Bob using only classical channels.

Since Alice only has one copy of this qubit, and cloning a quantum state is forbidden by the no-cloning theorem, she can't learn anything about the qubit without destroying it. Furthermore, even if she knew the precise state of the qubit, she wouldn't be able to relay that information classically to Bob, since a qubit is described (up to overall phase) by a complex number, and a general complex number is described by an infinite number of classical bits.

However, Alice can use the fact that she has a qubit which is entangled with Bob's. If

$$|\psi\rangle = \alpha |0\rangle + \beta |1\rangle, \tag{247}$$

then all three qubits are represented together by the state

$$|\Psi\rangle \equiv |\psi\rangle \otimes |\beta_{00}\rangle = \frac{1}{\sqrt{2}}(\alpha |0\rangle + \beta |1\rangle) \otimes (|00\rangle + |11\rangle), \tag{248}$$

where the first qubit is $|\psi\rangle$, the second is Alice's, and the third is Bob's. Alice now sends both qubits through a CNOT gate and the first qubit through a Hadamard gate (for our purposes it doesn't matter what these gates do exactly). After passing through these gates, the state of the three qubits becomes

$$|\Psi\rangle \mapsto \frac{1}{2}(|00\rangle (\alpha |0\rangle + \beta |1\rangle) + |01\rangle (\alpha |1\rangle + \beta |0\rangle) + |10\rangle (\alpha |0\rangle - \beta |1\rangle) + |11\rangle (\alpha |1\rangle - \beta |0\rangle)). \tag{249}$$

---

[75]A quantum measurement is preferred for this thought experiment, since unlike a classical coin toss, which is deterministic, the result of a measurement of spin is (as far as we know) truly random.

[76]The author would like to thank Daniel Gottesman and Aephraim Steinberg for discussions which proved helpful in writing this section.

Now Alice performs a measurement on her two qubits, and she will obtain one of four results: 00, 01, 10, or 11. These are two classical bits, which she can send to Bob. With this information, Bob can read from (249) exactly which operations he has to perform in order to obtain $|\psi\rangle$ from the qubit he has.

In particular, if Bob receives the bits 00, then we can see from (249) that he **already had** the qubit $|\psi\rangle = \alpha |0\rangle + \beta |1\rangle$ at time $t = 0$, even **before** Alice obtained it at time $t = 1$! The idea of P-CTCs is that we *postselect* the 25% of the cases where the classical bits happened to be 00, ignoring the other cases. While this is not actually time travel, it can be used to **simulate** time travel; in the case where the bits are 00, the state Bob has in the past is the same state Alice will get in the future, so it's as if Alice's state "went back in time" through a time machine. Furthermore, one may now conjecture that any time machine behaves, in a sense, like a quantum teleportation circuit which automatically projects the result of Alice's measurement on 00.

Both D-CTCs and P-CTCs suggest different ways in which quantum mechanics may be modified in the presence of CTCs, and they are both incompatible with the standard formulation of quantum mechanics, for example because the evolution is non-unitary – in D-CTCs due to the partial trace and in P-CTCs due to the projection, both of which happen at the CTC itself. One of them could be true, or neither of them could be true; probably the only way to know is to actually build a time machine.

While D-CTCs deal with paradoxes using multiple timelines, where the different timelines correspond to different "worlds" in the "many-worlds" interpretation, P-CTCs deal with paradoxes by assuming Novikov's self-consistency conjecture. This can be described in a simplistic way as follows. The grandfather paradox can be modeled by having the state $|\psi\rangle$ enter the time machine as $|1\rangle$, which represents the time traveler being alive, and, at some point between $t = 0$ and $t = 1$, turning it into $|0\rangle$, which represents the time traveler being dead. In this case, there is no measurement where Bob's state $|0\rangle$ is the same as Alice's state $|1\rangle$; the probability for the measurement to yield 00 is exactly zero.

Since the probability is zero, postselection doesn't work – you can't postselect an event with zero probability[77]. However, one can then add small perturbations (or quantum fluctuations) to the system. This will make the probability non-zero, even if still extremely small. Since the CTC projects the state on 00, even an extremely small probability for the state we are projecting on suddenly becomes probability 1! This is exactly the idea behind Novikov's conjecture – events with extremely small probabilities, such as my gun getting jammed every single time I try to kill my past self and create a paradox, somehow get selected over events with much higher probability. Of course, all this relies on time machines actually working in the same way as P-CTCs do, which is something that at the moment we have no reason to believe. Still, this provides a possible mechanism for Novikov's conjecture to be realized in nature.

## 7.5 Conclusions

As we have seen, the possibility of time travel has, thus far, not been ruled out by known physics. If Hawking's conjecture is wrong, and time travel is indeed possible, then it seems that paradoxes inevitably occur. Given that we want reality to make sense, there must be a way in which the universe avoids these paradoxes.

---

[77]Recall that the *conditional probability* of an event $A$ given an event $B$ is defined as

$$P(A|B) \equiv \frac{P(A \cap B)}{P(B)}. \tag{250}$$

If $P(B) = 0$, this expression is undefined, and therefore we are not allowed to postselect for $B$; this will make all other probabilities in our system undefined.

If our universe can have multiple timelines, then we must radically modify our laws of physics to accommodate this. Perhaps this modification can be made at the classical level, by replacing the spacetime manifolds we know and love with something more complicated, such as branching manifolds. If the "many-worlds" interpretation is true, it could provide a natural way for additional timelines to be created, using a modification of quantum mechanical such as the D-CTC model.

If there is only one timeline in our universe, then it seems that the Novikov conjecture must be true (or at least, we have not come up with any alternatives so far). Perhaps self-consistent solutions can always be found for any conceivable physical system at the classical level, although given Krasnikov's unsolvable paradox, this would probably require some modification of our laws of physics as well. A modification of quantum mechanics, as suggested in the P-CTC model, could provide a mechanism for events with infinitesimal probability to nonetheless always occur if that is required for the system's consistency.

Hawking famously held a party for time-travelers, to which no one showed up, and sent out invitations only after the party was over. He jokingly referred to this as "experimental evidence" for his conjecture. However, it is interesting to note that the alternatives we discussed above may also accommodate this experimental result. Novikov's conjecture would simply prevent the time travelers from attending in order to preserve consistency (maybe they all inevitably get stuck in traffic?), while multiple timelines will result in the party being full of time travelers, but only in a timeline different from the one we currently live in.

## 8   Further Reading

For more information about faster-than-light travel and time travel, the reader is referred to the books by Visser [1], Lobo [2] and Krasnikov [3]. For a discussion of time travel paradoxes from the philosophical point of view, see the book by Wasserman [25]. For qualitative discussions at the popular-science level, see the book by Everett and Roman [16] and the three books by Nahin [61–63].

## 9   Acknowledgments

The author would like to thank the students who attended this course for asking interesting questions and providing insightful comments which helped improve and perfect these notes. The author would also like to thank the referee Aron Wall for helpful comments and suggestions.

Research at Perimeter Institute is supported in part by the Government of Canada through the Department of Innovation, Science and Economic Development Canada and by the Province of Ontario through the Ministry of Economic Development, Job Creation and Trade.

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
