# Peer review of "Lectures on Faster-than-Light Travel and Time Travel"

_SciPost Physics Lecture Notes, doi:SciPost Phys. Lect. Notes 10 (2019)_

## Round 1 · Referee Report · Aron Wall · 2019-8-22

Strengths

1. Accessible to undergraduates
2. Long, complete introduction to necessary background material

Weaknesses

1. A great deal of important work on this topic has not been cited or discussed.
2. Focusses mainly on introducing the material but does not discuss questions of interpretation in very sophisticated way

Report

This is a good introductory set of lecture notes suitable for undergraduates to learn more about causality in relativity. It has a very complete introductory section to ensure that undergraduates will have the necessary pre-requisites to understand the more advanced topics covered later.

While I understand that there is a limit to the topics that can be included in a description for undergraduates, there were a few places where I thought that important work has not been properly discussed. The lecture notes would be stronger if the following points were acknowledged (even if only briefly) and cited:

1. The fact that the non-minimally coupled scalar field is equivalent to the minimally coupled one by means of a field redefinition, i.e. Einstein frame versus Jordan frame.

(This equivalence is valid only in a certain range of the fields, and construction of causality violations requires taking the field outside this range, which a skeptical minded person could argue is unphysical since the effective Newton's constant becomes negative. Note also that the definition of the NEC is not invariant under this transformation, but if the NEC holds in any frame then you can use to prove certain casuality violations are impossible.)

2. The author claims there is no proof of energy conditions, but in fact that the ANEC has now been proven to hold for general Lorentz invariant QFTs in Minkowski space, see for example these recent proofs:

https://arxiv.org/abs/1605.08072
https://arxiv.org/abs/1610.05308

There are violations of the ANEC in curved spacetime, however it is expected to hold for complete achronal null geodesics in curved spacetime on self-consistent backgrounds, yet this is sufficient to rule out a variety of causality violations. See the discussion in Graham and Olum:

https://arxiv.org/abs/0705.3193

Of course, QFT is beyond the scope of these notes but these papers should still be mentioned and cited in light of their importance for the topics being discussed.

3. Recently there has been a new interest in traversable wormholes, due to the discovery that it is possible to make them in a semiclassical setting with realistic matter fields, if you couple the two ends of the wormhole to each other. (Because the traversability depends on the coupling between the two sides, it is not possible to use this kind of wormhole to produce faster-than-light travel.)

The authors cite just one paper in this literature (Maldecena, Milikhen, and Popov) but this is part of a broader topic and more citations would be a good idea for those who wish to study the topic more deeply. (These papers also make it clear why only "long" wormholes are consistent with the achronal ANEC.)

For example, see the following articles (among many others):

https://arxiv.org/abs/1608.05687
https://arxiv.org/abs/1704.05333
https://arxiv.org/abs/1804.00491
https://arxiv.org/abs/1807.07917
https://arxiv.org/abs/1904.02187

[Disclosure: I am one of the authors of the first 2016 paper.] I recommend that the author should consider adding some or all of these citations.

4. Finally I wish the article had treated the question of Novikov consistency and branching timelines in a more sophisticated way, since there is extensive literature on these topics and only about 3 citations are engaged with.

People have done interesting work on post-selection effects / branching timelines in QM (at least I have attended multiple talks on these topics), but the author spends most of the brief section following a single paper which is worried only about the topology of the branching point in spacetime. It seems like a bit of an anticlimax to postulate branching timelines and then spend most of the time worrying about the mathematical quibble of whether there is topologically one point or two at the branch point.

Similarly, the author's main objection to Novikov consistency is that it gives different predictions from not assuming it, and that the probability overlap with standard QM goes to 0 in the limit of a large number of observations. But it is a good thing, not a bad thing, when a hypothesis makes testable predictions that are different from what would otherwise obtain. So I fail to see the objection here.
* * *
All told, these notes are a reasonable first look at the material they are trying to cover, and deserve to be published at SciPost.

Requested changes

1. Add discussion of field redefinitions of nonminimal scalar in section 3.3.
2. Add more citations (see report).

---

## Round 2 · Referee Report · Aron Wall (Referee 1) · 2019-9-14

Report

The author has taken all my suggestions into consideration, and in particular section 7 is a significantly more detailed and useful resource now. I recommend publication.

---

## Round 2 · List of Changes

I have improved the manuscript following the referee report. All of the referee's suggestions have been implemented in full. I added references to the suggested papers. In addition, the chapter on time travel paradoxes has been significantly expanded, including full and detailed treatment of D-CTCs (the quantum version of the multiple timeline solution) and P-CTCs (the quantum version of Novikov's conjecture). No other changes were made except a few minor fixes here and there.

---

## Editorial Decision

published